# Semiarid climate and hyposaline lake on early Mars inferred from reconstructed water chemistry at Gale

Keisuke Fukushi [1]*, Yasuhito Sekine [1,2], Hiroshi Sakuma [3], Koki Morida[4] & Robin Wordsworth [5]

Salinity, pH, and redox states are fundamental properties that characterize natural waters. These properties of surface waters on early Mars reflect palaeoenvironments, and thus provide clues on the palaeoclimate and habitability. Here we constrain these properties of pore water within lacustrine sediments of Gale Crater, Mars, using smectite interlayer compositions. Regardless of formation conditions of smectite, the pore water that last interacted with the sediments was of Na-Cl type with mild salinity (~0.1–0.5 mol/kg) and circumneutral pH. To interpret this, multiple scenarios for post-depositional alterations are considered. The estimated Na-Cl concentrations would reflect hyposaline, early lakes developed in $10^4$–$10^6$-year-long semiarid climates. Assuming that post-depositional sulfate-rich fluids interacted with the sediments, the redox disequilibria in secondary minerals suggest infiltration of oxidizing fluids into reducing sediments. Assuming no interactions, the redox disequilibria could have been generated by interactions of upwelling groundwater with oxidized sediments in early post-depositional stages.

[1] Institute of Nature and Environmental Technology, Kanazawa University, Kanazawa, Ishikawa, Japan. [2] Earth-Life Science Institute, Tokyo Institute of Technology, Meguro-ku, Tokyo, Japan. [3] National Institute for Materials Science, Tsukuba, Ibaraki, Japan. [4] Division of Natural System, Kanazawa University, Kanazawa, Ishikawa, Japan. [5] Department of Earth and Planetary Sciences, Harvard University, Cambridge, MA, USA. *email: fukushi@staff.kanazawa-u.ac.jp

There have been a number of attempts to estimate water chemistry (i.e., pH, salinity, redox state, and dissolved species) on early Mars by means of geochemical reaction-transport modeling[1–3]. These bottom-up modeling approaches can reproduce time and spatial variations of water chemistry based on the observed mineral assemblages. However, they need to assume initial solution compositions, reactant minerals, fluid permeability, water-rock ratios, and a series of chemical reactions. Thus, these studies involve a number of uncertainties depending on these a-priori parameters, which are always difficult to constrain on early Mars.

In contrast, a top-down approach for quantitative reconstruction of water chemistry of pore water in terrestrial clay-bearing rocks has been proposed by Gaucher et al.[4]. Using the mineralogical and chemical compositions of clay-bearing rocks and their cation exchange equilibria, this approach successfully reproduces the water chemistry of terrestrial pore waters without any a-priori parameters[4–6]. Smectite is unique in possessing exchangeable cations in interlayer sites[7]. Although there is a wide diversity of mineralogical characteristics of smectites including the chemical compositions and formation processes, all smectite possess the cation exchange capacity (Supplementary Fig. 1). The cation exchange reaction of smectite is very rapid process with sub-second timescale in suspension solutions[8,9]. The rate limiting step of the exchange reaction is diffusion and, thus, the activation energy is very low[8,9]. Therefore, the exchangeable cations in smectite can readily record the surrounding solution composition[4–6] even at low temperatures (e.g., ~0 °C). On Mars, there is a debate about the formation process of smectite in sediments, i.e., in situ formation by early diagenesis[10–13] vs. detrital deposition[3]. However, regardless of the chemical compositions and formation conditions of smectite, the top-down approach can provide a robust snapshot of the chemical properties of the pore water that finally interacts with the clay-bearing rocks[4] if the clay-bearing rocks contain smectite.

The Mars Science Laboratory rover Curiosity has analyzed the phase and chemical composition of fluvial-lacustrine sediments deposited in long-lived lakes within Gale Crater, a late Noachian to early Hesperian-aged crater[14,15]. Two drill core samples of lacustrine sediments were obtained from Yellowknife Bay Formation[12,14]. The Yellowknife Bay Formation was probably deposited by flow deceleration as water streams encountered a lake in Gale Crater[14,15] (Fig. 1a). The Yellowknife Bay sediments contain ~20 wt% of smectite[12]. These smectites would have formed in the early diagenesis within the lake[12,13] (Fig. 1b) and/or may have been of detrital origin[3] (Fig. 1a). The Yellowknife Bay sediments were subjected to burial pressure and pervasively fractured after the deposition and early diagenesis[14,15] (Fig. 1c). During the rewetting events, the late-diagenetic (post-depositional) fluids were introduced into the fractures, which are currently filled with calcium sulfates; presumably a substantial amount of time after the period of the early lakes[12,14,16] (Fig. 1d). These rewetting events occurred within Gale even after the formation of Aeolis Mons[17], a 5 km mound of layered sedimentary rock in Gale crater. After the final rewetting event, liquid water in the Yellowknife Bay sediments disappeared (Fig. 1e).

The proposed hydrogeological context suggests that the Yellowknife Bay sediments would have interacted with waters with different compositions at different stages since its deposition. Bottom water in lakes is continuously trapped within pores of sediments and buried together[18] (Fig. 1b). If groundwater largely upwelled into Gale Crater[19], this could have also affected pore water chemistry in the early post-depositional stages. After loss of liquid water trapped in pores of the Yellowknife Bay sediments, dissolved components would have been left in the sediments, e.g., as trace of evaporites (Fig. 1c). The presence of sulfate-rich

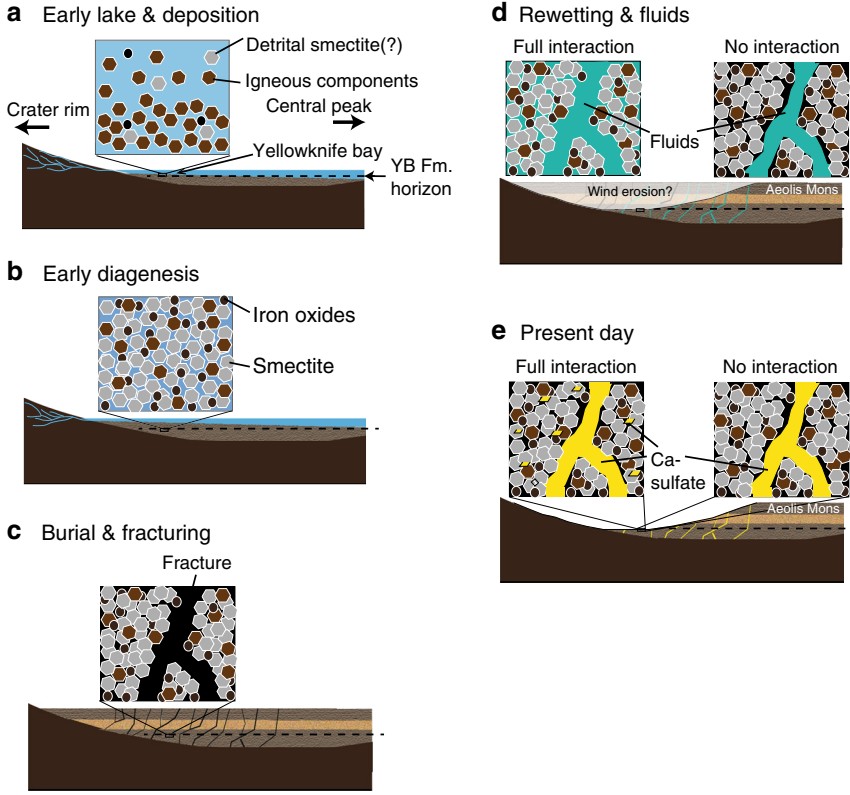

**Fig. 1** Illustrations of hydrogeological context of Yellowknife Bay Formation at Gale Crater (from **a**–**e** in time sequence). The dashed lines show the horizon of Yellowknife Bay Formation. In **d**, **e**, the full- and no-interaction scenarios are compared. See the main text for the details and references

fractures strongly suggests that the post-depositional fluids provided additional $SO_4^{2-}$ into the sediments in the last wetting event[11,16]. If the post-depositional $SO_4^{2-}$-rich fluids infiltrate into smectite-rich matrix of the sediments, the primary evaporites left in the matrix would have been re-dissolved and mixed with the additional components (Fig. 1d). The duration of the last wetting event determines whether the post-depositional fluids infiltrate into the matrix. Using both terrestrial analog permeability of dried marine clays ($10^{-7}$–$10^{-10}$ cm/s)[20] and the typical distance between the calcium sulfate fractures in the Yellowknife Bay sediments (~1 cm)[11], post-depositional fluids can chemically interact with the matrix of the Yellowknife Bay sediments in a wetting event for 1–$10^2$ years or longer through diffusion.

Here we apply the top-down approach[4] to the Yellowknife Bay sediments at Gale Crater, Mars. To quantitatively estimate the water chemistry of the pore water that finally interacted with the sediments, we use the exchangeable cation compositions of smectite's interlayer together with the presence of secondary minerals commonly found in Yellowknife Bay (see Methods). Our results show that the pore water is characterized as mildly saline, Na–Cl type water. We interpret that this salinity of the pore water originates from lake water trapped within sediments in early Gale lakes. The salinity of the lake water would have provided through hydrological cycles during warm and semiarid climate periods in the Hesperian.

## Results and Discussion

Based on the hydrogeological context (Fig. 1), we consider two end-member scenarios concerning the interactions between the primary and additional components in the sediments (see Methods): In one, the post-depositional $SO_4^{2-}$-rich fluids in the last wetting event fully interact with the sediments (the full-interaction scenario), while in the other we assume no chemical interactions between the post-depositional $SO_4^{2-}$-rich fluids and the sediments in the last wetting event (the no-interaction scenario; Fig. 1e). In the full-interaction scenario, we consider that the pore water was reacted with not only the primary components in the matrix of the sediments but also with the additional components (i.e., calcium sulfate) in veins (Fig. 1d). Thereby, the estimated water chemistry in the full-interaction scenario would reflect mixtures of the primary and additional components. In the no-interaction scenario, we assume that the observed secondary minerals in the matrix of the sediments are not influenced by the post-depositional $SO_4^{2-}$-rich fluids in the last wetting event (Fig. 1d). In the latter scenario, we estimate the chemical composition of the pore water using only the secondary minerals that are considered to be contained in the matrix of the sediments, i.e., akaganeite. The estimated pore water in the no-interaction scenario would reflect the primary components trapped within the sediments before the last wetting event.

### The full-interaction scenario.

We first examine the full-interaction scenario and discuss the water chemistry of the pore water at the last rewetting event immediately before the disappearance of liquid water (Fig. 1d). The obtained X-ray diffraction (XRD) patterns of smectite[12,13,21] provide information about exchangeable cations in the interlayer (Fig. 2). Basal spacing of smectite depends on the cationic composition in the interlayer and the relative humidity (RH). The cationic composition of smectite of John Klein and Cumberland are estimated from the peak deconvolutions based on the previous experimental studies of smectite basal spacing by RH controlled XRD analyses (see Method). In addition to smectite, the presence of calcium sulfate and akaganeite[12,14,16] constrain the composition of liquid water.

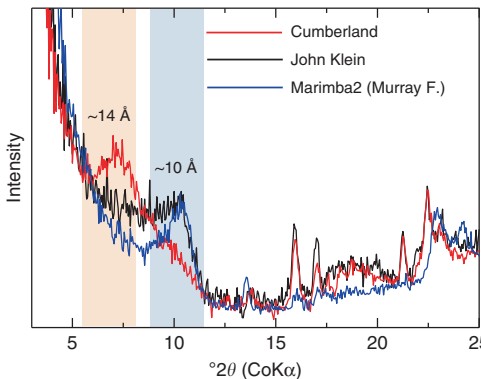

**Fig. 2** Basal spacings of smectite from John Klein (black) and Cumberland (red), compared with Marimba2 (blue) in Murray formation, obtained by Curiosity's CheMin. The 001 reflection peaks at 10 Å (blueish background) and 14 Å (orangish background) are mainly due to $Na^+$ and $Mg^{2+}$ in the interlayers, respectively. Yellowknife Bay smectites contain less $Na^+$ and more $Mg^{2+}$ in the interlayers than Murry smectites

Table 1 and Supplementary Fig. 2 show the compositions of the pore water that last interacted with the Yellowknife Bay sediments for the full-interaction scenario. The concentrations of all components from the Cumberland site overlap those from the John Klein site (Supplementary Fig. 2). This suggests that the pore-water compositions at the two adjacent sites would have been similar and may define the pore water chemistry throughout the Yellowknife Bay sediments. This is consistent with a view of the full-interaction scenario that the post-depositional fluids infiltrated into the sediments. The reconstructed similar compositions of pore water at these sites do not contradict with the fact that the top of the peaks of the 001 reflection of John Klein and Cumberland are different (Fig. 2). The peak deconvolution analyses show that the contribution of the predominant cations ($Na^+$ for John Klein and $Mg^{2+}$ for Cumberland) to the total interlayer cations are almost half (i.e., ~0.5) (Supplementary Table 1). This suggests that both of the smectites from John Klein and Cumberland are close to transition of the predominant cations in the interlayer sites, whereas that of Marimba2 is dominated by $Na^+$. In a solution within a range of compositions, a transition of the predominant cation occurs in smectite[22]. Within this range of solution composition, smectites with different of cation occupancies can co-exist. Since the pore-water composition from the Cumberland site is more constrained than those from John Klein site, the former would be the representative water chemistry of the pore water prevailing in the Yellowknife Bay sediment.

There are two possible ways for the liquid pore water to have disappeared from the sediments after the final wetting event (Fig. 1d, e). The first is drying out of the liquid pore water due to complete evaporation in a warm hyperarid climate above freezing temperature. According to this, concentrations of dissolved components should increase upon drying and finally reach saturation levels of high-solubility salts[23], such as halite. The estimated concentrations of $Na^+$ correspond to weakly-to-moderately saline (hyposaline) (0.1 mol/kg) and are far below the saturation level of halite (e.g., 10 mol/kg at ambient temperature). This suggests that the scenario for the complete evaporation in warm climate can be ruled out. The other is freezing of the liquid pore water and subsequent sublimation of ground ice in a cold climate. In this case, concentrations of salinity in remained liquid water would have also occurred upon freezing. However, freezing of pore water most likely initiates at the interfaces with solid particles[24]. This suggests that upon

**Table 1 Estimated solution compositions of pore water that finally interacted with sediments at John Klein and Cumberland for (a) full-interaction scenario and (b) no-interaction scenario**

|  | Full-interaction scenario | | No-interaction scenario | |
|---|---|---|---|---|
|  | John Klein | Cumberland | John Klein | Cumberland |
| Na (mol/kg) | 0.085–0.24 | 0.094–0.12 | 0.038–0.42 | 0.033–0.13 |
| K (mol/kg) | 0.0025–0.018 | 0.0014–0.0044 | 0.0011–0.030 | 0.00047–0.0044 |
| Mg (mol/kg) | 0.0046–0.11 | 0.035–0.060 | 0.00064–0.21 | 0.0034–0.065 |
| Ca (mol/kg) | 0.022–0.12 | 0.024–0.045 | <0.119 | <0.048 |
| Cl (mol/kg) | 0.072–0.59 | 0.11–0.25 | 0.05–1.3 | 0.05–0.34 |
| SO$_4$ (mol/kg) | 0.025–0.14 | 0.044–0.072 | N/A | N/A |
| ΣCO$_2$ (mol/kg) | 0.0019–0.041 | 0.0023–0.016 | N/A | N/A |
| Fe(II) (mol/kg) | 0.00024–0.038 | 0.00012–0.0058 | 0.00044–0.19 | 0.00011–0.013 |
| pH | 6.7–7.3 | 6.9–7.3 | 6.5–8.3 | 6.8–8.1 |
| $P_{CO_2}$ (mbar) | 6–130 | 6–40 | N/A | N/A |

The temperature is assumed to be 0 °C. Na, K, Mg, Ca, Cl, SO$_4$, Cl, ΣCO$_2$, Fe represent the total dissolved components of sodium, potassium, magnesium, calcium, chlorine, sulfate, dissolved inorganic carbon and iron, respectively

freezing, highly concentrated brine would be unable to directly interact with smectite particles. Hence, the pore-water chemistry recorded in the Yellowknife Bay sediments most likely represents saturated pore water immediately before freezing in the last wetting event.

The dissolved components in Table 1 originate from both the primary components from the pore water in the bottom sediment of the early lake and the additional components from post-depositional fluids in the rewetting events. We believe that the predominant dissolved components of Na and Cl in the pore water would have largely originated from the primary component; namely, lake water trapped within the sediments (Fig. 1a, b). This is because both Na$^+$ and Cl$^-$ behave as conservative species which are generally retained in subsequent chemical reactions (e.g., diagenesis), resulting in preservation of information about bottom water within co-buried smectite on Earth[18]. Additionally, the post-depositional SO$_4$$^{2-}$-rich fluids in the last wetting event were depleted in Na[16]. Na contents are almost zero (<0.05 wt%) at sulfate-rich veins with the highest SO$_3$ contents[16]. No enrichments of Na in sulfate-rich veins in the Murray and Stimson formations[25] may also support depletion of Na in post-depositional SO$_4$$^{2-}$-rich fluids, if the fluids are in common with Yellowknife Bay[26].

Groundwater within terrestrial basalts typically contains low Na concentrations ($10^{-4}$–$10^{-3}$ mol/kg: see Methods). Given the possibility that the primary components could have been leached due to upwelling groundwater into Gale[19], the estimated Na concentration would be a lower limit of the primary Na concentration in the bottom water of the early lakes. Nevertheless, we consider that the original Na concentration in the bottom water would not be remarkably higher than those in Table 1 because of lack of evidence for drying hypersaline lakes (e.g., desiccation cracks and rip-up chips)[27] at Yellowknife Bay[14]. We consider that if the early lakes existed, its Na$^+$ concentration would have been on the same order of magnitude as that of the pore water in the last wetting event.

Our results of the pore-water chemistry suggest that the water chemistry of the early Gale lakes would be characterized as hyposaline, at least, at the time of the Yellowknife Bay sediments deposition. The estimated Na–Cl concentrations are lower than terrestrial seawater ({Na} = 0.49 mol/kg and {Cl} = 0.56 mol/kg), but significantly higher than freshwater[23]. On Earth, hyposaline lakes are abundant in semiarid climate regions, such as inner-continental steppe areas[28,29]. Desiccation cracks and rip-up chips are not pervasive around terrestrial hyposaline lakes (see Supplementary Fig. 3). Thus, the absence of desiccation cracks

at Yellowknife Bay is compatible with our conclusion of hyposaline early Gale lakes.

Most of the terrestrial hyposaline lakes in continental areas are terminal lakes without any outflowing rivers. Lake levels of terminal lakes are maintained by a balance between inflowing water and evaporation. Through their hydrological cycles, solutes accumulate within the lakes. Our conclusion of hyposaline water suggests that the outflow of water from the Gale lakes was restricted, which is consistent with the geomorphic interpretation of the Gale lakes as potentially being closed-basin[30]. Additionally, formation of trioctahedral smectite, including saponite, usually occurs in terrestrial alkaline-saline lakes as an alteration product by early diagenesis[31] (Supplementary Table 2). Thus, the abundance of saponite in the Yellowknife Bay sediments is also supportive of hyposaline conditions for Gale lake water, although saponite formation may also occur through closed-system alterations of mafic minerals[32]. Furthermore, akaganeite, a chloride-bearing ferric oxyhydroxide, occurs with smectite in the Yellowknife Bay mudstone[11]. The formation of akaganeite requires >0.05 mol/kg of dissolved Cl concentrations in the solution[33]. Accordingly, the Cl concentrations (0.1–0.3 mol/kg from the Cumberland site) estimated by the present study are consistent with the formation conditions of akaganeite, although the akaganeite is most likely to have been formed during the rewetting events using primary Cl in the sediments in the full-interaction scenario (see below).

We constrain the pH of the pore water based on the stability relationship of pH-sensitive secondary minerals found in the sediments. Calcium carbonates are ubiquitous in terrestrial surface conditions[23] and frequently observed in terrestrial lacustrine sediments[29] regardless of their origins (authigenic or clastic). However, calcium carbonates are very low (<0.8 wt%) in the Yellowknife Bay sediments[12,34,35]. The estimated Ca concentrations in the pore water are 0.02–0.05 mol/kg from the Cumberland site (Table 1), which are high enough to produce calcium carbonates if sufficient amounts of CO$_3$$^{2-}$ are available. The absence of calcium carbonates must be ascribed to limitation of CO$_3$$^{2-}$ in the pore water in the last rewetting event, indicating that the pore water may also be characterized by low pH and/or low levels of dissolved ΣCO$_2$ (ΣCO$_2$ = {CO$_2$} + {HCO$_3$$^-$} + {CO$_3$$^{2-}$}, where {$i$} denotes concentration of the $i$th species in molal units)[23]. Gypsum, proposed as a late diagenetic product, is thought to have been ubiquitously present throughout the entire Yellowknife Bay sediments instead of calcium carbonates[12,14,16]. We believe that gypsum was the solubility-controlled phase of Ca at the time of the late diagenesis (see Methods). This means that

gypsum is more stable than calcium carbonates in the water-sediment system. The stability condition between gypsum and calcium carbonates can be calculated as functions of pH and dissolved $\Sigma CO_2$ levels (Fig. 3).

Figure 3 shows that, unless dissolved $\Sigma CO_2$ levels are extremely low, an alkaline pH can be ruled out for the pore water in the last wetting event (Fig. 1d). The estimated dissolved $\Sigma CO_2$ in the pore water would be a mixture of both a supply of $CO_2$ from the primary components (e.g., dissolution of carbonate) and $CO_2$ dissolved in the post-depositional fluids. Thus, an assumption of no $CO_2$ supply from the primary components (e.g., absence of carbonate)[34] provides an upper limit of the pore water's pH. Given the present level of partial pressure of atmospheric $CO_2$ ($P_{CO_2}$) on Mars as a conservative lower limit of ancient $P_{CO_2}$ levels[36] and dissolution equilibrium between the atmosphere and post-depositional fluids, the pore water's pH would have an upper limit of ~7.3.

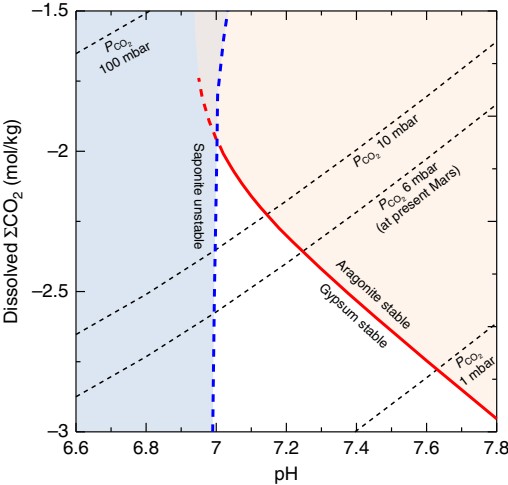

**Fig. 3** Constrained pH-$\Sigma CO_2$ conditions (white area) in the pore water for mean water composition of the full-interaction scenario (Cumberland). The red line shows the stability boundary between carbonate and gypsum. The orange and sky-blue areas correspond to aragonite stable conditions and saponite unstable conditions, respectively. The broken lines show $\Sigma CO_2$ concentrations for $P_{CO_2}$ assuming dissolution equilibrium. See Methods for calculation methodology

Another constraint on the pore-water pH relates to smectite. The Yellowknife Bay sediments contain ~20 wt% smectite[12]. The detailed mineralogical investigations suggest that the smectite is composed mainly of ferrian ($Fe^{3+}$) saponite[13,37]. Since $Fe^{3+}$ saponite generally forms through oxidative alterations of ferrous ($Fe^{2+}$) saponite, formation and deposition of $Fe^{3+}$ saponite through oxidative weathering of basaltic rocks are unlikely. $Fe^{3+}$ saponite needs to have been originally $Fe^{2+}$ saponite in the sediments, whichever it is of diagenesis or detrital origin. In this case, a concentration of $Fe^{2+}$ in the pore water in the last wetting event is likely to have been controlled by $Fe^{2+}$ saponite because it would be the predominant $Fe^{2+}$-bearing secondary mineral in the sediments, and because low water-rock ratios are expected in the water-sediment system. $Fe^{2+}$ concentrations in the pore water must be low to keep the observed interlayer spacing and compositions (see Methods). Since more $Fe^{2+}$ are provided from $Fe^{2+}$-bearing saponite at lower pH, the observed interlayer composition provides a lower limit of the pore water's pH at ~6.9. Together with the constraint from a carbonate-gypsum stability line, we propose a circumneutral pH for the pore water in the last wetting event (Fig. 1d).

The proposed neutral pH agrees with the iron-bearing secondary mineral assemblages found by Curiosity. The Yellowknife Bay sediments possess a wide variety of secondary iron-bearing minerals (akaganeite, magnetite, and $Fe^{2+}$ saponite)[12]. Although magnetite and saponite might be of detrital origins, their relatively-high abundance suggests that the pore-water pH would be compatible with their thermochemical stability. Our quantitative reconstruction of water chemistry in the pore water enables us to construct the stability relationship of these iron minerals in regard to pH and Eh (Fig. 4). The fact that all of the observed secondary iron-bearing minerals appear in the pH range of 6.8–8.0 (Fig. 4a) strongly supports our conclusion of circumneutral pH.

In contrast to a narrow range of pH constraints, the Eh covering these secondary minerals ranges widely from upper to lower limits for the presence of liquid water at circumneutral pH (Fig. 4a). To form akaganeite in particular, high-Eh oxidants should have existed in the water-sediment system (Eh > 0.3 V at pH 7: Fig. 4a). On Mars, potential oxidants capable of producing sufficient Eh include molecular oxygen, ozone, perchloric acid, and nitric acid. Given that all of these potential candidates are proposed to form via atmospheric processes[38,39], we theorize that oxidants were transported from the surface, plausibly together with protons (i.e., $H_2SO_4$), so that the fluids can contain the

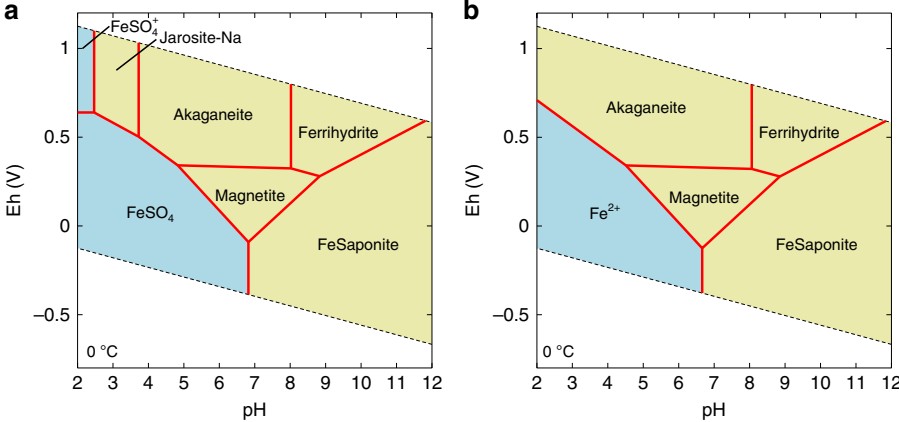

**Fig. 4** Eh-pH diagram of relevant iron species for the pore-water composition of the full-interaction scenario (**a**) and the no-interaction scenario (**b**) at 0 °C and 1 atm assuming equilibrium with albite and amorphous silica. The mean water composition at Cumberland are used for the full-interaction scenario, whereas the upper limits of the concentrations are used for the no-interaction scenario

photochemical products. Photochemically produced oxidants and volcanic acid fog[40] may have become trapped in surface ice/frost at Gale and subsequently melted. Fluids with oxidants would have been transported to the Yellowknife Bay sediments through cracks and then diffused into surrounding pores[1] (Fig. 1d). Persistence of the disequilibria in the mineral assemblages strongly suggests that redox reactions in the pore water were very slow because of low temperatures[11] and proceeded only partially due to the relatively short existence time of fluids until freezing[41], limited amounts of trapped oxidants, and/or high activation energy of $Fe^{2+}$ oxidation[42]. Although acidic-oxidizing alterations would have been widespread on Hesperian Mars[43], their durations and temperature conditions are poorly constrained. Our results of persistence of redox disequilibria suggest that acidic-oxidizing episodes of alteration would have occurred at low temperatures and were relatively short-lived at Gale.

**The no-interaction scenario**. We next consider the no-interaction scenario. In the no-interaction scenario, we consider that $SO_4^{2-}$-rich fluids did not chemically interact with the matrix of the sediments (see above). In this case, the composition of pore water that finally interacted with the matrix reflects that in the early post-depositional stages immediately before the disappearance of liquid water (Fig. 1b). In the no-interaction scenario, we did not use dissolution equilibrium of calcium sulfate but used the presence of akaganeite in the matrix in addition to the exchangeable cations in the interlayer of the smectite (see Methods). In the no-interaction scenario, akaganeite needs to be formed before intrusion of $SO_4^{2-}$-rich fluids given their occurrence in the matrix of the sediments[11]. The lower limit of pH are estimated from both the stability of $Fe^{2+}$ saponite; the same as in the full-interaction scenario. The upper limit of pH is estimated from the conditions that allow the formation of akaganeite in the matrix of the sediments (see Methods).

Table 1 and Supplementary Fig. 2 shows the water chemistry for the no-interaction scenario. The concentrations of Na and Cl are broadly close to those for the full-interaction scenario. Since the estimated Na concentrations are also significantly lower than the solubility of halite, the liquid pore water most likely disappeared via freezing and subsequent sublimation (see above). The lowest salinity of the pore water in the estimated range is 2300 mg/kg, which can still be classified as hyposaline. Our results of Na–Cl-type hyposaline water for the early Gale lakes in both the full- and no-interaction scenarios suggest that this conclusion is robust.

The pH is estimated to be 6.8–8.3. The obtained pH range is consistent with that of the full-interaction scenario. The stability fields of akaganeite and $Fe^{2+}$ saponite in regard to pH and Eh further provide constraints on the redox conditions of the pore water for the no-interaction scenario (Fig. 4b). Figure 4b shows that akaganeite is stable under low-to-neutral pH (<8) and highly oxidizing (Eh > 0.3 V) conditions (Fig. 4b). The pH-Eh conditions are consistent with the experimental investigations of the formation conditions of akaganeite[38]. The required high Eh (>0.3 V) infers an abundance of strong oxidants in the lake water in the no-interaction scenario. On the other hand, reducing $Fe^{2+}$ saponite are also suggested to exist in the matrix of the Yellowknife Bay sediments[12]. In the no-interaction scenario, the observed redox disequilibria (i.e., co-existence of akaganeite, magnetite, and $Fe^{2+}$ saponite) need to be formed other than intrusion of $SO_4^{2-}$-rich fluids.

One possibility to generate the redox disequilibria in the no-interaction scenario is interactions of upwelling groundwater at Gale[19] with initially oxidized sediments[10] after deposition. Groundwater within Martian crust would become reducing and

alkaline[10]. If reducing groundwater upwelled into oxidized sediments at Gale, $Fe^{2+}$ saponite would have formed through reduction of oxidized smectite, e.g., nontronite, by groundwater. Akaganeite in the sediments could have been also reduced into magnetite by groundwater; however, this conversion might have proceeded only incompletely due to kinetics. The occurrence of the redox disequilibria due to groundwater upwelling would be also applicable to the full-interaction scenario.

**The full-interaction scenario vs the no-interaction scenario**. Although we cannot conclude whether post-depositional $SO_4^{2-}$-rich fluids interacted with the matrix of the Yellowknife Bay sediments, we prefer the full-interaction scenario over the no-interaction scenario for the following reasons. First, the results of detailed analyses of Curiosity's data show that even the Yellowknife Bay mudstones without visual calcium sulfate veins may contain small amounts of $CaSO_4$ ($1.4 \pm 1.4$ wt.% of $SO_3$)[44]. Although this calcium sulfate in the matrix could have been of diagenetic origin, this suggests that $SO_4^{2-}$-rich fluids would have diffused into the matrix of the mudstone in the last wetting event. Additionally, post-depositional fluids could seep into the matrix of the mudstones if the rewetting event persisted for $>1$–$10^2$ years (see above). The rewetting with this timescale could have been achieved by hydrological activity that formed small gullies and deltas at Gale after the formation of Aeolis Mons[45]. Finally, because of the similar elemental compositions between the altered and unaltered mudstones, the early diagenesis could have occurred under isochemical conditions[11]. Under the isochemical conditions, the simultaneous formation of the authigenic minerals with different stability fields is unlikely.

Below, we discuss the implications of the estimated water chemistry for climate and redox interactions on early Mars, as well as a possible trigger for the last wetting event. Early climate and redox interactions are discussed based on the estimated salinity of the early lakes. Given the similar values of salinity, the former discussion is applicable to both the full- and no-interaction scenarios. In contrast, we can only constrain the water chemistry of post-depositional fluids for the full-interaction scenario; accordingly, the latter discussion is only for the case in which the fluids fully interacted with the mudstone.

**Implications for early Martian climate and redox states**. Water chemistry revealed by the present study provides constraints on the palaeoclimate and palaeohydrology at the time when early major lake systems were active. Although the Gale impact occurred at around the Noachian/Hesperian boundary[14], the timing and duration of the early lakes are poorly constrained[15]. The estimated Na concentrations of the early lakes allow us to estimate a total duration of hydrological activity at Gale until the time of the Yellowknife Bay sediments' deposition. For both of the full- and no-interaction scenarios, the Na concentration of the early lakes is estimated to be $\sim$0.03–0.2 mol/kg (Table 1) from Cumberland site. Combined with the proposed lake volume[14], a total amount of Na in the lake is obtained when the Yellowknife Bay sediments were deposited (see Methods). A total duration of hydrological activity can be estimated by division of the total amount of Na in the lake by a Na influx into the lake. Thus, short-term intermittency of lakes without removal of Na as halite burial does not reset the total duration. Assuming a steady-state lake level, the Na influx is equivalent to the product of Na concentrations in groundwater and an evaporation (or sublimation if lake surface was frozen) rate of lake water (see Methods). Using Na concentrations in groundwater in terrestrial basalts and evaporation rates derived from three-dimensional general circulation model (GCM) results, a total duration of hydrological activity

until the deposition of the Yellowknife Bay sediments would be $10^5$–$10^6$ years for the mean surface temperature near the $H_2O$ freezing point (see Methods). Higher temperatures (~25 °C) result in shorter durations (~$10^4$ years) due to efficient evaporation (see Methods), although higher-temperature conditions are harder to achieve climatically[46,47]. This duration is broadly consistent with the duration of the lacustrine environment ($10^4$–$10^7$ years) estimated from the thickness of the Gale sediments and typical sediment accumulation rates on Earth[15]. Our results also show that the early Gale lakes would have been hyposaline and closed, where evaporation dominated fluid input. Together with the occurrence of similar hyposaline lakes in terrestrial steppe area[28] and results of hydrological modeling to make a closed-basin lake at Gale[19], we also suggest that the early Gale lakes developed under semiarid climatic conditions, at least, during the deposition of the Yellowknife Bay sediments, which is consistent with the low values of the chemical index of alteration for these sediments[10,11]. Our results of water chemistry suggest that Gale would have experienced prolonged ($10^4$–$10^6$ years) episodes of warm and semiarid climates in total, although these episodes might have been intermittent. Similar to soils on present-day Mars, Na-bearing salts would have contained in soils within Gale Crater before the appearance of the lakes[2]. In warm periods, surface runoff could have transported them to the lake[2], providing salinity efficiently to the lake water. With this regard, our estimate of lake duration would be an upper limit. Assuming ~1 wt% of the salt content based on present-day soils[48], such an accumulation of surface salts through surface runoff could explain ~50% of the estimated salinity of the lakes in maximum (see Methods). Thus, the required lake duration could be shortened if saline surface runoff occurred.

The estimated total duration of the early Gale lakes agrees with some formation timescales proposed for valley networks[49] and Al-rich surface clays[50] that formed in the late Noachian to the early Hesperian, although the timing of formation of valley networks and Al-rich surface clays may not be coincident with the existence of the early lakes. Although an ancient Martian ocean might have also existed in the early Hesperian[51], three-dimensional GCM results show that its presence is not required to explain fluvial activity at Gale Crater[52]. The GCM results[52] also suggest that when the total Martian surface water inventory is high, climate conditions at the Aeolis quadrangle are humid rather than semiarid to arid (e.g., see Fig. 9 of ref. [52]), although locally semiarid climates at Gale can be achieved for some orbital and atmospheric conditions. These suggest that the early Gale lakes, valley networks, and ancient shorelines might not have formed at the same time in a single climate, but they would have formed gradually through dynamic climates of prolonged warming on a cold early Mars[46,50].

Given the difficulty of sufficient warming on early Mars solely by $CO_2$-$H_2O$ greenhouse gases, the total duration for warm episodes may reflect residence times of additional greenhouse gases[46,47] (e.g., $H_2$ and $CH_4$) in the atmosphere, repeated periods of high solar insolation at snowy low-latitude highlands upon obliquity cycles[52,53], or combinations of both[53]. Although outgassing $H_2$/$CH_4$ flux is uncertain, once released, $CO_2$-$H_2$-$CH_4$ greenhouse gases would persist longer than $10^5$ years against the diffusion-limited escape and photodissociation[46]. In a manner similar to acidic-oxidizing alterations at Gale in later stages for the full-interaction scenario, we can infer that oxidants, e.g., perchlorate, chlorate, ozone, and nitrate, might have also accumulated in surface ice/frost or upstream soils during a cold period in the early Hesperian prior to warming, which in turn could have been released into aqueous environments upon warming, e.g., triggered by $H_2$/$CH_4$ outgassing[46]. The released

oxidants might have been consumed via reactions with abundant reductants from rock components during prolonged warm episodes in the early Hesperian; whereas redox disequilibria would have been preserved due to short-term warming in the late Hesperian or later (see below). We propose that dynamic climates on early-Hesperian Mars[46,50] might have been a thermochemical drive to promote redox interactions in aqueous environments, which could have provided free energy to drive the onset of chemical evolution on this planet[54].

**Timing and trigger for acidic-oxidizing alterations**. We then turn to a climatic trigger for the last warming event at Gale that caused post-depositional, acidic-oxidizing alterations after sedimentation of the Yellowknife Bay mudstone in the full-interaction scenario. Our results of low levels of dissolved $CO_2$ in the pore water (Table 1) require both low $CO_2$ supply from the primary components and low dissolved $CO_2$ in the post-depositional fluids (see above). The former suggests the absence of carbonate in the matrix of the sediments[34]. Given that post-depositional fluids would have originated from the surface in the full-interaction scenario, the latter may reflect low $P_{CO_2}$ at the time of the last wetting event at Gale. Alternatively, dissolution equilibrium between post-depositional $SO_4^{2-}$-rich fluids with atmospheric $CO_2$ was not achieved (e.g., due to rapid melting of ice). Drawdown of $CO_2$ from fluids during transportation to Yellowknife Bay via carbonate formation is unlikely given the acidity of fluids. Based on a lower limit of pH, we propose an upper limit of $P_{CO_2}$ of ~100 mbar if atmospheric $CO_2$ was in equilibrium with the fluids (Table 1). Although formation of Fe carbonate can be kinetically inhibited even at high $CO_2$ levels[55], a previous bottom-up approach to estimate dissolved $CO_2$ provided an upper limit of $P_{CO_2}$ levels as tens of mbar[34]. The consistency between these independent approaches shows robustness of low $CO_2$ in the pore water. We consider that the estimated $P_{CO_2}$ would reflect a recent atmosphere at the time of the last wetting event at Gale. The estimated $P_{CO_2}$ suggests that the last wetting event may have occurred in the late Hesperian or even the Amazonian, when atmospheric pressure would have already declined closer to the present-day level[36]. The estimated timing of this last wetting event is consistent with the youthful age of jarosite in the Mojave2 sample ($2.12 \pm 0.36$ Ga)[17], drilled from a lacustrine mudstone ~60 m higher in the stratigraphic section than the Yellowknife Bay sediments. The last wetting event might be caused by a relatively short warming pulse from the obliquity variations[56] would last 1–$10^3$ years, rather than episodic warming due to the release of $CH_4$ by clathrate destabilization[46] into a dense $CO_2$ atmosphere.

## Methods

**Mineralogical constraints on water chemistry**. We estimate water chemistry of pore water based on the following constraints from the samples at the Cumberland and John Klein drilling sites: (a) the basal spacing of smectite, (b) the presence of saponite, (c) the mineral composition of sulfate, and d) the presence of akaganeite.

The interlayer cationic compositions of smectites were estimated from the 001 reflections. The peak intensities of the 001 reflections are related to the abundance of the cationic components in the interlayer, the atomic structure factors of each components, and the Lorentz-polarization (LP) factors[57]. The peak deconvolutions of the 001 reflections of John Klein and Cumberland sites after LP factor and the background corrections were conducted using the Pseudo-Voigt function to estimate the cation compositions (Supplementary Fig. 4). The combinations of the three peaks at 13.7, 11.5–11.7, and 10.0–10.1 Å can reasonably reproduce the 001 reflections for both smectites.

The basal spacing of smectite depends on a cation composition in the interlayer and RH. The RH remains <1% inside CheMin under temperatures 5–25 °C (ref. [13]). Supplementary Fig. 5 summarizes the basal spacings of several smectites with different cations ($Na^+$, $K^+$, $Ca^{2+}$, $Mg^{2+}$, $Fe^{2+}$ and $Fe^{3+}$) at RH = 0%. The basal spacings of $Na^+$ and $K^+$ smectites are ~10 Å; whereas, those of $Ca^{2+}$ smectite are

scattered, but most of the smectites exhibit peaks at ~12 Å. The basal spacing of $Fe^{2+}$ and $Fe^{3+}$ smectite at RH = 0% (ranging 11–12.5 Å) are similar to that of $Ca^{2+}$ smectite. Those of Mg smectite vary from 11.5 to 13.9 Å. These values are generally higher than those of Ca smectite because the hydration states of $Mg^{2+}$ in smectite are kept even at low relative humidity[58]. According to the relationship, the peak with 10.0–10.1 Å from the two smectites represent the $Na^+$ and/or $K^+$. That with 12 Å represent the $Ca^{2+}$ and Fe ($Fe^{2+}$ and/or $Fe^{3+}$). That with 13.7 Å most likely represents the $Mg^{2+}$. Partially intercalated hydroxylated Mg (i.e., $MgOH^+$) in the interlayer site of the smectite at Cumberland has been proposed as an explanation for the expanded structure[12,13]. The dissolved $MgOH^+$ species becomes important at extremely alkaline conditions (pH > 12: pKa = 12.8 at 0 °C) at ambient to low temperatures. However, concentration of $\Sigma Mg^{2+}$ (=$Mg^{2+}$ + $MgOH^+$) also becomes extremely low at high pH (pH > 10) because of the formation of Mg-bearing secondary minerals (saponite and brucite). Therefore, given the low {$MgOH^+$} at alkaline pH, the exchange of the dissolved $MgOH^+$ species into the interlayer occurs very inefficiently at ambient to low temperatures unless the $Mg^{2+}/(Na^+ + K^+)$ ratio in the pore water is extremely high. Assuming the peak from 10 Å represents $Na^+$ and/or $K^+$, the extremely high $Mg^{2+}/(Na^+ + K^+)$ ratio contradicts with $Na^+/K^+$ interlayers. Although the reported basal spacings of Mg-saturated saponite vary from 11.5 to 12.0 Å, we believe that $Mg^{2+}$-saturated Mg smectite is responsible for the spacing of ~14 Å of the smectites. This is the case because the swelling behaviors of interlayer cations mainly depend on the layer charge and charge location[59]. Vermiculite, high-layer-charge montmorillonite, and well-crystalline synthesized smectite exhibit higher basal spacings of ~14 Å (Supplementary Fig. 5). These specimens are expected to possess high layer charges. The smectite found in the Yellowknife Bay sediments is considered to be $Fe^{3+}$ saponite, which would have been formed by the alteration of $Fe^{2+}$ saponite[37]. Since the oxidation of $Fe^{2+}$ to $Fe^{3+}$ should have resulted in an increase in the layer charge by substation of $O^{2-}$ for $OH^-$[37], the smectite in the Yellowknife Bay sediments would have larger spacings, e.g., ~14 Å.

Mass balance calculations using bulk-rock chemical data from alpha particle X-ray spectrometer (APXS) and the chemistry of crystalline components indicate that the $Na^+/K^+$ ratios of the clay mineral and amorphous material were 4 and 5 for John Klein and Cumberland, respectively[60]. Thus, the observed basal spacings of the smectites suggest their cation compositions in cationic electrical charge of $Na^+$: $K^+$ to be ~4.5:1. Based on the peak deconvolutions of 001 reflections (Supplementary Fig. 4) and the $Na^+:K^+$, the cationic composition of smectites are determined. The concentration of dissolved ferric iron is extremely low except for the extremely low or high pH conditions. The direct intercalation of $Fe^{3+}$ into the interlayer of smectite from the solution was not likely. Therefore, Fe in the interlayer of smectite is most likely originated from $Fe^{2+}$, although the part of or all $Fe^{2+}$ possibly oxidized to $Fe^{3+}$ in the interlayer. The equivalent ratios of the cationic composition from John Klein and Cumberland site are given in Supplementary Table 1.

Under terrestrial conditions, saponite depositions are observed in saline lakes, low-temperature upwelling solution from seafloor, and modern oceans (Supplementary Table 2). The pH of the solutions is commonly neutral to alkaline. The predominate cations in these natural waters are always $Na^+$ rather than $Mg^{2+}$. Under such alkaline conditions, concentrations of divalent cations, such as $Mg^{2+}$ and $Ca^{2+}$, must be very low due to formation of Mg/Ca-bearing secondary minerals[23]. Consequently, the low concentrations of divalent cations lead to the dominance of $Na^+$ in the solutions. Thus, the presence of saponite in the Yellowknife Bay sediments strongly suggests that the predominant cation in the pore water is most likely $Na^+$.

CheMin detected anhydrite (anhydrous $CaSO_4$) and bassanite ($CaSO_4 \cdot 0.5H_2O$) from both of the drill core samples[12,61]. Both of the minerals rarely form under terrestrial surface conditions. Bassanite is considered to be a likely product of gypsum ($CaSO_4 \cdot 2H_2O$) dehydration[61]. We consider that bassanite and anhydrite were alteration products after the disappearance of pore water and assume that $Ca^{2+}$ and $SO_4^{2-}$ concentrations of the pore water were controlled by dissolution and precipitation equilibrium of gypsum[62].

The Yellowknife Bay sediments contain akaganeite, a chloride-bearing ferric oxyhydroxide. Akaganeite is associated with smectite in the matrix of the mudstone[11]. The formation of akaganeite requires >0.05 mol/kg of dissolved $Cl^-$ concentrations in the solution[33]. The presence of akaganeite in the sediments suggests that the dissolved $Cl^-$ concentrations in the solution must be higher than 0.05 mol/kg. Given the fact that akaganeite can accommodate the anions other than chloride, the presence of $Cl^-$ type akaganeite in Yellowknife Bay strongly suggests that the akaganeite was in contact with $Cl^-$ dominant solutions compared with other anions, such as sulfate[63]. By combining the dominant cation type of the water constrained from the presence of saponite, the original pore water in the sediments would be Na–Cl water type.

**Thermodynamic calculations.** To estimate water chemistry based on the above constraints, the modeling was performed using REACT from the Geochemist's Workbench (GWB)[64]. We assume a temperature of 0 °C and a pressure of 1 bar[13,34]. Thermodynamic data for aqueous species and mineral solubility, except for the akaganeite, ferrihydrite and $Fe^{2+}$-saponite, were from the "thermo.dat" in the GWB. The detailed descriptions for the database construction of these minerals are given in Supplementary Note 1. The activity coefficient was calculated using the

Helgeson, Kirkham and Flower version of the extended Debye–Hückel equation[65] for NaCl media at 0 °C. In the modeling, we used the molal (mol/kg $H_2O$) unit notified in {$i$}, in $i$th chemical species in solution. The solution compositions are given by the component concentrations, which are the total concentrations of the species containing each component. In the following, we used the notation of {$X$} to express the molal concentrations in a solution of component $X$.

**Constraining major components.** Under terrestrial conditions, the major cations of natural waters are $Na^+$, $K^+$, $Mg^{2+}$ and $Ca^{2+}$, while major anions are $Cl^-$, $SO_4^{2-}$, and $HCO_3^-$(ref. [23]). The concentration of $Fe^{2+}$ becomes important only under reducing conditions[23]. The concentration of $H^+$ and $OH^-$ are important only at extremely low or high pH. We assume that the $H^+$, $OH^-$, and $HCO_3^-$ are negligible in the charge balance calculations compared with other major species. The validity of the assumption will be quantitatively examined (Table 1). Perchlorate ($ClO_4^-$) has been detected in the Yellowknife Bay sediments[35]. Both $Cl^-$ and $ClO_4^-$ are monovalent anions, which do not take part in the cation exchange reactions. According to the present approach to constrain major components, the chemical behavior of $ClO_4^-$ can be treated as same as that of $Cl^-$. This means that the concentrations of $Cl^-$ obtained from the present study actually represent the concentrations of total chloride ({$Cl^-$} + {$ClO_4^-$}) and that the results of cation compositions, pH, and salinity of the pore water do not change even when the pore water contained $ClO_4^-$. However, the presence of the $Cl^-$ bearing akaganeite in the Yellowknife Bay sediments suggests the concentration of $Cl^-$ must be higher than that of $ClO_4^-$(ref. [63]).

The cation compositions in the interlayer of smectite were calculated using the cation exchange equilibria as follows:

$$>X : Na^+ + K^+ = >X : K^+ + Na^+,$$

$$K_{Na\_K} = \frac{\beta_{>X:K^+} a_{Na^+}}{\beta_{>X:Na^+} a_{K^+}}, \tag{1}$$

$$2>X : Na^+ + Di^{2+} = >X_2 : Di^{2+} + 2Na^+,$$

$$K_{Na\_Di} = \frac{\beta_{>X_2:Di^{2+}} a_{Na^+}^2}{\beta_{>X:Na^+}^2 a_{Di^{2+}}}, \tag{2}$$

In those equations, Di represents the divalent metal (Ca, Mg and Fe), >X denotes the exchangeable site. $a_i$ denotes activity of $i$th species. $\beta_j$ stands for the activity of $j$th species complexed with an exchange site. Since surface activity coefficients are not well known, we assume that they are unity. Based on the Gaines–Thomas convention[7], $\beta_j$ is given in terms of the fraction $j$th species in an exchangeable site of the total electrical equivalents of exchange capacity occupied by cations;

$$\sum_j \beta_j = 1 \tag{3}$$

The interlayer compositions of smectites observed from the sediments (Supplementary Table 1) provide the constrains for $\beta_j$. $K_{Na\_i}$ represents the selectivity coefficients of the exchange of $i$th species to $Na^+$ in the interlayer. $K_{Na\_i}$ is highly dependent on the exchanger composition[22]. We estimated the $K_{Na\_i}$ for smectite as a function of exchanger compositions (Supplementary Table 1) by using the generic empirical model[22]. The uncertainties of the estimated log $K_{Na\_i}$ are ±0.2[22]. The temperature dependences of selectivity coefficients were reported to be small within these uncertainties[6].

The charge balance of the solution can be expressed as:

$$\{Na^+\} + \{K^+\} + 2\{Mg^{2+}\} + 2\{Ca^{2+}\} + 2\{Fe^{2+}\} = \{Cl^-\} + 2\{SO_4^{2-}\}. \tag{4}$$

Although we consider aqueous complexes in the calculation, the charge balance expressions in the manuscript ignore the aqueous complexes for illustrative purpose. {$Ca^{2+}$} and {$SO_4^{2-}$} are constrained from dissolution equilibrium of gypsum (see above) for full-interaction scenario:

$$CaSO_4 \cdot 2H_2O = Ca^{2+} + SO_4^{2-} + 2H_2O. \tag{5}$$

The corresponding mass action expression is expressed as follows:

$$K_5 = a_{Ca^{2+}} a_{SO_4^{2-}} = \gamma_{Ca^{2+}} \gamma_{SO_4^{2-}} \{Ca^{2+}\}\{SO_4^{2-}\}. \tag{6}$$

where, $\gamma_i$ represents the activity coefficients of $i$th species. Based on the constrain from the water type of Na–Cl, the equivalent concentrations of $Na^+$ and $Cl^-$ are higher than the major species such as $Mg^{2+}$, $Ca^{2+}$ and $SO_4^{2-}$. Therefore, the endmember compositions of solutions at an interlayer cationic composition under a single set of selectivity coefficients can be determined from following two boundary conditions:

$$\{Na^+\} = 2\{Di^{2+}\}, \tag{7}$$

$$\{Cl^-\} = 2\{SO_4^{2-}\}, \tag{8}$$

where {$Di^{2+}$} represents the concentrations of divalent cations, of which concentrations are highest next to that of $Na^+$. We calculated the two endmember solution compositions from Eqs. (7) and (8) under the ranges of the selectivity coefficients with the uncertainties.

In the no-interaction scenario, the matrix of the mudstone is assumed to be isolated from post-depositional fluids enriched in sulfate. Therefore, the charge balance of the solution can be expressed as:

$$\{Na^+\} + \{K^+\} + 2\{Mg^{2+}\} + 2\{Ca^{2+}\} + 2\{Fe^{2+}\} = \{Cl^-\}. \quad (9)$$

The constraint on $\{Cl^-\}$ with regard to the existence of akaganeite in the matrix of the mudstone is applied to the no-interaction scenario (i.e., $\{Cl^-\} \geq 0.05$ mol/kg)[33] instead of the constrain from Eq. (8).

**Constraining pH and dissolved $\Sigma CO_2$ in the full-interaction scenario.** The formation of calcite, most stable calcium carbonate phase, is known to be kinetically inhibited in the presence of significant amount of $Mg^{2+}$ in solutions[7]. Instead, metastable aragonite is favored to form from the solutions[7]. The stability relationship between aragonite and gypsum is written as:

$$CaCO_3 + SO_4^{2-} + 2H^+ + H_2O = CaSO_4 \cdot 2H_2O + CO_2(aq). \quad (10)$$

The Gibbs free energy ($\Delta G_R$) of the reaction can be written as follows:

$$\Delta G_{R,10} = \Delta G_{R,10}^0 + RT \ln\left(\frac{a_{CaSO_4 \cdot 2H_2O} a_{CO_2(aq)}}{a_{CaCO_3} a_{SO_4^{2-}} a_{H^+}^2 a_{H_2O}}\right). \quad (11)$$

$\Delta G_R^0$ represents the standard free energies of the reactions and is related to the equilibrium constants[23]. The equilibrium constant of Eq. (10) was calculated using the RXN program in the GWB. The activity of $SO_4^{2-}$ comes from the estimated major components (Table 1). In Eq. (11), gypsum is thermochemically stable at $\Delta G_R < 0$. The stability conditions between gypsum and aragonite were calculated as functions of pH and $a_{CO_2(aq)}$. Once $a_{CO_2(aq)}$ and pH are obtained, $\{HCO_3^-\}$ and $\{CO_3^{2-}\}$ are calculated from their dissociation constants. Thus, the stability conditions between gypsum and aragonite can be expressed as functions of pH and $\Sigma CO_2$ (Fig. 3).

Saponite is trioctahedral smectite bearing Mg and Fe, which is unstable at low pH[31]. Smectite usually occurs as fine particles and the dissolution rates must be higher than the primary minerals. It can be assumed that the concentrations of Mg and Fe in the solution were equilibrium with respect to saponite. The dissolution reaction of Fe-saponite can be written as follows;

$$Na_{0.175}Mg_{0.0875}Fe_{1.5}Mg_{1.5}Al_{0.35}Si_{3.65}O_{10}OH_2 + 7.4H^+$$
$$= 0.175Na^+ + 1.5Fe^{2+} + 1.5875Mg^{2+} + 0.35Al^{3+} + 3.65SiO_2(aq) + 4.7H_2O. \quad (12)$$

$\{Al^{3+}\}$ and $\{SiO_2(aq)\}$ would be equilibrated with respect to albite ($NaAlSi_3O_8$) and amorphous silica ($SiO_2(am)$), which are found in the Yellowknife Bay sediments[34] through the following reaction;

$$Na_{0.175}Mg_{0.0875}Fe_{1.5}Mg_{1.5}Al_{0.35}Si_{3.65}O_{10}OH_2 + 6H^+ + 0.175Na^+$$
$$= 0.35NaAlSi_3O_8 + 1.5Fe^{2+} + 1.5875Mg^{2+} + 2.6SiO_2(am) + 4H_2O. \quad (13)$$

The mass action expression is given as follows:

$$K_{13} = \frac{a_{Fe^{2+}}^{1.5} a_{Mg^{2+}}^{1.5875}}{a_{H^+}^6 a_{Na^+}^{0.175}}. \quad (14)$$

The equilibrium constants of $K_{13}$ were calculated using the RXN program in the GWB. The activities of $Na^+$, $Fe^{2+}$ and $Mg^{2+}$ derive from the estimated major components (Table 1). An lower limit of pH was calculated from the higher limits of $Fe^{2+}$ and $Mg^{2+}$ of the estimated major components.

**Constraining pH in the no-interaction scenario.** Because we exclude gypsum from the assumed system, the $\Sigma CO_2$ cannot be constrained in the no-interaction scenario. The lower limit pH is estimated from the stability of smectite as described above. The upper limit of pH is estimated based on the formation conditions of akaganeite. Peretyazhko et al.[33] showed that the upper limit pH allowing akaganeite formation is pH 8, above which ferrihydrite is formed instead of akaganeite. The transformation reaction of akaganeite to ferrihydrite ($Fe(OH)_3$) can be written as follows:

$$FeO(OH)_{0.7}Cl_{0.3} + 1.3H_2O = Fe(OH)_3 + 0.3H^+ + 0.3Cl^-. \quad (15)$$

The mass action expression of Eq. (15) is given as follows:

$$K_{15} = \frac{a_{Cl^-}^{0.3} a_{H^+}^{0.3}}{a_{H_2O}^{1.3}} \quad (16)$$

The activity of $Cl^-$ is derived from the estimated major components (Table 1) assuming $Cl^-$ is predominant species relative to $ClO_4^-$. The obtained activity of $H^+$ from Eq. (16) represents the upper limit of pH, which enables the formation of akaganeite.

**Estimates of total duration for hydrological activities at Gale.** We assume a simplified hydrological system in a closed-basin lake, in which Na supplied from the surface/groundwater accumulate in the lake (see the main text and Supplementary Note 2 for a terrestrial analog). In this system, a mass balance of supplied $Na^+$ can be expressed as follows;

$$C_{ground} \times F_{ground} \times t = M_{lake} \times C_{lake} \quad (17)$$

where $C_{ground}$ and $C_{lake}$ are Na concentrations (mol/kg) in the surface/groundwater and lake, respectively, $F_{ground}$ is the flux (kg/sec) of inflowing water, $t$ is the duration (sec), and $M_{lake}$ is the mass (kg) of lake water. Although the mass of lake water at the time of deposition of Yellowknife Bay is poorly constrained, $M_{lake}$ may become $(3–6) \times 10^{15}$ kg for a Gale basin-sized lake (i.e., diameter of ~100 km with the central peak with ~45 km diameter[15], mean lake depth of a few hundred meters). We use typical Na concentrations in terrestrial surface/groundwater within terrestrial basaltic rocks ($10^{-4}–10^{-3}$ mol/kg for circumneutral-pH waters in Iceland[66], Australia[67] and North America[68]) for $C_{ground}$. Lower-temperature groundwater[67] tends to contain lower Na; yet, Na concentrations[66–68] vary only in a factor of ~5 for groundwater temperatures of 4–25 °C. Accumulation of volcanic acids during intermittency of hydrological cycles would increase Na concentration in groundwater due to high rates of chemical weathering[2]; however, low solution pH and acidity would maintain only for ~$10^2$ years owing to neutralization by interactions with rocks[1] (also see the main text). In a steady state, $F_{ground}$ is balanced with the total mass flux of evaporating $H_2O$, $F_{evap}$ (kg/sec). The potential evaporation (or sublimation if lake surface is frozen) flux, $S_{pot}$ (kg/m²/sec), per unit surface area of the lake can be expressed as follows[52]:

$$S_{pot} = \frac{C_D |v|}{R_{H_2O} T_a} p_{sat}(Ts) \quad (18)$$

where $c_D$ is the drag coefficient[52] ($= 2.75 \times 10^{-3}$), $|v|$ is the surface wind velocity, $R_{H_2O}$ is the specific gas constant for $H_2O$, $T_a$ is the atmospheric temperature near the surface, and $p_{sat}(T_s)$ is the $H_2O$ saturation vapor pressure at the surface temperature, $T_s$. We used 3–4 m/s for $|v|$ at Gale based on results of a three-dimensional GCM for a 1-bar atmosphere and obliquity of 25 degrees[52] (Supplementary Fig. 6) (c.f., $|v| = 5–6$ m/sec at Gale on present-day Mars[69]) and assumed $T_s \approx T_a$. Using the above values, $S_{pot}$ becomes $(1–4) \times 10^{-5}$ kg/m²/sec for $T_s = 260–273$ K. Thus, $F_{evap}$ (=$F_{ground}$) becomes ~$10^5$ kg/sec for a Gale-sized lake. By introducing $C_{lake} \sim 0.1$ mol/kg (Table 1) into Eq. (18), $t$ becomes ~$10^5–10^6$ years. The Gale lakes could have salinated more quickly under warmer and drier conditions due to high $S_{pot}$. For instance, $t$ is reduced to ~$10^4$ years for $T_s = 298$ K. Nevertheless, higher temperatures would have been harder to achieve on early Mars[47,52]; accordingly, the estimated duration for $T_s = 298$ K can serve as a conservative lower limit. $S_{pot}$ is the maximum evaporation rate assuming relative humidity of 0% (ref. [52]). However, given the proposed semiarid to arid climates[11,19], the assumption of a relative humidity of 0% would not affect our conclusions significantly. The estimated duration includes an order(s) of magnitude of uncertainty mainly derived from an uncertainty of the lake depth. Validity and uncertainties of our methodology can be evaluated through estimating the ages of hyposaline closed-basin lakes developed in semiarid areas on Earth (see Supplementary Note 2).

Martian soils, in general, contain abundant Na-bearing salts (e.g., chloride and perchlorate)[48]. As mentioned in the main text, surface runoff could have transported surface salts accumulated within soils efficiently to the early Gale lake in warm periods[2]. We estimate a quantitative effect of this process. Present-day soils on Mars typically contain ~1 wt% of salts[48]. Given ~$(2–3) \times 10^{10}$ m² of the catchment area of the early lake (=the surface area of Gale Crater's cavity) and ~$3 \times 10^3$ kg/m³ of bulk density of soils, the maximum mass of transported salts through surface runoff can be ~$5 \times 10^{11}$ kg assuming a 1 m-thick soil layer and ~$5 \times 10^{12}$ kg for a 10 m-thick soil layer. According to the infiltration theory of rain into dried soils[70], persistent inputs of surface water due to rain fall and/or snow melting needs to continue for ~20 h and ~2000 h to reach the soils (assuming volcanic loam) in depth of 1 and 10 m, respectively. Since this timescale would roughly correspond to those of precipitation/snow melting on early Mars[52], we consider that the wetting depth within soils upon rain fall and/or snow melting over Gale Crater would be at most ~10 m. On the other hand, given ~0.03–0.2 mol/kg of lake salinity (see the main text) and $(3–6) \times 10^{15}$ kg of lake water mass (see Eq. (17)), the total salt mass in the lakes would become ~$(1–10) \times 10^{13}$ kg. This suggests that if the soils in the catchment area of the lakes contained salts comparable to that of today's soils, saline surface runoff could provide ~50% of the total salinity of the lakes in maximum. Assuming the saline surface runoff[2], thereby, the required lake duration could be reduced in a factor of ~2 or less.

## Data availability
The authors declare that the data supporting the findings of this study are available within the article and its Supplementary Information File.

## Code availability
Code for all analyses will be made available on request.

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

## Acknowledgements

Discussions with T. Kasama, T. Sato, T. Shibuya, and S. Yokoyama are greatly appreciated. Financial support was provided by KAKENHI from the Ministry of Education, Culture, Sports, Science, and Technology (MEXT) (Grant Nos. JP17H06458, JP17H06454, and JP17H06456) and a cooperative research program of the Institute of Nature and Environmental Technology, Kanazawa University (No. 19035).

## Author contributions

K.F., H.S and K.M. modeled the water chemistry. Y.S. interpreted the estimated water chemistry. The estimate of the duration of the early Gale lakes was conceptualized by R.W. and Y.S., and calculated by Y.S. and R.W. All authors wrote the paper.

## Competing interests

The authors declare no competing interests.
