## [Peer Review File · Nature Communications]

Reviewers' comments:

Reviewer #1 (Remarks to the Author):

The manuscript is well written. It suggests that the lakes developed at Gale on Mars are part of a global hydrological cycle during warmer episodes in the early Hesperian. In my opinion this is an overstatement. Based on the top-down approach that the authors are discussing in this manuscript, and that is deduced from Curiosity observations of a few places within Gale, a local hydrological cycle can be the maximum that could be inferred (although I see in the manuscript, even that would need some more proofs). The manuscript is based in too many assumptions that would need evidences to support the conclusions in this article.

I find that the conclusions in the manuscript are based in assumptions that are taken as granted with no justification. Please find them below:

- As the authors state "Although the top-down approach cannot trace the water chemistry temporal evolution", YES, this is an important limitation of top-down approach and this is why inferring any dynamism (like "dynamic climate" as even indicated the title of the manuscript, requires extreme caution and supporting evidences. I really do not see these evidences in the manuscript.

- This sentence (lines 73-75 in the original WORD document): "Hence, the pore-water chemistry recorded in the Yellowknife Bay sediments most likely represents saturated pore water immediately before freezing." This is not sufficient enough to exclude the possibility of intermittent melting and evaporation. If the authors are talking about a persistent 10^4 – 10^6 years' "water cycle" then completely excluding the evaporations scenarios (which are more likely in the suggested arid climate) is not a valid approach. Since it is such a huge time-scale and the authors are considering climate dynamism, would it not be appropriate to consider wind erosion or wind deposition layering to play some part in regulating the ionic concentrations which we are observing today?

Since the authors are relying on terrestrial analogies for deriving such inferences they must also consider the major role that various environmental factors play in depositional events of inorganics over a period of time. A good example of this which will modify the deductions of the present research can be found in <https://www.hydrol-earth-syst-sci.net/14/801/2010/> .

- Line 89: "Despite the occurrence of post-depositional alterations". What about Aeolian deposition at millennial scales? Can't that be another possible contributor?

- Lines 102-104: "On Earth, saline lakes are abundant in semiarid to arid climate regions, such as inner-continental steppe/desert areas^{21,22}. Most of the terrestrial saline lakes in continental areas are terminal lakes without any outflowing rivers. " Yes, and such lakes are also a result of deliquescence facilitated by salt-rich regolith in the terrestrial environments. This might would have been the case in the time period that the present study is considering for the Gale lakes. However, the study here is completely neglecting this possibility.

- Lines 104-106: "Lake levels of terminal lakes are at lower elevations than potential outflowing rivers and are maintained by a balance between inflowing water and evaporation. " Such continental saline lakes in salt flats are a major source of ground water recharge and due to their often shallow nature, their boundaries are not very rigid and may vary at seasonal to yearly or decadal scales since deliquescence also plays significant role in their water balance in the arid climate. Please consider such possibilities if you intend to establish a good terrestrial analogy.

- Lines 221-223: "Our results of water chemistry suggest that Gale would have experienced prolonged (10^4 – 10^6 years) episodes of warm and semiarid climates in total, although these episodes might have been intermittent." Exactly what I mentioned earlier. Such episodes are intermittent and completely substituting evaporation with sublimation is a very weak analogy.

Reviewer #2 (Remarks to the Author):

Summary:

In the manuscript 'Dynamic climate and redox interactions on early Mars inferred from water chemistry at Gale' constraints on the chemistry of ancient pore waters of lacustrine sediments are obtained based on inferences of clay mineral interlayer exchangeable cation content, the presence of calcium sulfate, halite, and several Fe-bearing minerals reported by MSL at Yellowknife Bay, Gale Crater. The derived fluid chemistry is used to inform the hydrology and duration of ancient lakes based on a mass balance model. Estimates of pH and redox disequilibria exhibited by co-occurring

Fe-bearing phases are used to argue for a dynamic climate, with periods of acid-oxidizing alteration driven by melting of surface ice.

To my knowledge this approach for estimating ancient pore water chemistry has not been applied to martian sediments before and thus has the potential to provide new insights into the hydrological and geochemical cycles of early Mars. This is highly topical subject and suited to the broad scope of this journal.

However, some of the assumptions underlying the calculations are questionable and alternative explanations are underrepresented in the manuscript. Table 1 presents a unified, comprehensive estimate of pore water chemistry of Yellowknife Bay, but as alluded to throughout the manuscript, the aqueous history of the sediments is complex and the influence of multiple generations fluids are expected over a geologically protracted time (e.g. Martin et al., 2017 – JGR Planets 122, 2803-2818). The manuscript does not successfully reconcile the chemical constraints presented in Table 1 with the aqueous history discussed. This is because the manuscript contains multiple examples of conflicting statements, such as:

Line 62: 'Given evidence for the occurrence of local wetting events within Gale Crater after deposition of lacustrine sediments^{16,17}, the estimated compositions would represent the chemistry of the pore water provided in the last wetting event at Yellowknife Bay.' [In this case the 'wetting event' being referred are not related to the original lake waters – based on references cited they are waters from much later lacustrine systems or glacial meltwaters]

Line 83: 'Given the proposed geo-hydrological evolution of Gale Crater^{16,17}, the estimated pore-water chemistry would reflect combinations of both components from lake water trapped within the pores of the Yellowknife Bay sediments upon deposition, and components from post-depositional fluids during late diagenesis.'

Line 99: 'Our results of the pore-water chemistry suggest that the water chemistry of the early Gale lakes would be characterized as moderately high in salinity at the time of the Yellowknife Bay sediments deposition.'

As a result it is unclear what and when chemical constraints apply to the various generations of fluids influencing the sediments. Moreover the methods of calculation assume mineral equilibria between phases that were products of temporally and geologically distinct aqueous events.

Recommendation:

Overall I would encourage the authors to pursue this approach, but a heavily revised manuscript is needed that presents a coherent discussion of the aqueous history of the sediments and what chemical constraints are applicable to the fluid events. A more open discussion of potential problems with model assumptions would be appreciated.

Details:

Interlayer chemistry: Determining the composition of exchangeable cations within smectite in Yellowknife Bay samples provides a central constraint on fluid chemistry. To do this the manuscript examines the d-spacings of smectite in John Klein and Cumberland. The sensitivity of d-spacing of smectites to exchangeable cation species is well documented and these two samples have d-spacings of ~ 10 angstroms and ~ 13 angstroms. This is explained as a result of mixed Na⁺/K⁺ and Mg²⁺ cations in interlayers, because small variations in the ratio of monovalent and divalent species can produce quite different d-spacing. This is required because the two samples are within meters of each other and pore fluids shouldn't have been vastly different.

The manuscript should discuss why an alternative explanation for 13 angstrom spacing of Cumberland presented in Vaniman et al., 2014 and Bristow et al., 2015 should be ruled out. In this scenario the interlayer is kept open by partial intercalation of Mg-hydroxy interlayers.

In addition, I don't think the manuscript provides adequate proof that Mg²⁺ exchanged saponite will have a d-spacing of 13 angstroms under the zero humidity conditions inside the CheMin instrument. From the studies available, including Suquet et al. 1975 cited in the manuscript, Mg-exchanged saponites have smaller d-spacings of 11-12 angstroms. The argument that Mg exchanged smectites will have d-spacings of 13 is based on data from montmorillonite and synthetic smectites, which may not be appropriate

It is also noteworthy that all other clay minerals reported from other lacustrine mudstones in Gale have d-spacings of 10 angstroms (e.g. Rampe et al., 2017; Bristow et al., 2018) – which argues against an interlayer cation content that lies on a cusp that allows d-spacing to flip between 10 and 13 angstroms.

Assumptions in calculations: The derivation of pore water chemistries presented in Table 1 assumes that [Ca²⁺] and [SO₄²⁻] of fluids were in equilibrium with gypsum. While Ca-sulfates are components of both John Klein and Cumberland samples, they are 'attributed to small veinlets that were quantified in the Yellowknife Bay boreholes (supplement to Vaniman et al. 2014), with little or no Ca-sulfate cement in the mudstone matrix.' (Vaniman et al., 2018 – Am. Mineralogist, v. 103 page 1015.) The veinlets that record the passage of Ca and sulfate bearing fluids show that this fluid event post-dates lithification of the sediments and thus it may not be valid to assume that pore waters that were in equilibrium with exchangeable cation sites of clays were also in equilibrium with gypsum. The main problem being the inefficiency of post-lithification cation exchange of clays. In the cited paper describing the method for deriving pore water chemistry from clay rocks it is stated that: 'It may be difficult, if not impossible, to obtain water samples for chemical analysis from such rocks because of their low hydraulic conductivity' (Gaucher et al., 2009). This brings up the question of how effective and pervasive post-lithification 'wetting' and alteration events were at influencing exchangeable smectite interlayer sites?

Overprinting and evolution of pore water chemistry: The manuscript argues that 'pore-water chemistry suggest that the water chemistry of the early Gale lakes would be characterized as moderately high in salinity at the time of the Yellowknife Bay Sediment deposition. The estimated

Na-Cl concentrations are slightly lower than terrestrial seawater, but significantly higher than freshwater.’ This is the basis of mass balance calculations used to estimate lake lifetime. In contrast, other chemical characteristics of the pore fluids such as pH, Eh and inorganic carbon content are proposed as being governed by later fluid events – the passage of melt-induced acidic-oxidizing fluids is stressed toward the end of the manuscript, although ‘wetting events’ are mentioned earlier. It is unclear to me how the characteristics of multiple generations of fluids can be isolated like this with the available data and this is the heart of the problem with this manuscript.

Other comments:

Line 454: ‘Sulfate: XRD detected anhydrite (anhydrous CaSO_4) and bassanite ($\text{CaSO}_4 \cdot 0.5\text{H}_2\text{O}$) from both of the drill core samples 11,64. Both of the minerals rarely form under terrestrial surface conditions.’

The origin of these minerals is also discussed in Vaniman et al., 2018 – *American Mineralogist* 103 (7): 1011-1020. Bassanite is a likely product of gypsum dehydration within the low humidity conditions inside the CheMin instrument.

Line 110: ‘Additionally, formation of trioctahedral smectite, including saponite, usually occurs in terrestrial alkaline-saline lakes as an alteration product by early diagenesis. Thus, the abundance of saponite in the Yellowknife Bay sediments is also supportive of saline conditions for Gale lake water.’

Saponite does tend to occur in alkaline-saline lakes on Earth, but in Gale saponite is thought to be the product of largely closed system alteration of reactive mafic minerals like olivine rather than via concentration of ions through evaporation – so may not be a good indicator of salinity.

Line 65 ‘There are two possible ways for the liquid pore water to have disappeared from the sediments after the final wetting event.....evaporation and freezing and sublimation.’

How about the pore water drains away as the water table drops?

Line 221 ‘Our results of water chemistry suggest that Gale would have experienced prolonged (104–106 years) episodes of warm and semiarid climates in total, although these episodes might have been intermittent.’

The manuscript should discuss the constraints on duration of lakes provided by extent and thickness of lacustrine deposits (e.g. Grotzinger et al., 2015 – *Science*).

Reviewer #3 (Remarks to the Author):

Review of: Dynamic climate and redox interactions on early Mars inferred from water chemistry at Gale, by: Keisuke Fukushi, Yasuhito Sekine, and Robin Wordsworth.

Synopsis: In this manuscript, the authors make use of a geochemical model to derive pore water fluid chemistry from the structure and composition of clay minerals in the Yellowknife Bay formation in Gale Crater, Mars. This model has been validated in published studies of clay minerals and co-existing pore waters on Earth, lending confidence that it can be successfully applied to Mars. The use of clay structural and chemical properties to place quantitative constraints on the chemical composition of the fluid that the clays last interacted with represents a novel and compelling application to data returned from the surface of Mars. Using this model, the authors make an estimate of the chemical composition of the lake water in Gale Crater at the time that clay minerals in the Yellowknife Bay formation were deposited. That chemical composition is used as input to a model that provides order of magnitude constraints on the duration of lake water activity in Gale crater. The modeled durations are broadly consistent with independent published estimates based on sediment deposition rates.

Overall, the modeling approaches applied here have high merit, with the potential to provide useful estimates of lake water composition and duration and Gale crater. Unfortunately, as detailed in the comments below, the authors have so thoroughly misinterpreted the detailed geological context of the rocks that their model is applied to that their estimates of lake water chemical composition are likely to be highly inaccurate. This inaccuracy has follow-on effects on their estimates of lake duration, which rely on estimated lake water chemical compositions. This misinterpretation can be readily corrected through more careful reading of the available literature on the sedimentology of the Bradbury Group. Correcting these misinterpretations will result in significant reworking of the modeling and interpretation; accordingly, I recommend that the manuscript be rejected in its current form but would encourage the authors to resubmit this manuscript elsewhere in an updated form, after taking the issues below into consideration.

Major issues with regard to geological context:

1. I will begin with a walkthrough of the available constraints on the depositional and diagenetic/burial history of the Sheepbed member, with appropriate references, and then try to help explain why getting this history correct is so important to the application of the techniques used in this paper.

- a. The Sheepbed member was deposited by flow deceleration as a river(s) encountered a lake in Gale crater. Massive (i.e., poorly-laminated) textures are consistent with rapid rainout of suspended silt- to clay-sized particles in a near-shore (proximal) environment [1,2,3].
- b. No textures characteristic of saline lakes (e.g., dessication cracks, rip-up chips), glacial lakes (e.g., dropstones, tillites), or subglacial lakes (e.g., sand mounds, strongly bi-modal sand and mud grain size distributions) were observed in images of the Sheepbed member [1,2,4]. Observed textures and stratigraphic relationships appear consistent with deposition in an open body of dilute water.
- c. The bulk chemical composition of the Sheepbed member mudstones indicate that the sources of the silt- and clay-sized particles experienced a low degree of chemical weathering, consistent with the hypothesis that the catchment region for rivers flowing into the lake in Gale crater experienced cold climate conditions at the time of Sheepbed member deposition [3,5].
- d. The observation that the bulk chemical composition of the rock approximates the composition of unaltered basalt, while the bulk mineralogy contains ~25-55% secondary (non-detrital) phases, has been used to argue that most of the chemical reactions that resulted in the generation of phyllosilicates and other secondary minerals and mineraloids took place after deposition, during early diagenesis [3,5,6,7].
- e. During early diagenesis a variety of textural features were produced, including: subaqueous cracks filled with isopachous cements, interpreted as syneresis cracks [1,8], and dark concretions and voids with cemented rims [1,9]. These features are interpreted as the products of mineralization during early diagenesis, which potentially resulted in gas production in the sediment [1,3,7,8,9,10].
- f. During burial, likely after the generation of features described in e. (above), the Sheepbed member sediment was subjected to enough burial pressure to produce a sedimentary dike composed of fluidized Sheepbed member sediment, preserved as a feature informally named “the snake” [1].
- g. During later diagenesis (i.e., after the events described in a-f, and after the sediment had been lithified), the Sheepbed member mudstones were pervasively fractured and those fractures were filled with Ca-sulfate. These Ca-sulfate filled fractures cross cut the mudstone and the early diagenetic features described in e. (above) [1]. Based on measurements of fracture orientation and morphology, the Ca-sulfate filled fractures in the Sheepbed mudstones are interpreted to have formed by hydrofracturing at a minimum burial depth of 1.2 km [11]. Notably, the mineralizing fluid that filled these fractures with Ca-sulfate apparently did not penetrate the host mudrock, as the Sheepbed member mudstones exhibit among the lowest total sulfur contents of any rock ever analyzed on Mars [1,5].
- h. While it is difficult to draw strong conclusions based on only two samples, imaging of the drill hole walls after the collection of the John Klein (JK) and Cumberland (CB) samples revealed that the JK sample contained abundant fracture-filling Ca-sulfate (formed by the process described in g., above) as compared to CB. Not surprisingly, the JK sample contains more Ca-sulfate by XRD than CB, and also more halite [3, 12], suggesting that the halite detected in JK and CB is associated with the

Ca-sulfate vein fills rather than the impermeable host mudrock. The presence of halite in Ca-sulfate veins is also suggested by LIBS spectra collected from them [13].

i. Finally, the bulk chemical composition of the CB sample indicates higher total Cl abundance than that of the JK sample [14], but the Cl in both samples is largely attributed to the presence of ~wt. % abundances of oxychlorine species, which are hypothesized to have been an important component of the lakewater at the time of Sheepbed member deposition [14]. The mineralogical host of these oxychlorine species, which are ubiquitous in rock samples analyzed by the SAM and CheMin instruments, remains one of the outstanding open questions from the Curiosity mission to Gale crater.

2. Taken together, the constraints described above reveal three potentially significant flaws in the geochemical modeling described by Fukushi and co-workers:

a. The calculation of idealized fluid compositions (Table 1 in manuscript) from clay interlayer spacing relies on the constraint: $2\{Na\} > \{Mg\}$ (supplemental information file lines 33-35), but this constraint assumes that halite was formed from pore fluids that were in contact with the clay minerals in the JK and CB drill samples. Based on the sedimentological constraints, it is entirely possible that halite is part of the Ca-sulfate fracture filling mineral assemblage (h., above), and it is likely that the Sheepbed member mudstones were impermeable to the mineralizing fluids that filled these fractures (g., above). Therefore, the high concentrations of dissolved Na and Cl required to form halite may have been components of a fluid that were never “in communication” with the phyllosilicate minerals in the JK and CB samples. Furthermore, given the absence of sedimentological evidence for evaporation or freezing as a significant control on rock textures during Sheepbed member deposition (b., above), it seems unlikely that the high salinities required to precipitate halite from solution were present in the lake during the deposition of the mudstones. At the very least, a more valid approach, one which honors the in-situ sedimentological and geochemical results, would be to present a range of possible solution chemistries that satisfy the constraints from phyllosilicate crystal chemistry under the assumption that halite was either present or absent during clay mineral precipitation and equilibration with its co-existing fluid.

b. Figure 2 and the supporting text have great potential to reveal important constraints on the pH and DIC concentration of fluids from which Ca-sulfate fracture filling minerals were formed. This issue is actually a matter of some debate within the MSL Curiosity science team [15-17]. Unfortunately, Fukushi and co-workers use the relationships in Figure 2 in an attempt to provide constraints on the nature of fluids in the surface environment at Gale Crater. Given the deep burial conditions under which the fractures that host Ca-sulfate were formed, and the fact that the host rocks were impermeable to these mineralizing fluids (g., above), it is highly unlikely that the fluids from which Ca-sulfate formed had anything to do with surface environmental conditions. The relationships on Figure 2 should therefore not be used to place constraints on paleo-pCO₂ conditions in the atmosphere, or the nature of fluids that equilibrated with phyllosilicates. Instead, what is potentially revealed by the relationships shown in Figure 2 is that the fluids from which Ca-sulfate precipitated were low in total DIC, and circum-neutral in pH, both of which are expected consequences of rock buffering to remove DIC and acidity derived from the atmosphere during deep burial. Cast in this new light, Fukushi and co-workers may have provided very useful constraints on

the nature of deep subsurface diagenetic fluids in Gale crater! An additional useful modeling exercise would be to evaluate the gypsum-calcite relationships depicted on Figure 2 as a function of temperature, which is expected to be higher in the deep subsurface.

c. Finally, as consequence of the previous comment (b. immediately above) the acid conditions called on to neutralize the alkaline fluids from which clay minerals may have precipitated in order to form Ca-sulfate are unnecessary (manuscript lines 169-176). Instead, the phyllosilicate and sulfate minerals can be readily interpreted in the context of two different generations of fluid: an alkaline pH fluid that formed phyllosilicates during early diagenesis, and another circum-neutral pH fluid that formed Ca-sulfate after lithification during deep burial. On this point, it is worth noting that there is not a single mineral phase in the JK or CB samples that requires low-pH conditions to form [6,12], and so the link to the “acid diagenesis” described by some authors [e.g., 18] for lithologies higher in the Gale stratigraphy is tenuous.

Other comments:

1. The way that the methods and supplemental information file are broken up to make this paper suitable for the Nature Communications format make the body text of this manuscript particularly difficult to read. The arguments in the body text require the reader to read the methods and supplemental before even beginning to read the body text. I think that this submission would be better suited to a longer format journal where the supplementary information can be folded into a longer methods section and incorporated into the article, making for a much more logical flow of ideas.
2. Sentence starting on line 70 of the manuscript should have a reference.
3. Line 75-76 of the manuscript: where does this constraint on freezing point depression come from?
4. Line 97 of the manuscript: what are the Na⁺ concentrations in calcium sulfate veins lower than? This sentence needs to be rephrased for clarity. Also, see reference [13, below].
5. Lines 6-8, line 101 of the manuscript, and elsewhere in the text: there seems to be some confusion in the text about whether you are discussing the conditions in the porewater that the clays last equilibrated with, and whether those conditions are reflective of the lake water, the porewater, or the porewater at the moment before it froze or evaporated, etc. I recognize that the paper is trying to say something about chemical conditions in the lake and the ancient climate in Gale, but the authors occasionally toss in these caveats about the pore fluids representing a snapshot of the last fluid the phyllosilicates were in contact with, which may not have much to do with broader conditions in the lake/climate system when the sediments were first deposited in it.
6. Line 157-158: This seems like a weak constraint on Fe²⁺ concentrations given that there is only one data point for interlayer spacing in Fe-smectites in Fig 1a of the supplementary. I recognize that the authors can't do anything about this shortcoming of the available experimental literature, but the text on lines 157-158 is more definitive than it should be.

7. Line 184: What in-situ observations of glacio-fluvial features in Gale are consistent with the presence of surface ice? The referenced paper in this sentence (manuscript reference 17) is outdated and based on orbital data collected prior to Curiosity's arrival in Gale crater. To the best of my knowledge, none of the purported glacial/ice related features identified along Curiosity's traverse in that paper have been demonstrated to have such an origin by subsequent in-situ observations.
8. Line 186-187: Why can't redox disequilibrium at the time that the Sheepbed sediments formed just as easily explain the coexistence of the Fe²⁺/³⁺+saponite, magnetite, akageneite, and detrital sulfide minerals found in the Sheepbed member sediments, rather than calling on late stage oxidation by high Eh, low pH fluids?
9. Line 220-221: I think it's important to be forthcoming and state that these climatological constraints come from analysis of two drill holes from the base of the section – they may not apply to the full stratigraphic section at Gale crater. Only a similarly detailed analysis of lake water conditions based on phyllosilicate structure-composition from other drill holes allow you to extend these conclusions regarding climate higher up in the stratigraphy. It's also worth noting that the inference of arid/semi-arid conditions developed under cold climate conditions are consistent with independent constraints from bulk chemistry [3,5].
10. Line 412: reference needed.
11. Lines 450-453: see comment 2a (above).
12. Lines 454-459: see comment 2b (above).
13. Line 484: How can one assume Fe²⁺ concentrations are negligible pore waters? Sheepbed member mudstones contain secondary Fe-oxides (akageneite, hematite, magnetite) and Fe-phyllosilicate... surely there must have been some Fe²⁺ in solution for these phases to have formed?
14. Equation S3 of the Supplementary Info: shouldn't the right side of the charge balance equation be written $3\{SO_4^{2-}\}$?
15. Line 45 of the Supplementary info: should read Figs. 5a, 5b, and 5c
16. Line 46 of the supplemental: why is the condition $\log K_{Na_Ca} = \log K_{Na_Mg} = 0.2$ being considered at all? I thought the clay basal reflections constrained your analysis to 0.4-0.6 (Supplementary Figure 2).
17. Lines 144-145 are simply incorrect with regards to the JK and CB drill holes.

References:

1. Grotzinger, J. P., et al. (2014). "A Habitable Fluvio-Lacustrine Environment at Yellowknife Bay, Gale Crater, Mars." *Science* 343(6169).
2. Grotzinger, J. P., et al. (2015). "Deposition, exhumation, and paleoclimate of an ancient lake deposit, Gale crater, Mars." *Science* 350(6257).

3. Hurowitz, J. A., et al. (2017). "Redox stratification of an ancient lake in Gale crater, Mars." *Science* 356(6341).
4. Rivera-Hernandez F., et al. (2018). "In a PICL: the sedimentary deposits and facies of perennially ice-covered lakes." *Sedimentology* doi: 10.1111/sed.12522.
5. McLennan, S. M., et al. (2014). "Elemental Geochemistry of Sedimentary Rocks at Yellowknife Bay, Gale Crater, Mars." *Science* 343(6169).
6. Vaniman, D. T., et al. (2014). "Mineralogy of a Mudstone at Yellowknife Bay, Gale Crater, Mars." *Science* 343(6169).
7. Bristow, T. F., et al. (2015). "The origin and implications of clay minerals from Yellowknife Bay, Gale crater, Mars." *American Mineralogist* 100(4): 824-836.
8. Siebach, K. L., et al. (2014). "Subaqueous shrinkage cracks in the Sheepbed mudstone: Implications for early fluid diagenesis, Gale crater, Mars." *Journal of Geophysical Research-Planets* 119(7): 1597-1613.
9. Stack, K. M., et al. (2014). "Diagenetic origin of nodules in the Sheepbed member, Yellowknife Bay formation, Gale crater, Mars." *Journal of Geophysical Research-Planets* 119(7): 1637-1664.
10. Tosca, N. J., et al. (2018). "Magnetite authigenesis and the warming of early Mars." *Nature Geoscience*.
11. Caswell, T. E. and R. E. Milliken (2017). "Evidence for hydraulic fracturing at Gale crater, Mars: Implications for burial depth of the Yellowknife Bay formation." *Earth and Planetary Science Letters* 468: 72-84.
12. Vaniman, D. (2012). "Mars Science Laboratory Chemistry & Mineralogy RDR Data V1.0, MSL-M-CHEMIN-5-RDR-V1.0, NASA Planetary Data System."
13. Nachon, M., et al. (2014). "Calcium sulfate veins characterized by ChemCam/Curiosity at Gale crater, Mars." *Journal of Geophysical Research-Planets* 119(9): 1991-2016.
14. Ming, D. W., et al. (2014). "Volatile and Organic Compositions of Sedimentary Rocks in Yellowknife Bay, Gale Crater, Mars." *Science* 343(6169).
15. Frydenvang, J., et al. (2017). "Diagenetic silica enrichment and late-stage groundwater activity in Gale crater, Mars." *Geophysical Research Letters* 44(10): 4716-4724.
16. Hausrath, E. M., et al. (2018). "Reactive transport and mass balance modeling of the Stimson sedimentary formation and altered fracture zones constrain diagenetic conditions at Gale crater, Mars." *Earth and Planetary Science Letters* 491: 1-10.
17. Yen, A. S., et al. (2017). "Multiple stages of aqueous alteration along fractures in mudstone and sandstone strata in Gale Crater, Mars." *Earth and Planetary Science Letters* 471: 186-198.

18. Rampe, E. B., et al. (2017). "Mineralogy of an ancient lacustrine mudstone succession from the Murray formation, Gale crater, Mars." *Earth and Planetary Science Letters* 471: 172-185.

Responses to comments from Reviewer #1 for ‘Dynamic climate and redox interactions on early Mars inferred from water chemistry at Gale’ by Fukushi, K. et al. (MS# NCOMMS-18-21962)

We are grateful to reviewer #1 for the constructive comments and suggestions. Below, we give our responses in turn following each comment, with the reviewers’ comments being in Arial font and our responses being in Times font.

General Comment

The manuscript is well written. It suggests that the lakes developed at Gale on Mars are part of a global hydrological cycle during warmer episodes in the early Hesperian. In my opinion this is an overstatement. Based on the top-down approach that the authors are discussing in this manuscript, and that is deduced from Curiosity observations of a few places within Gale, a local hydrological cycle can be the maximum that could be inferred (although I see in the manuscript, even that would need some more proofs). The manuscript is based in too many assumptions that would need evidences to support the conclusions in this article.

Response to General Comment

The major concerns raised by the reviewer can be summarized as follows; 1) the need to consider the effect of local phenomena, such as contributions of eolian dust and deliquescence of NaCl into the Gale lakes, and 2) the need to verify the assumptions used in our geochemical calculations.

Concerning the point 1), the reviewer mentions that the water chemistry and hydrological cycles reproduced by the present study were local. This is correct. In especially, the rewetting event that caused the post-depositional fluids would have been local within Gale. Nevertheless, such near-surface acidic fluids are proposed to have generated at Meridiani, as observed by the Opportunity rover (e.g., Squyres et al., 2004 Science; Toska et al., 2005 EPSL; Hurowitz et al., 2010 Nature Geosci.; Arvidson et al., 2016 Am. Mineral.) and are suggested to have formed at multiple locations on Mars from orbital observations (e.g., Mangold et al., 2005 Icarus; Bibring et al., 2006 Science; Ehlmann et al., 2016 JGR Planets). In the present study, we show that similar acidic alterations have caused at Gale. We further propose that acidic fluids contain the abundance of high-Eh oxidants, which is new findings of our study. This has not been confirmed at other locations. However, given the extensive occurrence of acidic

near-surface fluids, we consider that similar oxidant supply would have occurred at multiple locations on Mars. We believe that this has important implications for chemical evolution of surface materials and habitability of this planet.

As for warming periods on early Mars, Gale Crater is expected to preserve a typical record of paleoclimate of early Mars, based on climate models of early Mars (e.g., Wordsworth et al., 2015 JGR Planets and new Supplementary Fig. 3 of the present paper). Gale Crater is located in a low-latitude area near the boundary of southern highlands and northern lowlands, where high levels of hydrological activity are suggested to have occurred from evidence of widespread valley networks and clay mineral formations (Bishop et al., 2018 Nature Astron.; Wordsworth et al., 2015 JGR Planets). Thus, paleoclimate constrained at Gale can be applicable to other low-latitude regions.

As mentioned in the specific comments below, the reviewer also concerns about local inputs of eolian silicate dust, eolian NaCl, and deliquescence of NaCl into the Gale lakes. The inputs of deliquescence of NaCl can occur at terrestrial hypersaline lakes in hyperarid, hot desert climates. If these occurred at Gale, a total duration of the lakes would be shortened (see below in Response to Comment 4). However, the paleoclimate and paleolake water constrained by the present study are only semiarid and hyposaline (weakly-to-moderately saline), respectively. Thus, we believe that a supply of deliquescence of NaCl would be very limited, compared with Na supply from surface/groundwater (see Responses to Comments 2 and 3 below for details). As shown below, we have added the discussion on the effect of eolian NaCl input in the revised manuscript (see Responses to Comments 2–4 below for details).

Regarding the point 2), as the reviewer mentioned, the estimated water chemistry depends on the assumptions used in the geochemical modeling. Some of these assumptions are based on the hydrogeological context of Gale Crater. To clarify/verify our assumptions, we have made a remarkable revision in the manuscript. First, we have added an overview of the hydrogeological context of Gale Crater together with schematic illustrations (new Fig. 1) based on the previous studies (see Response to Comment 6 below for details). Second, we have added our results of re-analyses of Curiosity's data (new Supplementary Fig. 1) and geological surveys for terrestrial hyposaline lakes in semiarid regions on Earth (see Response to Comment 5) to explain the validity of our assumptions. We believe that these additional explanations and figures in the revised manuscript would improve the validity of our assumptions.

The largest uncertainty in the original manuscript was perhaps the assumption that the post-depositional fluids fully interacted with the matrix of the Yellowknife Bay

mudstones in the rewetting event. However, this assumption would not be valid if the post-depositional fluids were only short-lived and tenuous. To investigate a full range of possibility, we consider two end-member scenarios concerning the interactions between the fluids and the mudstones in the revised manuscript: One is the “full-interaction” scenario, in which the fluids fully interacted with the mudstones in the rewetting event as considered originally, and the other is the “no-interaction” scenario, in which the fluids did not interact with the mudstones. To this end, we have newly performed a series of calculations for the no-interaction scenario and have added the results as a new section of “no interaction scenario” in lines 270–315 of the revised manuscript. Although we consider two different scenarios, our conclusions (hyposaline paleolakes, moderate pH, and a total lake duration of 10^4 – 10^6 years) do not change significantly, suggesting robustness of our conclusions.

Specific Comments

Comment 1: A Need of dynamism

As the authors state "Although the top-down approach cannot trace the water chemistry temporal evolution", YES, this is an important limitation of top-down approach and this is why inferring any dynamism (like “dynamic climate” as even indicated the title of the manuscript), requires extreme caution and supporting evidences. I really do not see these evidences in the manuscript.

Response to Comment 1: We have clarified the need of dynamic interactions of water chemistry in the revised manuscript.

The top-down approach cannot trace the evolution of water chemistry, but it can distinguish whether the snapshot of the water chemistry can be generated by a single process or would require a combination of multiple distinct processes. In the original manuscript, we try to propose that the estimated water chemistry cannot be interpreted without dynamic mixing of multiple waters with different water chemistries. For example, formation of saponite requires alkaline pH (> 9) based on its thermochemical stability and terrestrial field observations (see Bristow et al., 2015 Am. Minel. and our new Fig. 4). However, pH of the pore water that finally interacts with the Yellowknife Bay sediments is estimated to be circumneutral (pH ~ 7) (see the main text). This apparent discrepancy can be explained by an intrusion of acidic post-depositional fluids into the alkaline pore of the sediments. Akaganeite, associated with saponite in the matrix of the Yellowknife Bay mudstone (McLennan et al. 2014 Science), also supports

intrusions of acidic fluids. This is the case because formation of akaganeite needs relatively low pH (pH 2–8) (Peretyazhko et al. 2018 JGR Planets), suggesting that akaganeite and saponite are highly unlikely to be generated together by a single process. Akaganeite in Yellowknife Bay sediments was most likely occurred by reactions of acidic post-depositional fluids with the matrix of the mudstone. These results suggest the dynamic changes in aqueous chemistry at Gale. To emphasize the need of dynamic interactions of different water chemistries, we have added the following sentence in lines 240–243 of the revised manuscript.

“Given that the formation of akaganeite requires relatively low pH (2–8)²⁶, the co-existence of smectite, magnetite, and akaganeite in the sediments can be explained by the alteration of the alkaline matrix of the mudstone by post-depositional acidic fluids, thereby suggesting dynamic changes in water chemistry.”

Concerning climate, we suggest that the acidic post-depositional fluids was generated through melting of surface ice/snow by a temporal warming event. In this regard, we suggested the occurrence of “dynamic” climate changes (i.e., temporal warming) on early Mars.

Comment 2: Effect of eolian dust and intermittent evaporation

This sentence (lines 73-75 in the original WORD document): "Hence, the pore-water chemistry recorded in the Yellowknife Bay sediments most likely represents saturated pore water immediately before freezing." This is not sufficient enough to exclude the possibility of intermittent melting and evaporation. If the authors are talking about a persistent 10^4 – 10^6 years' “water cycle” then completely excluding the evaporations scenarios (which are more likely in the suggested arid climate) is not a valid approach. Since it is such a huge time-scale and the authors are considering climate dynamism, would it not be appropriate to consider wind erosion or wind deposition layering to play some part in regulating the ionic concentrations which we are observing today?

Response to Comment 2: Deposition of eolian dust does not changes our estimate of the ion concentrations. We have added the effect of intermittence of lakes for an estimate of a total lake duration in the revised manuscript.

We appreciate that the reviewer pointed out the possibility of evaporation of the early Gale lakes and deposition of eolian dust onto the lake sediments. Indeed, the deposition/erosion of eolian dust onto the lake sediments does not change the estimated

ion concentrations of the pore water. This is because the ion concentrations are estimated from the basal spacing of smectite and the mineral composition (not abundance) of salt minerals/iron oxides (i.e., gypsum/akaganeite) (see Methods), which are not regulated by eolian dust deposition. In addition, the Yellowknife Bay (Sheepbed) mudstone shows no evidence of drying of lakes, such as desiccation cracks, at the time of deposition (e.g., Grotzinger et al., 2014 Science). This suggests that, at least during the time of deposition of the Sheepbed mudstones, the possibility of evaporation of lake waters can be ruled.

Having said that, the Sheepbed mudstone is only 1.5 m-thick lacustrine sediments (e.g., Grotzinger et al., 2014 Science), and thicker sediments may lie beneath the Sheepbed mudstone. Our estimate of a total lake duration of 10^4 – 10^6 year includes the deposition time of the underlying lacustrine sediments beneath the Sheepbed mudstones. Intermittence of lakes during the deposition of the underlying lacustrine sediments cannot be ruled out. If all of Na in the earlier lakes was deposited as halite and buried in the deep sediments upon long-term intermittence of lakes, a total duration of lake must be reset. This means that the estimated total duration by the present study is a cumulative time of the lakes since the last timing of complete burial of halite, if occurred. In other words, short-term intermittence of lakes without removal of Na^+ as halite burial does not reset a total duration. To clarify this point, we have added the following sentence in lines 350–351 of the revised manuscript.

“Thus, short-term intermittency of lakes without removal of Na^+ as halite burial does not reset a total duration.”

Comment 3: Effect of eolian NaCl deposition

Since the authors are relying on terrestrial analogies for deriving such inferences, they must also consider the major role that various environmental factors play in depositional events of inorganics over a period of time. A good example of this which will modify the deductions of the present research can be found in <https://www.hydrol-earth-syst-sci.net/14/801/2010/> .

Line 89: "Despite the occurrence of post-depositional alterations". What about Aeolian deposition at millennial scales? Can't that be another possible contributor?

Response to Comment 3: Supply of eolian NaCl into the early Gale lakes would reduce the estimated total duration of the lakes. We added this possibility in the

text.

This is an interesting comment. As discussed in Response to Comment 2 above, eolian NaCl would not be directly embed into the Sheepbed mudstone given the lack of evidence for drying of lakes at the time of deposition (e.g., Grotzinger et al., 2014 Science). However, eolian NaCl deposition to the early Gale lakes can be another source of Na in the lake water. This could reduce a total duration of early Gale lakes shown in the original manuscript because we consider only surface/groundwater as the Na sources.

Nevertheless, XRD patterns of the Rocknest eolian deposit near the Yellowknife Bay sediments show the absence of chloride salts within the modern eolian dust. In addition, formation of eolian NaCl requires the existence of high salinity oceans at that time. However, it is highly unclear whether such high salinity oceans existed on early Mars given a lack of extensive halite layers in the northern lowlands (e.g., Osterloo et al., 2008 Science). According to the above discussion, we also added the following sentence in lines 357–359 of the revised manuscript.

“This could be the upper limit of a total duration given possible inputs of eolian NaCl from an ancient Martian ocean; nevertheless, it is highly uncertain whether high salinity oceans existed on early Mars at that time.”

Comment 4: Effect of deliquescence of salts as water source

Lines 102-104: "On Earth, saline lakes are abundant in semiarid to arid climate regions, such as inner-continental steppe/desert areas^{21,22}. Most of the terrestrial saline lakes in continental areas are terminal lakes without any outflowing rivers." Yes, and such lakes are also a result of deliquescence facilitated by salt-rich regolith in the terrestrial environments. This might would have been the case in the time period that the present study is considering for the Gale lakes. However, the study here is completely neglecting this possibility.

Response to Comment 4: Given the proposed semiarid climates at Gale, the possibility of deliquescence of salt layers would be unlikely. We have clarified this point in the revised manuscript.

This is another interesting possibility. According to the terrestrial arid environmental studies, however, deliquescence of salt-rich regolith plays a major role in hydrological cycles and ecosystem only in hyperarid, hot desert regions (precipitation < 100 mm/year), such as Atacama Desert (e.g., Wierzchos et al., 2006 Astrobiology; Wierzchos et al., 2012 Biogeosciences). By contrast, our results suggest a semiarid,

steppe climate to support the early Gale lakes (see the main text). In semiarid steppe regions, deliquescence of salts usually plays only a minor role in hydrological cycles, compared with precipitation, groundwater upwelling, and snow melting (e.g., Szumińska, 2016 Sediment. Geol.). In addition, no chloride-rich layers are observed at Gale by both of the Mars orbiters and Curiosity Rover thus far, which also does not support deliquescence of salts as a major water source. As shown below in Response to Comment 5, we have clarified that the estimated salinity of the early lakes by the present study is not hypersaline but is only weakly-to-moderately saline (hyposaline) in the revised manuscript.

Comment 5: Possibility of the presence of salt flats at early Gale lakes

Lines 104-106: "Lake levels of terminal lakes are at lower elevations than potential outflowing rivers and are maintained by a balance between inflowing water and evaporation." Such continental saline lakes in salt flats are a major source of ground water recharge and due to their often shallow nature, their boundaries are not very rigid and may vary at seasonal to yearly or decadal scales since deliquescence also plays significant role in their water balance in the arid climate. Please consider such possibilities if you intend to establish a good terrestrial analogy.

Response to Comment 5: We have added discussion on the possibility of salt flat in the main text, although this would be unlikely for early Gale lakes. We also added discussion and observations of hyposaline lakes on Earth, as terrestrial analogs.

Although we appreciate the reviewer's suggestion, we consider that formation of early Gale lake in salt flats is unlikely for the following reasons; a) our results of weakly-to-moderately saline (hyposaline) water are inconsistent with the presence of extensive salt flats. This is because lake water in salt flats should be saturated with halite. b) Mars orbiters and Curiosity rover do not observe any chloride-rich deposits/layers at Gale (as mentioned above in Response to Comment 4). c) Curiosity also does not find geological evidence of drying of lakes, such as desiccation cracks, for the Yellowknife Bay sediments (Grotzinger et al., 2014 Science), which usually occur in association with salt flats (e.g., Gutiérrez, 2005; ref. 22).

We would suppose that the wording of "saline" in the original manuscript might be misleading. Due to this, the reviewer could imagine hypersaline lakes in hot desert climates as terrestrial analogs of the early Gale lakes. However, we propose only

weakly-to-moderately saline (hyposaline) lake water, as the estimated salinity ranges only 0.1–0.2 mol/kg. To avoid this misunderstanding, we have changed the wording of “saline” in the original manuscript to “hyposaline (or weakly-to-moderate saline)” throughout the revised manuscript.

Regarding good terrestrial analogs of early Gale lakes, we have added discussion and pictures of our field surveys for hyposaline lakes on Earth, including those in south Mongolia and central India. A lack of extensive desiccation cracks surrounding these hyposaline lakes developed in semiarid climates are consistent with the observations of the Yellowknife Bay sediments by Curiosity (Grotzinger et al., 2014 Science). In the revised manuscript, we have added the following sentences in lines 147–152 and new Supplementary Fig. 2 in revised Supplementary Information.

“At Yellowknife Bay, Curiosity has found no textural characteristics showing lake drying, such as desiccation cracks and rip-up chips⁹. These characteristics are extensive in drying hypersaline lakes (e.g., playa lakes or salt pans) developed in (hyper)arid desert climates²², but they are not pervasive around terrestrial hyposaline lakes (see Supplementary Fig. 2). Thus, the absence of desiccation cracks at Yellowknife Bay would not be incompatible with our conclusion of hyposaline, early Gale lakes.”

Comment 6

Lines 221-223: "Our results of water chemistry suggest that Gale would have experienced prolonged (10^4 – 10^6 years) episodes of warm and semiarid climates in total, although these episodes might have been intermittent." Exactly what I mentioned earlier. Such episodes are intermittent and completely substituting evaporation with sublimation is a very weak analogy.

Response to Comment 6: We have modified the original confusing explanation of our scenario by adding a brief overview and new figure of deposition of Gale sediments.

We are afraid that there may be misunderstanding. In the original manuscript, we tried to propose that freezing and subsequent sublimation of the pore water occurred after the final rewetting event (not the termination of the early Gale lakes). The formation and termination of the early lakes would have occurred much earlier (e.g., > 0.1 billion years) than the final wetting event (Grotzinger et al., 2014; 2015 Science).

The misunderstanding would have caused by the confusing explanation of our scenario in the original manuscript. To improve this point, we have added a brief overview of the hydrogeological context of the Yellowknife Bay sediments in lines 36–

46 of the revised manuscript together with schematic illustrations (new Fig. 1: also attached below). The hydrogeological context is based on the previous studies on geological, geochemical, and mineralogical features of the Yellowknife Bay sediments (Grotzinger et al. 2014 Science; 2015 Science; Vaniman et al. 2014 Science; McLennan et al. 2014 Science; Hurowitz et al. 2017 Science, and others). We hope that this would be useful to understand our results and interpretations.

“The Yellowknife Bay Formation was deposited by flow deceleration as a river(s) encountered the lake in Gale Crater^{9,10,12} (Fig. 1a). The bulk mineralogy contains significant amounts of secondary non-detrital phases, including smectite and iron oxides¹¹. This suggests that most of chemical reactions that resulted in the generation of phyllosilicates and other secondary minerals took place after deposition, during early diagenesis¹¹⁻¹⁴ (Fig. 1b). The Yellowknife Bay sediments were subjected to burial pressure and pervasively fractured after the deposition and early diagenesis^{9,10} (Fig. 1c). During the rewetting events, the late-diagenetic (post-depositional) fluids were introduced into the fractures, which are currently filled with calcium sulfate (CaSO₄); presumably a substantial amount of time after the period of the early lakes^{9,11,15} (Fig. 1d). These rewetting events occurred within Gale even after the formation of Aeolis Mons¹⁶. After the final rewetting event, liquid water in the Yellowknife Bay sediments disappeared (Fig. 1e).”

New Fig. 1. Schematic illustrations of proposed hydrogeological context of Yellowknife Bay Formation at Gale Crater⁹⁻¹⁶. The panels (a)–(e) represent hydrogeological evolution of a part of Gale Crater and close-up illustrations of the Yellowknife Bay sediments (from (a) to (e) in time sequence): (a) the early lakes and deposition of the sediments, (b) early diagenesis, (c) burial and fracturing, (d) rewetting with post-depositional (late-diagenetic) fluids, and (e) the present day. The dashed lines show the horizon of Yellowknife Bay Formation. In (d) and (e), the full- and no-interaction scenarios are compared in the close-up illustrations. Formations of secondary minerals, e.g., smectite, iron oxides, halite, and sulfate, are also described. See the main text for details.

Responses to comments from Reviewer #2 for ‘Dynamic climate and redox interactions on early Mars inferred from water chemistry at Gale’ by Fukushi, K. et al. (MS# NCOMMS-18-21962)

We are grateful to reviewer #2 for the constructive comments and suggestions. Below, we give our responses in turn following each comment, with the reviewers’ comments being in Arial font and our responses being in Times font.

General Comment

Some of the assumptions underlying the calculations are questionable and alternative explanations are underrepresented in the manuscript. Table 1 presents a unified, comprehensive estimate of pore water chemistry of Yellowknife Bay, but as alluded to throughout the manuscript, the aqueous history of the sediments is complex and the influence of multiple generations fluids are expected over a geologically protracted time (e.g. Martin et al., 2017–JGR Planets 122, 2803-2818). The manuscript does not successfully reconcile the chemical constraints presented in Table 1 with the aqueous history discussed. This is because the manuscript contains multiple examples of conflicting statements, such as:

Line 62: ‘Given evidence for the occurrence of local wetting events within Gale Crater after deposition of lacustrine sediments^{16,17}, the estimated compositions would represent the chemistry of the pore water provided in the last wetting event at Yellowknife Bay.’ [In this case the ‘wetting event’ being referred are not related to the original lake waters – based on references cited they are waters from much later lacustrine systems or glacial meltwaters]

Line 83: “Given the proposed geo-hydrological evolution of Gale Crater^{16,17}, the estimated pore-water chemistry would reflect combinations of both components from lake water trapped within the pores of the Yellowknife Bay sediments upon deposition, and components from post-depositional fluids during late diagenesis.”

Line 99: “Our results of the pore-water chemistry suggest that the water chemistry of the early Gale lakes would be characterized as moderately high in salinity at the time of the Yellowknife Bay sediments deposition.”

As a result, it is unclear what and when chemical constraints apply to the various generations of fluids influencing the sediments. Moreover, the methods of

calculation assume mineral equilibria between phases that were products of temporally and geologically distinct aqueous events.

Response to General Comment

The major concerns raised by the reviewer can be summarized as follows; 1) the need to clarify how we can distinguish the effect of various generations of fluids from the estimated pore water chemistry, and 2) the uncertainty to assume mineral equilibria between the post-depositional fluids and the matrix of the Yellowknife Bay mudstones.

With respect to the point 1), we consider that the chemical properties of the final pore water are determined by mixtures of the primary components originating from the lake water and the additional components from the post-depositional fluids. Namely, salinity, pH, Eh, and inorganic carbon contents in the final pore water are controlled by both of the lake water trapped within the sediments and the post-depositional fluids. As shown below (in Response to Comment 5), the salinity is considered to come mainly from the lake water because the post-depositional fluids were depleted in Na. Thus, we mentioned that “Our results of the pore-water chemistry suggest that the water chemistry of the early Gale lakes would be characterized as moderately high in salinity....” in the original manuscript. In the original manuscript, the explanation of the source of Na was missing; accordingly, the reviewer may feel that this is a conflicting statement.

In the revised manuscript, we have explained this point very carefully. For instance, we show geochemical evidence that Na would have been depleted in the post-depositional fluids based on our re-analyses of Curiosity’s ChemCam data (new Supplementary Fig. 1) and we have clearly mentioned that estimated Na in the final pore water was mainly derived from the trapped paleolake water within the sediments (see Response to Comment 5 for details). Additionally, we have added the statements to explain that pH, Eh, and inorganic carbon contents are determined by both of the primary and additional components (see Response to Comment 5 below for details). In the revised manuscript, we try to explain these situations as much as possible within the word limitation.

Concerning the point 2), we admit that there is an alternative possibility that mineral equilibria was not achieved in the rewetting event due to a short residence time of the post-depositional fluids (see Response to Comment 4 for details). To examine a full range of possibility, we have remarkably modified the structure of the manuscript. Namely, we have discussed two end-member scenarios in the revised manuscript: One is the “full-interaction” scenario, in which mineral equilibria were achieved in the

rewetting event as considered originally, and the other is the “no-interaction” scenario, in which the mineral equilibria were not achieved. In the latter scenario, we have newly performed a series of calculations to estimate the water chemistry of the paleolakes based on the spacing of smectite and secondary minerals contained in the matrix of the Sheepbed mudstones. Our results show that the estimated concentrations of Na and pH of the paleolake water for the no-interaction scenario show agreements with those for the full-interaction scenario. Thus, our conclusions (hyposaline lake water, moderate pH, and a total duration of the early lakes) do not change significantly (see Response to Comment 4 below for details). We believe that this agreement suggests robustness of our conclusions.

Based on the reviewer’s other major comments, the discussion of the validity of important assumptions has been also added in the revised manuscript (see Responses to Comments 1–3 below for details). We appreciate the reviewer’s constructive comments and helpful suggestions, which have greatly improved the manuscript. We hope that our revision would address the reviewer’s concerns.

Major Comments

Comment 1: Interlayer chemistry 1 (explanation of 13 angstrom spacing)

Determining the composition of exchangeable cations within smectite in Yellowknife Bay samples provides a central constraint on fluid chemistry. To do this the manuscript examines the d-spacings of smectite in John Klein and Cumberland. The sensitivity of d-spacing of smectites to exchangeable cation species is well documented and these two samples have d-spacings of ~10 angstroms and ~13 angstroms. This is explained as a result of mixed Na^+/K^+ and Mg^{2+} cations in interlayers, because small variations in the ratio of monovalent and divalent species can produce quite different d-spacing. This is required because the two samples are within meters of each other and pore fluids should not have been vastly different.

The manuscript should discuss why an alternative explanation for 13 angstrom spacing of Cumberland presented in Vaniman et al., 2014 and Bristow et al., 2015 should be ruled out. In this scenario the interlayer is kept open by partial intercalation of Mg-hydroxy interlayers.

Response to Comment 1: We have added the discussion on the possibility of MgOH^+ interlayers, but we consider that this cannot explain ~13 angstroms

spacing in our scenario, as described below.

In the revised manuscript, we have added discussion why the partial interaction of Mg-hydroxyl (MgOH^+) interlayers can be negligible when assuming the water chemistries reproduced at between the two drill holes were similar. The dissolved MgOH^+ species become predominant among Mg-bearing ions only at extremely high pH (pH > 12: $\text{pK}_a = 12.8$ at 0 °C) at ambient to low temperatures. However, solubility of ΣMg^{2+} ($= \{\text{Mg}^{2+}\} + \{\text{MgOH}^+\} + \text{etc}$) also becomes extremely low at these high pHs, because of precipitation of brucite ($\text{Mg}(\text{OH})_2$) at pH > 10 (see Ancillary Fig. 1 below). Hence, the probability of exchange between dissolved MgOH^+ species and other interlayer cations, such as Na^+ , must be extremely low, resulting in negligible amounts of MgOH^+ interlayers. In order to form MgOH^+ -rich interlayers, extremely high $\text{Mg}^{2+}/(\text{Na}^++\text{K}^+)$ ratios would be required for solutions. However, since we consider that pore water chemistries at two drilling sites were similar, the extremely high $\text{Mg}^{2+}/(\text{Na}^++\text{K}^+)$ ratios, in turn, contradicts with Na/K interlayers in the sediments at the other drilling site. Since the previous studies (Vaniman et al., 2014 Science; Bristow et al., 2015 Am. Mineral.) assume that the water chemistries of these two drilling sites are different, the idea of MgOH^+ interlayer is valid for their models.

Ancillary Fig. 1 Speciation of Mg as function of pH considering the formation of brucite at 0°C (a). Formation of brucite ($\text{Mg}(\text{OH})_2$) as function of pH at 0°C (b). (same as new Supplementary Fig. 1)

Additionally, partially hydroxylated Mg intercalated smectite were found in terrestrial sediments of hypersaline lakes with extremely high salinity (> 300,000 mg/kg of total dissolved solid; Jones and Weir, 1983) and via neutral to mildly alkaline hydrothermal alterations at temperatures of 200–240 °C (Pevear et al. 1982). According to the proposed geological and mineralogical characteristics of the Yellowstone Bay sediments (e.g., Grotzinger et al. 2014 Science; Vaniman et al. 2014 Science),

nevertheless, such high salinity and/or high temperatures are not plausible for the depositional/alteration conditions for the Yellowknife Bay sediments.

To summarize, we have ruled out the possibility of MgOH^+ interlayer for self-consistency of our scenario. Based on the reviewer's comment, we have added the following sentences in lines 447–455 of the revised Methods.

“Partially intercalated hydroxylated Mg (i.e., MgOH^+) in the interlayer site of the smectite at Cumberland has been proposed as an explanation for the expanded structure^{11,14}. The dissolved MgOH^+ species becomes important at extremely alkaline condition ($\text{pH} > 12$: $\text{pK}_a = 12.8$ at 0°C) at ambient to low temperatures. However, solubility of Mg also becomes extremely low at high pH ($\text{pH} > 10$) because of formation of brucite. Therefore, given the low $\{\text{MgOH}^+\}$ at alkaline pH, the exchange of the dissolved MgOH^+ species into the interlayer occurs very inefficiently at ambient to low temperatures unless the $\text{Mg}^{2+}/(\text{Na}^+ + \text{K}^+)$ ratio in the pore water is extremely high. If assuming the water chemistries between the two drilling sites were similar, the extremely high $\text{Mg}^{2+}/(\text{Na}^+ + \text{K}^+)$ ratio contradicts with Na^+/K^+ interlayers for the John Klein smectite.”

Comment 2: Interlayer chemistry 2 (d-spacing of Mg^{2+} -exchanged saponite)

In addition, I don't think the manuscript provides adequate proof that Mg^{2+} exchanged saponite will have a d-spacing of 13 angstroms under the zero humidity conditions inside the CheMin instrument. From the studies available, including Suquet et al. 1975 cited in the manuscript, Mg-exchanged saponites have smaller d-spacings of 11–12 angstroms. The argument that Mg exchanged smectites will have d-spacings of 13 is based on data from montmorillonite and synthetic smectites, which may not be appropriate

Response to Comment 2: We have added the discussion on spacing of saponite in the revised manuscript. Given layer charge of ferrian saponite, however, we consider that Mg-saturated ferrian saponite at Yellowknife Bay would have large spacing, e.g., 13 angstroms.

In the revised manuscript, we have added the experimental data of interlayer spacings of Mg^{2+} -saturated smectites and vermiculate at $\text{RH} = 0\%$ from Rampe et al. (2014). They showed that the interlayer spacings of vermiculate (trioctahedral expanded clay minerals with high layer charge) and nontronite (dioctahedral Fe^{3+} smectite) exceeded 13 angstroms, while those of montmorillonite and Mg saponite were less than 13 angstroms.

The previous study suggests that the observed swelling behavior of different Mg^{2+} -saturated smectites are mainly due to variations in the layer charge and the charge

locations (Sato et al. 1992; Clays and Clay minerals). In fact, the basal spacings of Mg-saturated montmorillonite widely vary from 12.0 to 13.9 angstroms among the examined samples, supporting the idea that the basal spacings does not simply depend on the smectite species. Ancillary Fig. 2 shows the relationship between the cation exchange capacity and interlayer spacing for various smectites. This figure indicates a general trend that smectites with larger charge tend to have larger spacing.

Ancillary Fig. 2 Relationship between cation exchange capacities and interlayer spacings after Mg saturation from the previous studies (Suquet et al. 1975; Iwasaki, 1979; Cases et al. 1997; Morodome and Kawamura 2011 and Rampe et al. 2014), where cation exchange capacity data of smectite were available. The dotted line represents the regression of the data except for saponite from Suquet et al. (1975).

Smectite in the Yellowknife Bay sediments is ferrian saponite, which is highly likely to have been formed by alterations of ferrous saponite (Treiman et al., 2014 Am. Mineral.). The oxidation of Fe^{2+} to Fe^{3+} results in an increase of the layer charge. Accordingly, smectite in the Yellowknife Bay would have larger spacing, such as ~13 angstroms, rather than smaller spacing, such as ~11 angstroms, among Mg saponites examined in the previous studies (Suquet et al., 1975). Although the reviewer’s concern is important, we still consider that full discussion and examinations of the determining factors of smectite spacing would be beyond the scope of this study. In the present paper, we have briefly discussed the above point in lines 456–465 of revised Methods, as follows.

“Although the reported basal spacings of Mg-saturated Mg saponite vary from 11.5 Å

to 12.0 Å, we believe that Mg²⁺-saturated smectite is responsible for the spacing of ~13 Å of the Cumberland smectite. This is the case because the swelling behaviors of interlayer cations mainly depend on the layer charge and charge location⁵⁶. Vermiculite, high-layer-charge montmorillonite, and well-crystalline synthesized smectite exhibit higher basal spacing of ~14 Å (Supplementary Fig. 3a). These specimens are expected to possess high layer charges. The smectite found in the Yellowknife Bay sediments is considered to be ferrian saponite, which would have been formed by alteration of ferrous saponite³⁰. Since the oxidation of Fe²⁺ to Fe³⁺ should have resulted in an increase in the layer charge, the smectite in the Yellowknife Bay sediments would have larger spacings, e.g., ~13 Å, among the measured values of Mg-smectite (Supplementary Fig. 3a).”

Comment 3: Interlayer chemistry 3 (other lacustrine mudstones)

It is also noteworthy that all other clay minerals reported from other lacustrine mudstones in Gale have d-spacings of 10 angstroms (e.g. Rampe et al., 2017; Bristow et al., 2018)—which argues against an interlayer cation content that lies on a cusp that allows d-spacing to flip between 10 and 13 angstroms.

Response to Comment 3: We would consider that spacing of clays from other mudstones does not always contradict with our interpretation. Spacing of clays in the Murray mudstones can be explained by higher Na contents of the later lakes. This can occur due to an accumulation of Na within Gale Crater over time and/or a decrease in a lake water volume.

The fact that the upper lacustrine mudstones (i.e., the Murray mudstones) at Gale have spacing of 10 angstroms can be explained by an accumulation of Na within the lakes. The observed spacing of 10 angstrom suggests that the Na⁺ concentrations of the lake water during the deposition of the Murray mudstones are significantly higher than those when the Yellowknife Bay sediments deposited. An accumulation of Na in the Gale lakes over time would have resulted in higher Na concentrations when the Murray mudstones deposited. Additionally, any decrease in a lake volume should increase Na concentrations of the lake water. In fact, Curiosity recently found the evidence of desiccation cracks in the upper Murray mudstones (Stein et al., 2018 Geology), suggesting lake drying and, thus, low lake levels when the upper Murray mudstones deposited. Thus, we believe that the observed spacing of 10 angstroms would not always contradict with our interpretations.

Although we appreciate the reviewer’s interesting comment, the above discussion is still speculative and are out of scope of the present study. Thus, we would not discuss this topic in the present paper.

Comment 4: Assumptions in calculations

The derivation of pore water chemistries presented in Table 1 assumes that $[\text{Ca}^{2+}]$ and $[\text{SO}_4^{2-}]$ of fluids were in equilibrium with gypsum. While Ca-sulfates are components of both John Klein and Cumberland samples, they are “attributed to small veinlets that were quantified in the Yellowknife Bay boreholes (supplement to Vaniman et al. 2014), with little or no Ca-sulfate cement in the mudstone matrix.” (Vaniman et al., 2018–Am. Mineralogist, v. 103 page 1015.) The veinlets that record the passage of Ca and sulfate bearing fluids show that this fluid event post-dates lithification of the sediments and thus it may not be valid to assume that pore waters that were in equilibrium with exchangeable cation sites of clays were also in equilibrium with gypsum. The main problem being the inefficiency of post-lithification cation exchange of clays. In the cited paper describing the method for deriving pore water chemistry from clay rocks it is stated that: “It may be difficult, if not impossible, to obtain water samples for chemical analysis from such rocks because of their low hydraulic conductivity” (Gaucher et al., 2009). This brings up the question of how effective and pervasive post-lithification “wetting” and alteration events were at influencing exchangeable smectite interlayer sites?

Response to Comment 4: Agreed. The original manuscript discussed only a single scenario of full exchange of ions between the fluids and the sediments in the rewetting event. To examine a full range of possibility, we have newly calculated the water chemistry for the alternative scenario of no exchanges between the fluids and the sediments.

We agree that the assumption that the post-depositional fluids were equilibrated with both gypsum and smectite is the largest uncertainty of the present study. This may not be valid if the post-depositional fluids were only short-lived and tenuous. To examine a full range of possibility, we have investigated the alternative scenario, in which the post-depositional fluids and the matrix of the mudstone were not in chemical equilibrium (we call this scenario as the “no-interaction” scenario) in the revised manuscript. The no-interaction scenario implies that the residence time of the fluids is too short to seep into the matrix of the mudstone. This can happen when the fluids persisted in fractures of the mudstone for $< \sim 1-10^2$ years, using terrestrial analog permeability of dried marine claystone ($10^{-7}-10^{-10}$ cm/s; Davis, 1969) and typical distance of fractures in the Yellowknife Bay sediments (~ 1 cm).

In the no-interaction scenario, we have newly calculated the water chemistry of the early lakes based on the spacing of saponite and the presence of akaganeite and halite, which are considered to be contained in the matrix of the mudstone (e.g., Vaniman et al., 2014 Science; McLennan et al., 2014 Science; Hurowitz et al. 2017 Science). Our new results show that 0.04–0.27 mol/kg of Na concentrations for the paleolake water for the no-interaction scenario (see new Table 2 in the revised manuscript). The estimated Na concentrations show good agreements with those of the full-interaction scenario (0.14 mol/kg: Table 1), in which the fluids and the mudstones were equilibrated. Although the allowed pH range is slightly wider than that of the full-interaction scenario, the estimated pH is also circumneutral (pH ~7–8) for the no-interaction scenario. Thereby, our major conclusions (hyposaline lakes, moderate pH, and its total duration of 10^4 – 10^6 years) do not change significantly for both of the scenarios. In the revised manuscript, we have added a new section of “no-interaction scenario” to explain these results in lines 270–315 of the revised manuscript. The calculation methodology of the no-interaction scenario has been also added in revised Methods.

Although we have investigated the no-interaction scenario, we still consider that the full-interaction scenario would be favorable for the following reasons. a) The detailed analyses for Curiosity’s APXS data show that small amounts of CaSO_4 (1.4 wt.% SO_3 in average) may exist in the matrix of the mudstone without visual Ca-sulfate veins (Schmidt et al., 2018 JGR Planets). Given the penetration depth of 0.1 mm for alpha particles, hidden Ca-sulfate veins beneath the surface would be unable to explain the CaSO_4 in the mudstone matrix. The existence of CaSO_4 in the matrix of the mudstones suggests that the sulfate-rich post-depositional fluids may have diffused into the mudstone in the rewetting event.

In addition, as discussed in the revised main text, the no-interaction scenario requires extremely rapid transport and high accumulation rates of sediments in order to explain co-existence of oxidizing akaganeite and reducing pyrrhotite in the matrix of the mudstone. This is because reducing pyrrhotite is readily oxidized under the oxidizing conditions where akaganeite can be formed. This, in turn, implies a short duration of the Gale lakes given the thickness of the Gale sediments. Such a short duration could be incompatible with development of hyposaline lakes since a substantial duration would be required to accumulate the estimated salinity. Thus, although we cannot conclude which scenario was the case, we consider that the most straightforward interpretation would be the full-interaction scenario; namely, seepage of acidic-oxidizing post-depositional fluids into the alkaline-reducing mudstones to form circumneutral pH

and redox disequilibria.

Based on the above discussion, we also have added a new section of “The full-interaction scenario vs the no-interaction scenario” in lines 316–338 of the revised manuscript to compare these two scenarios.

Comment 5: Overprinting and evolution of pore water chemistry

The manuscript argues that ‘pore-water chemistry suggests that the water chemistry of the early Gale lakes would be characterized as moderately high in salinity at the time of the Yellowknife Bay Sediment deposition. The estimated Na-Cl concentrations are slightly lower than terrestrial seawater, but significantly higher than freshwater.’ This is the basis of mass balance calculations used to estimate lake lifetime. In contrast, other chemical characteristics of the pore fluids such as pH, Eh and inorganic carbon content are proposed as being governed by later fluid events—the passage of melt-induced acidic-oxidizing fluids is stressed toward the end of the manuscript, although ‘wetting events’ are mentioned earlier. It is unclear to me how the characteristics of multiple generations of fluids can be isolated like this with the available data and this is the heart of the problem with this manuscript.

Response to Comment 5: Agreed. The original descriptions would be confusing due to a lack of detailed explanations. We consider that all of the chemical properties of the final pore water are determined by mixtures of the primary components (from the lake water) and the additional components (from post-depositional fluids). We have clarified this in the revised manuscript.

In the full-interaction scenario, we consider all of salinity (Na contents), pH, Eh, and inorganic carbon contents are determined by mixtures of the primary components originating from the paleolake water and the additional components originating from the post-depositional fluids. We have emphasized this point repeatedly in the revised manuscript. For instance, we have added the following sentences in lines 68–71 of the revised manuscript;

“In the full-interaction scenario, we consider that the primary components (such as smectite) in the matrix of the sediments equilibrate with the additional components (such as calcium sulfate) in pore water (Fig. 1d). Thereby, the estimated water chemistry would reflect mixtures of the primary and additional components.”

and added the following sentence in lines 133–136 of the revised manuscript;

“Although the dissolved species in Table 1 are mixtures of the primary and additional components within the sediments, we believe that the predominant dissolved species of Na^+ and Cl^- in the pore water would have largely originated from the primary component; namely, lake water trapped within the Yellowknife Bay sediments (Fig. 1a, b).”

and added the following sentence in lines 172–174 of the revised manuscript.

“Hence, in the full-interaction scenario, the pore-water pH for the Yellowknife Bay sediments is determined by both of the primary components from lake water in the early diagenesis and the additional components from the post-depositional fluids in the late diagenesis.”

As for Na contents, the post-depositional fluids are highly likely to have been depleted in Na based on the Curiosity’s observational data (see Supplementary Fig. 1, data from Nachon et al., 2014 JGR Planets). To show this, we have performed re-analysis of Curiosity’s ChemCam data (Nachon et al., 2014 JGR Planets) for the Yellowknife Bay sediments that contain sulfate veins (new Supplementary Fig. 1). Our new results of Supplementary Fig. 1 show that the Na contents in the sediments exhibit an inverse correlation with those of SO_3 and a positive correlation with those of detrital components, such as Al (see below). This strongly suggests that Na was mainly contained in the matrix of the mudstone and was also depleted in the post-depositional fluids. Thus, we interpret that Na in the final pore water was largely derived from the paleolake water.

Supplementary Fig. 1 Plots of CaO vs. SO_3 (a), Al_2O_3 vs. SO_3 (b), NaO vs. SO_3 (c) and NaO vs Al_2O_3 (d) based on the LIBS measurements for the Yellowknife Bay mudstones (Sheepbed member) with Ca-sulfate veins¹⁵. The straight line in panel (a) indicates the stoichiometry of calcium sulfates (CaSO_4). The agreements of the CaO and SO_3 with the stoichiometry suggests that calcium and sulfur in the bulk mudstone predominantly are largely derived from CaSO_4 in the veins formed by late diagenetic fluids. The Al_2O_3 contents are well inversely correlated with those of SO_3 (panel b), indicating that Al_2O_3 is highly depleted in the Ca-sulfate veins. The NaO contents are also inversely correlated with those of SO_3 (panel c). The NaO contents are almost zero at the highest SO_3 content (panel c), indicating that NaO is also highly depleted in the veins. On the other hand, the NaO contents are well correlated with those of Al_2O_3 (panel d), showing that NaO is most likely to be contained in the matrix of the mudstones. These results strongly suggest that sodium in the mudstones would have originated from lake water together with the detrital components, such as Al, rather than the post-depositional fluids together with SO_3 .

To clarify this point in the main text, we have added the following sentences in lines 50–56 of the revised manuscript.

“For instance, the presence of sulfate-rich fractures strongly suggests that the post-depositional fluids were enriched in SO_4^{2-} (refs. ^{13,15}). On the other hand, the Yellowknife Bay sediments contain 0.1–0.4% of halite^{11,12}. This salinity most likely originates from the primary pore water trapped upon deposition of the sediments (Fig. 1a, b), rather than being provided by the post-depositional fluids (Fig. 1d). This is because the concentrations of Na in the Yellowknife Bay sediments near the sulfate-rich fractures show both an inverse correlation with SO_3 and a positive correlation with detrital components¹⁵,

such as Al (Supplementary Fig. 1).”

Concerning pH and Eh, we suggest that, in the full-interaction scenario, neutral pH and redox disequilibria of the final pore water would have been caused by mixing of alkaline-reducing primary components and acidic-oxidizing additional components. This view is supported by recent publications on the formation conditions of magnetite (its precursor, green rust) (Tosca et al., 2018 *Nature Geosci.*) and akaganeite (Peretyazhko et al., 2018 *JGR Planets*). More particular, alkaline-reducing primary components are supported by the formation conditions of Fe saponite and magnetite; whereas, acidic-oxidizing additional components are suggested by the formation condition of akaganeite. To clarify this, we have added the above references in the main text and modified the original expression as follows in lines 230–243 of the revised manuscript.

“Since pH conditions for saponite formation on Earth are usually required to be alkaline ($\text{pH} > 9$)²⁴, the primary pore-water pH for the Yellowknife Bay sediments in the early diagenesis is also most likely alkaline. The alkaline pH for the primary pore water is also favored for authigenic precipitation of magnetite found in the Yellowknife Bay sediments³¹. Accordingly, in the full-interaction scenario, the primary alkaline components need to be neutralized in the late stages by post-depositional fluids (Fig. 1d). The previous studies on mineralogy and geochemistry of the Murray and Stimson formations—overlying Yellowknife Bay Formation—show evidence of acidic alterations by fluids enriched in H_2SO_4 after sediment depositions³². The present study supports the occurrence of post-depositional acidic fluids that would have neutralized primary alkaline components in the pores at Yellowknife Bay. This idea would be supported by the presence of akaganeite in the matrix of the Yellowknife Bay sediments¹³. Given that the formation of akaganeite requires relatively low pH (2–8)²⁶, the co-existence of smectite, magnetite, and akaganeite in the sediments can be explained by the alteration of the alkaline matrix of the mudstone by post-depositional acidic fluids, thereby suggesting dynamic changes in water chemistry.”

Regarding inorganic carbon, we suggest low dissolved CO_2 in the final pore water (Table 1). Given that the CO_2 contents in the final pore water are mixtures of the primary and additional components, this requires both of a) low dissolved CO_2 in the additional components (i.e., post-depositional fluids); and b) low CO_2 contribution from the primary components (e.g., the absence of carbonates in the matrix of the mudstones, as discussed in Bristow et al., 2017 *PNAS*). Although we discussed low dissolved CO_2 in post-depositional fluids in the original manuscript, we did not mention low CO_2 contributions from the primary components. Thus, the original statements—such as, low CO_2 in the pore water reflects the post-depositional fluids, and Na contents reflect the paleolake water—seem to be apparently conflicting. To avoid this, we have added the following sentences in lines 190–196 of the revised manuscript;

“The estimated dissolved ΣCO_2 in the pore water would be a mixture of both a supply

of CO₂ from the primary components (e.g., dissolution of carbonate) and CO₂ dissolved in the post-depositional fluids. Thus, an assumption of no CO₂ supply from the primary components (e.g., absence of carbonate)²⁸ provides the upper limit of the pore water's pH. Given the present level of partial pressure of atmospheric CO₂ (P_{CO_2}) on Mars as a conservative lower limit of ancient P_{CO_2} levels²⁹ and dissolution equilibrium between the atmosphere and post-depositional fluids, the pore water's pH would have an upper limit of ~7.2.”

and have modified the sentences in lines 319–322 of the original manuscript as follows in lines 402–406 of the revised manuscript.

“Our results of low levels of dissolved CO₂ in the pore water (Table 1) require both low CO₂ supply from the primary components and low dissolved CO₂ in the post-depositional fluids (see above). The former suggests the absence of carbonate in the matrix of the sediments²⁸. Given that post-depositional fluids would have originated from the surface in the full-interaction scenario, the latter may reflect low P_{CO_2} at the time of the last wetting event at Gale.”

Minor Comments

Comment 6: Gypsum dehydration inside CheMin instrument would make bassanite

Line 454: ‘Sulfate: XRD detected anhydrite (anhydrous CaSO₄) and bassanite bassanite (CaSO₄·0.5H₂O) from both of the drill core samples 11,64. Both of the minerals rarely form under terrestrial surface conditions.’

The origin of these minerals is also discussed in Vaniman et al., 2018 – American Mineralogist 103 (7): 1011-1020. Bassanite is a likely product of gypsum dehydration within the low humidity conditions inside the CheMin instrument.

Response to Comment 6: We have added this information in the revised manuscript.

We appreciate the valuable information. We have added the following description in lines 501–502 of revised Methods.

“Bassanite is considered to be a likely product of gypsum (CaSO₄·2H₂O) dehydration within the low humidity conditions inside the CheMin instrument⁵⁷.”

Comment 7: Saponite formation does not always suggest saline conditions.

Line 110: ‘Additionally, formation of trioctahedral smectite, including saponite, usually occurs in terrestrial alkaline-saline lakes as an alteration product by early

diagenesis. Thus, the abundance of saponite in the Yellowknife Bay sediments is also supportive of saline conditions for Gale lake water.'

Saponite does tend to occur in alkaline-saline lakes on Earth, but in Gale saponite is thought to be the product of largely closed system alteration of reactive mafic minerals like olivine rather than via concentration of ions through evaporation—so may not be a good indicator of salinity.

Response to Comment 7: Agreed. We have softened the expression in the revised manuscript.

Since saponite formation can occur as a result of aqueous alterations of mafic rocks with low water/rock ratios, we have softened the expression as follows in lines 160– 162 of the revised manuscript.

“Thus, the abundance of saponite in the Yellowknife Bay sediments is also supportive of hyposaline conditions for Gale lake water, although saponite formation may also occur through closed-system alterations of mafic minerals²⁵.”

Comment 8: A drop of water table?

Line 65 'There are two possible ways for the liquid pore water to have disappeared from the sediments after the final wetting event.....evaporation and freezing and sublimation.'

How about the pore water drains away as the water table drops?

Response to Comment 8: We consider that the presence of halite would not support the possibility of a drop of water table.

If the pore water drained away as water table drops, the formation of halite after the disappearance of water would not have occurred. Thus, we consider that the scenario of a drop of water table can be ruled out.

Comment 9: Comparison of lake duration with the previous estimate

Line 221 'Our results of water chemistry suggest that Gale would have experienced prolonged (10^4 – 10^6 years) episodes of warm and semiarid climates in total, although these episodes might have been intermittent.'

The manuscript should discuss the constraints on duration of lakes provided by extent and thickness of lacustrine deposits (e.g. Grotzinger et al., 2015 – Science).

Response to Comment 9: Agreed. Our estimate of duration is broadly consistent with the minimum duration proposed by the previous study.

We appreciate the reviewer's constructive comment. We compare of our estimate of a total duration of lakes with those estimated based on the thickness of the Gale sediments and terrestrial accumulation rates of sediments (Grotzinger et al., 2015 Science). Grotzinger et al. (2015) estimate a duration roughly as 10^4 – 10^7 years, which broadly agrees with our estimate. Based on the comment, we have added the following sentence in the revised manuscript in lines 361–363 of the revised manuscript.

“This duration is broadly consistent with the duration of the lacustrine environment (10^4 – 10^7 years) estimated from the thickness of the Gale sediments and typical sediment accumulation rates on Earth¹⁰.”

Responses to comments from Reviewer #3 for ‘Dynamic climate and redox interactions on early Mars inferred from water chemistry at Gale’ by Fukushi, K. et al. (MS# NCOMMS-18-21962)

We are grateful to reviewer #3 for the constructive comments and suggestions. Below, we give our responses in turn following each comment, with the reviewers’ comments being in Arial font and our responses being in Times font.

General Comment

Overall, the modeling approaches applied here have high merit, with the potential to provide useful estimates of lake water composition and duration and Gale crater. Unfortunately, as detailed in the comments below, the authors have so thoroughly misinterpreted the detailed geological context of the rocks that their model is applied to that their estimates of lake water chemical composition are likely to be highly inaccurate. This inaccuracy has follow-on effects on their estimates of lake duration, which rely on estimated lake water chemical compositions. This misinterpretation can be readily corrected through more careful reading of the available literature on the sedimentology of the Bradbury Group. Correcting these misinterpretations will result in significant reworking of the modeling and interpretation; accordingly, I recommend that the manuscript be rejected in its current form but would encourage the authors to resubmit this manuscript elsewhere in an updated form, after taking the issues below into consideration.

Response to General Comment

The major concern raised by the reviewer can be summarized as follows; 1) the need to evaluate the consistency between our assumptions used in geochemical modeling and the hydrogeological context of the Yellowknife Bay Formation; 2) the need to evaluate the assumption that mineral equilibria were achieved between the post-depositional fluids and the matrix of the sediments. Concerning the point 1), the reviewer has kindly provided an overview of the hydrogeological context of the Yellowknife Bay Formation, which is very helpful to revise our manuscript.

As for the point 1), we believe that the assumptions using in our geochemical modeling are basically consistent with the hydrogeological context provided by the reviewer except one point, which is the source of Na in the Yellowknife Bay (Sheepbed) mudstones. The reviewer mentions that Na in the mudstones was provided through the

post-depositional fluids. However, the observational data provided by Curiosity would not support this idea. As shown below (Response to Comment 1-h for details), our new results of re-analyses of Curiosity's ChemCam data (Nachon et al., 2014 JGR Planets) show that the Na contents in the mudstones near Ca-sulfate veins exhibit a clear inverse correlation with those of SO_3 , and a positive correlation with those of detrital components, such as Al (see new Supplementary Fig. 1, also attached in this letter below). This strongly suggests that Na was depleted in the post-depositional fluids and was mainly contained in the matrix of the mudstone (see Response to Comment 1-h for details).

In addition, we would suppose that there may be a small misunderstanding of our results. The reviewer might consider that we insist "hypersaline (high salinity)" lake water for the early Gale lakes, such as high salinity lakes developed in hyperarid, hot desert regions on Earth (e.g., the Dead Sea). However, if so, this is misunderstanding. As shown in Table 1, the estimated salinity is only 0.1–0.2 mol/kg, which fall in the category of hyposaline (weakly-to-moderately saline) water. On Earth, such hyposaline lakes typically develop in semiarid, steppe regions (not hot desert regions). The reviewer mentions that our conclusions of saline lakes are incompatible to the absence of desiccation cracks at Yellowknife Bay. However, these texture characteristics are also not so common in hyposaline lakes in semiarid steppe regions, although they are frequently observed in hypersaline lakes in hyperarid hot desert regions (e.g., Gutiérrez, 2005; new ref. 22).

We consider that this would have been caused by our confusing expression. To improve this point, we have modified the wording of "saline" to "hyposaline or weakly-to-moderately saline" throughout the revised manuscript. Additionally, we have added our results of geological surveys for hyposaline lakes in semiarid steppe regions on Earth (new Supplementary Fig. 2). In this new figure, we suggest that the texture characteristics of hyposaline lakes (e.g., a lack of extensive desiccation cracks) do not contradict with the geological observations of the Yellowknife Bay sediments by Curiosity (see Response to Comment 2-a-2 below for details).

Based on the reviewer's comment, we have added new paragraphs and new figure (new Fig. 1) in order to explain the consistency between our calculation results and the hydrogeological context of the Yellowknife Bay Formation (see Responses to Comments 1 and 2 below for details). We hope that this revision would improve the understanding of our results.

Concerning the point 2), we have largely modified the structure of the manuscript. In the original manuscript, we have discussed the single scenario, in which

the post-depositional fluids fully interacted with the matrix of the mudstone. However, whether the fluids really seeped into the mudstone in the rewetting event is highly uncertain. In the revised manuscript, thereby, we consider two end-member scenarios for the interactions between the fluids and the mudstones: One is the “full-interaction” scenario, in which we assume the fluids interacted with the mudstones as considered originally, and the other is the “no-interaction” scenario, in which the fluids and mudstones do not interacted. To this end, we have newly performed a series of calculations for the no-interaction scenario to estimate the water chemistry of the early lakes using secondary minerals contained in the matrix of the mudstones and have added the new results in a new section of “the no-interaction scenario”. Our results show that the estimated Na concentrations and pH of the paleolake water for the no-interaction scenario agree with those for the full-interaction scenario. Thus, our conclusions (hyposaline lakes, moderate pH, and a total lake duration of 10^4 – 10^6 years) do not change significantly. We believe that this agreement between the two end-member scenarios suggests robustness of our conclusions. (see Responses to Comments 1-g and 2-b).

Major issues with regard to geological context:

Comment 1: An overview of hydrogeological context of Gale lakes

I will begin with a walkthrough of the available constraints on the depositional and diagenetic/burial history of the Sheepbed member, with appropriate references, and then try to help explain why getting this history correct is so important to the application of the techniques used in this paper.

Response to Comment 1: We agree an overview of the hydrogeological context of the Yellowknife Bay Formation provided by the reviewer, except the Na source (see below in Response to Comment 1-h). In response to the reviewer’s comment, we have added an overview of the hydrological context in the revised manuscript.

We greatly appreciate that the reviewer has provided a helpful overview of the hydrogeological context of the early Gale lakes and the Yellowknife Bay (Sheepbed) mudstones. Our assumptions used in the geochemical modeling are basically consistent with the hydrogeological context provided by the reviewer except the source of Na in the mudstone (see Response to Comment 1-h below for details). Since the hydrogeological context of the Gale lakes is critical to the assumptions of our calculations, we consider that an addition of an overview of the hydrological context

would be helpful for readers. Based on the reviewer's overview, we have added the following sentences together with schematic illustrations (new Fig. 1) of the hydrological context in lines 36–46 of the revised manuscript.

“The Yellowknife Bay Formation was deposited by flow deceleration as a river(s) encountered the lake in Gale Crater^{9,10,12} (Fig. 1a). The bulk mineralogy contains significant amounts of secondary non-detrital phases, including smectite and iron oxides¹¹. This suggests that most of chemical reactions that resulted in the generation of phyllosilicates and other secondary minerals took place after deposition, during early diagenesis^{11–14} (Fig. 1b). The Yellowknife Bay sediments were subjected to burial pressure and pervasively fractured after the deposition and early diagenesis^{9,10} (Fig. 1c). During the rewetting events, the late-diagenetic (post-depositional) fluids were introduced into the fractures, which are currently filled with calcium sulfate (CaSO₄); presumably a substantial amount of time after the period of the early lakes^{9,11,15} (Fig. 1d). These rewetting events occurred within Gale even after the formation of Aeolis Mons¹⁶. After the final rewetting event, liquid water in the Yellowknife Bay sediments disappeared (Fig. 1e).”

Below, we provide our response to each comment provided by the reviewer.

Comment 1-a. The Sheepbed member was deposited by flow deceleration as a river(s) encountered a lake in Gale crater. Massive (i.e., poorly-laminated) textures are consistent with rapid rainout of suspended silt- to clay-sized particles in a near-shore (proximal) environment [1,2,3].

Response to Comment 1-a: Agreed.

We agree the view of the hydrogeological setting of the early Gale lakes (as shown in new Fig. 1a).

Comment 1-b. No textures characteristic of saline lakes (e.g., desiccation cracks, rip-up chips), glacial lakes (e.g., dropstones, tillites), or subglacial lakes (e.g., sand mounds, strongly bi-modal sand and mud grain size distributions) were observed in images of the Sheepbed member [1,2,4]. Observed textures and stratigraphic relationships appear consistent with deposition in an open body of dilute water.

Response to Comment 1-b: Agreed, but there may be a small misunderstanding.

We agree that the reviewer's overview on the geological evidence at Gale, but we are afraid that there may be a small misunderstanding. The reviewer might consider that we propose the presence of “hypersaline (high salinity)” lakes at Gale, which typically develop in hyperarid, hot desert regions on Earth (e.g., the Dead Sea).

However, the estimated Na concentrations by the present study is only 0.1–0.2 mol/kg and fall in the category of “hyposaline (weakly-to-moderately saline)”. Hyposaline lakes often develop in semiarid, steppe climates on Earth. We consider that this misunderstanding would have caused by our unclear expression of “salinity” in the original manuscript. To improve this point, we have changed the wording of “saline” to “hyposaline or weakly-to-moderately saline” to emphasize the mild salinity for the early Gale lakes throughout the manuscript.

In addition, we have performed hydrogeological surveys of terrestrial hyposaline lakes in south Mongolia and central India for several years. The reviewer mentions that a lack of texture characteristic of saline lakes, such as desiccation cracks and rip-up chips, at the Sheepbed mudstones suggests no saline lakes at the time of deposition. These texture characteristics are common near hypersaline lakes in hot desert regions (e.g., Gutiérrez 2005 Climatic Geomorphology; new ref. 22), but these are not always extensive near hyposaline lakes developed in steppes regions, as shown in our results of geological surveys (see new Supplementary Fig. 2). Our new Supplementary Fig. 2 suggests that a lack of these features at the Sheepbed mudstone do not always contradict with hyposaline lake water for the early Gale lakes. To explain this point, we have added new Supplementary Fig. 2 in the revised Supplementary Information.

Comment 1-c. The bulk chemical composition of the Sheepbed member mudstones indicate that the sources of the silt- and clay-sized particles experienced a low degree of chemical weathering, consistent with the hypothesis that the catchment region for rivers flowing into the lake in Gale crater experienced cold climate conditions at the time of Sheepbed member deposition [3,5].

Response to Comment 1-c: Agreed.

We agree with the reviewer’s description of the degree of chemical weathering, which is also consistent with our view of semiarid climate at the time of deposition.

Comment 1-d. The observation that the bulk chemical composition of the rock approximates the composition of unaltered basalt, while the bulk mineralogy contains ~25-55% secondary (non-detrital) phases, has been used to argue that most of the chemical reactions that resulted in the generation of phyllosilicates

and other secondary minerals and mineraloids took place after deposition, during early diagenesis [3,5,6,7].

Response to Comment 1-d: Agreed.

The reviewer's comment is consistent with our assumptions that saponite was formed upon early diagenesis in the matrix of the Sheepbed mudstone. This is shown in our new Fig. 1b.

Comment 1-e. During early diagenesis a variety of textural features were produced, including: subaqueous cracks filled with isopachous cements, interpreted as synaeresis cracks [1,8], and dark concretions and voids with cemented rims [1,9]. These features are interpreted as the products of mineralization during early diagenesis, which potentially resulted in gas production in the sediment [1,3,7,8,9,10].

Response to Comment 1-e: Agreed.

Although the above results are not mentioned in our manuscript, we agree that the mudstones were cracked prior to the intrusion of post-depositional fluids.

Comment 1-f. During burial, likely after the generation of features described in e. (above), the Sheepbed member sediment was subjected to enough burial pressure to produce a sedimentary dike composed of fluidized Sheepbed member sediment, preserved as a feature informally named "the snake" [1].

Response to Comment 1-f: Agreed.

We agree this point. This is shown in our new Fig. 1c.

Comment 1-g. During later diagenesis (i.e., after the events described in a-f, and after the sediment had been lithified), the Sheepbed member mudstones were pervasively fractured and those fractures were filled with Ca-sulfate. These Ca-sulfate filled fractures cross cut the mudstone and the early diagenetic features described in e. (above) [1]. Based on measurements of fracture orientation and morphology, the Ca-sulfate filled fractures in the Sheepbed

mudstones are interpreted to have formed by hydrofracturing at a minimum burial depth of 1.2 km [11]. Notably, the mineralizing fluid that filled these fractures with Ca-sulfate apparently did not penetrate the host mudrock, as the Sheepbed member mudstones exhibit among the lowest total sulfur contents of any rock ever analyzed on Mars [1,5].

Response to Comment 1-g: Partly agreed. The recent detailed analyses of Curiosity's APXS data suggest the presence of small amounts of Ca-sulfate in the matrix of the Sheepbed mudstones. This suggests the possibility of seepage of the sulfate-rich post-depositional fluids into the mudstone matrix. To investigate a full range of possibility, we discuss two end-member scenarios for the interactions between the fluids and the mudstone, "full interactions" and "no interactions", in the revised manuscript.

We agree that the post-depositional fluids entered into the fractures, as shown in our new Fig. 1d. The reviewer considers that the sulfate-rich post-depositional fluids did not interact with the matrix of the mudstone; however, the recent results of detailed analyses of Curiosity's APXS data suggest the possibility of interactions between the fluids and the matrix of the mudstones (Schmidt et al., 2018 JGR Planets). Schmidt et al. (2018) show that even the mudstones without visual Ca-sulfate veins also contain small but certain amounts of CaSO₄. They carefully remove the contributions of sulfate from dust and reveal that the mudstone would contain 1.4 wt.% of SO₃ in average, ranging 0–2.8 wt.%. These results suggest the possibility of seepage of the sulfate-rich fluids into the matrix of the mudstone. Given the penetration depth of alpha particles (< 0.1 mm), SO₃ contributions from hidden veins beneath the surface would be less likely. Thus, the occurrence of interactions between the post-depositional fluids and the matrix of the mudstones cannot be ruled out at this stage.

Using relatively low permeability of dried marine clays as terrestrial analog values for the permeability of the Yellowknife Bay mudstones (10^{-7} – 10^{-10} cm/s: Davis 1969; new ref. 17) and typical distance of Ca-sulfate fractures in the Yellowknife Bay sediments (~1 cm), the post-depositional fluids can seep into the matrix of the sediments if the fluids existed for the order of 1–100 years or longer. The rewetting with this timescale can have happened upon the proposed temporal warming events in post-Noachian periods (e.g., Kite et al., 2017 Nature Geosci.; Palucis et al., 2016 JGR Planets and literatures in our reference list).

Having said that, given the limitation of the observational data, it is difficult to conclude whether the post-depositional fluids really interacted with the mudstones.

Thus, we consider two end-member scenarios in the revised manuscript: One is the “full-interaction” scenario, in which the fluids interacted with the mudstones as considered originally, and the other is the “no-interaction” scenario, in which the fluids did not interact with the mudstone. To this end, we have newly performed a series of calculations to estimate the water chemistry of the paleolake water based on the minerals contained in the matrix of the Sheepbed mudstones. Our new results for the no-interaction scenario show that the estimated Na concentrations of the lake water show agreements with those for the full-interaction scenario. The estimated pH for the no-interaction scenario is also circumneutral (pH 7–8). Thus, our conclusions (hyposaline lakes, moderate pH, and a total lake duration of 10^4 – 10^6 years) do not change significantly.

To clarify the need to consider the no-interaction scenario, we have added the following sentences in lines 60–75 of the revised manuscript.

“Here we consider two end-member scenarios concerning the interactions between the primary and additional components in the sediments: In one, that the post-depositional fluids fully interact with the sediments upon the intrusion (the full-interaction scenario), while in the other we assume no chemical interactions between the post-depositional fluids and the sediments (the no-interaction scenario; see also Fig. 1f). The duration of the rewetting event determines which scenario was the case. Using both terrestrial analog permeability of dried marine clays (10^{-7} – 10^{-10} cm/s)¹⁷ and the typical distance of the Ca-sulfate fractures in the Yellowknife Bay sediments (~ 1 cm)¹³, post-depositional fluids can chemically interact with the matrix of the Yellowknife Bay sediments if the fluids existed for the order of 1 – 10^2 years or longer. In the full-interaction scenario, we consider that the primary components (such as smectite) in the matrix of the sediments equilibrate with the additional components (such as calcium sulfate) in pore water (Fig. 1d). Thereby, the estimated water chemistry would reflect mixtures of the primary and additional components. In the no-interaction scenario, we assume that the observed secondary minerals in the matrix of the sediments are not influenced by the post-depositional fluids (Fig. 1d). In the latter scenario, we estimate the chemical composition of the lake water only using the secondary minerals that are considered to be contained in the matrix of the sediments.”

To describe the results of the no-interaction scenario, we have added a new section of “no interaction scenario” in lines 270–315 of the revised manuscript. The calculation methodology of the no-interaction scenario is summarized in lines 430–509 and 639–650 of revised Methods.

Comment 1-h. While it is difficult to draw strong conclusions based on only two samples, imaging of the drill hole walls after the collection of the John Klein (JK) and Cumberland (CB) samples revealed that the JK sample contained abundant

fracture-filling Ca-sulfate (formed by the process described in g., above) as compared to CB. Not surprisingly, the JK sample contains more Ca-sulfate by XRD than CB, and also more halite [3, 12], suggesting that the halite detected in JK and CB is associated with the Ca-sulfate vein fills rather than the impermeable host mudrock. The presence of halite in Ca-sulfate veins is also suggested by LIBS spectra collected from them [13].

Response to Comment 1-h: More detailed analyses of Curiosity's data strongly suggest that Na would have been depleted in the post-depositional fluids. To explain this, we have added new Supplementary Fig. 1 in Supplementary Information.

The reviewer considers that Na in the Sheepbed mudstones was provided from the post-depositional fluids; however, detailed analyses of Curiosity's data would not support this idea. Our results of re-analysis of the data shown in Nachon et al. (2014) indicate that the Na concentrations in the Yellowknife Bay sediments near Ca-sulfate veins show a clear inverse correlation with those of SO₃ and Ca (see new Supplementary Fig. 1; also attached to this letter below). Instead, the Na concentrations show a positive correlation with those of the detrital components, such as Al (new Supplementary Fig. 1). These results strongly suggest that Na was not provided by the post-depositional fluids but is most likely to have been contained originally in the mudstones. To clarify this point, we have added the following sentences in lines 51–56 of the revised manuscript and have added new Supplementary Fig. 1 in revised Supplementary Information.

“On the other hand, the Yellowknife Bay sediments contain 0.1–0.4% of halite^{11,12}. This salinity most likely to originates from the primary pore water trapped upon deposition of the sediments (Fig. 1a, b), rather than being provided by the post-depositional fluids (Fig. 1d). This is because the concentrations of Na in the Yellowknife Bay sediments near the sulfate-rich fractures show both an inverse correlation with SO₃ and a positive correlation with detrital components¹⁵, such as Al (Supplementary Fig. 1).”

Supplementary Fig. 1 Plots of CaO vs. SO₃ (a), Al₂O₃ vs. SO₃ (b), NaO vs. SO₃ (c) and NaO vs Al₂O₃ (d) based on the LIBS measurements for the Yellowknife Bay mudstones (Sheepbed member) with Ca-sulfate veins¹⁵. The straight line in panel (a) indicates the stoichiometry of calcium sulfates (CaSO₄). The agreements of the CaO and SO₃ with the stoichiometry suggests that calcium and sulfur in the bulk mudstone predominantly are largely derived from CaSO₄ in the veins formed by late diagenetic fluids. The Al₂O₃ contents are well inversely correlated with those of SO₃ (panel b), indicating that Al₂O₃ is highly depleted in the Ca-sulfate veins. The NaO contents are also inversely correlated with those of SO₃ (panel c). The NaO contents are almost zero at the highest SO₃ content (panel c), indicating that NaO is also highly depleted in the veins. On the other hand, the NaO contents are well correlated with those of Al₂O₃ (panel d), showing that NaO is most likely to be contained in the matrix of the mudstones. These results strongly suggest that sodium in the mudstones would have originated from lake water together with the detrital components, such as Al, rather than the post-depositional fluids together with SO₃.

Comment 1-i. Finally, the bulk chemical composition of the CB sample indicates higher total Cl abundance than that of the JK sample [14], but the Cl in both samples is largely attributed to the presence of ~wt. % abundances of oxychlorine species, which are hypothesized to have been an important component of the lake water at the time of Sheepbed member deposition [14]. The mineralogical host of these oxychlorine species, which are ubiquitous in rock samples analyzed by the SAM and CheMin instruments, remains one of the

outstanding open questions from the Curiosity mission to Gale crater.

Response to Comment 1-i: Agreed. Even considering perchlorates in lake water, our conclusions of estimated water chemistry do not change.

We agree that perchlorate (ClO_4^-) might have been present in lake water, although the quantitative ratio of $\{\text{Cl}^-\}$ to $\{\text{ClO}_4^-\}$ is poorly constrained. Both Cl^- and ClO_4^- are monovalent anions, which do not take part in the cation exchange reactions. In our top-down approach to constrain major components, the chemical behavior of ClO_4^- is indifferent from that of Cl^- . Thus, the concentrations of Cl^- obtained from the present study actually represents the concentrations of total chloride ($= \{\text{Cl}^-\} + \{\text{ClO}_4^-\}$), and our major conclusions of the estimated ion concentrations, pH, and Eh do not change even considering the presence of ClO_4^- .

However, we still consider that $\{\text{ClO}_4^-\}$ would have been lower than $\{\text{Cl}^-\}$. This is the case because NaCl salt (halite) was detected from the XRD from the mudstone, but NaClO_4 salt not. This may be reasonable given the fact that the solubility of NaCl salt is significantly lower than NaClO_4 . Nevertheless, the estimated concentrations of total chlorine (i.e., $= \{\text{Cl}^-\} + \{\text{ClO}_4^-\}$) are always slightly higher than Na^+ . The fact that most of Na^+ reacted predominantly with Cl^- to form halite suggests that the concentration of ClO_4^- must be lower than that of Cl^- . In the present study, we thus consider that only Cl^- for total chloride concentration. Based on the reviewer's comment, we have added the following sentences in lines 536–542 of the revised manuscript.

“The perchlorate (ClO_4^-) has been detected in the Yellowknife Bay sediments⁶³. Both Cl^- and ClO_4^- are monovalent anions, which do not take part in the cation exchange reactions. According to the present approach to constrain major components, the chemical behavior of ClO_4^- is not different from that of Cl^- . Therefore, the concentrations of Cl^- obtained from the present study actually represents the concentrations of total chloride ($\{\text{Cl}^-\} + \{\text{ClO}_4^-\}$), and our conclusions of concentrations of major components, pH, and Eh do not change even when considering ClO_4^- .”

Comment 2. Taken together, the constraints described above reveal three potentially significant flaws in the geochemical modeling described by Fukushi and co-workers.

Comment 2-a-1. a. The calculation of idealized fluid compositions (Table 1 in manuscript) from clay interlayer spacing relies on the constraint: $2\{\text{Na}\} > \{\text{Mg}\}$ (supplemental information file lines 33-35), but this constraint assumes that halite was formed from pore fluids that were in contact with the clay minerals in

the JK and CB drill samples. Based on the sedimentological constraints, it is entirely possible that halite is part of the Ca-sulfate fracture filling mineral assemblage (h., above), and it is likely that the Sheepbed member mudstones were impermeable to the mineralizing fluids that filled these fractures (g., above). Therefore, the high concentrations of dissolved Na and Cl required to form halite may have been components of a fluid that were never “in communication” with the phyllosilicate minerals in the JK and CB samples.

Response to Comment 2-a-1: As discussed above in Response to Comment 1-h, we consider that halite is most likely be associated with the matrix of the mudstones. Concerning the interactions between the fluids and the matrix of the mudstones, we have considered two end-member scenarios, the full- and no-interaction scenarios, in the revised manuscript.

As discussed above in Response to Comment 1-h, we would argue against the idea that Na was derived from post-depositional fluids. Detailed analyses of Curiosity’s data shown in Nachon et al. (2014) would support our interpretation that Na was mainly contained in the matrix of the mudstones (see Response to Comment 1-h for details). We have added the explanations of this evidence in lines 51–56 of the revised manuscript and have added new Supplementary Fig. 1 in revised Supplementary Information, as shown in Response to Comment 1-h above.

Concerning the possibility of seepage of the post-depositional fluids into the matrix of the mudstone, we admit the importance to discuss the alternative scenario of “no interactions” between the fluids and the matrix of the mudstone. To this end, we have newly performed calculations to estimate the water chemistry (no-interaction scenario), as suggested by the reviewer. In the revised manuscript, we have largely modified the structure of the manuscript by adding a new section of “no-interaction scenario” in lines 270–315, as mentioned in Response to Comment 1-g for details.

Although we cannot conclude whether the matrix of the mudstones and the post-depositional fluids were in equilibrium, we would still believe that the full-interaction scenario would be favorable for the following reasons. a) The detailed analyses for Curiosity’s APXS data show that small amounts of CaSO_4 (1.4 wt.% SO_3 in average, ranging 0–2.8 wt.%) may exist in the matrix of the mudstone without visual Ca-sulfate veins (Schmidt et al., 2018 JGR Planets). The existence of CaSO_4 in the matrix of the mudstones suggests that the sulfate-rich post-depositional fluids could have diffused into the surrounding mudstone in the rewetting event, as discussed in Response to Comment 1-g above.

In addition, as discussed in the revised main text in lines 300–310 of the new section of “the no-interaction scenario”, the no-interaction scenario requires extremely rapid transport and high accumulation rates of sediments in order to explain co-existence of oxidizing akaganeite and reducing pyrrhotite in the matrix of the mudstone. This is because reducing pyrrhotite is readily oxidized under the oxidizing conditions that allow akaganeite formation. This, in turn, suggests a short duration of the Gale lakes given the thickness of the Gale sediments, which might be incompatible with the development of hyposaline lakes since a substantial duration would be required to accumulate the estimated salinity.

Although we cannot conclude whether the fluids interacted with the mudstone given the limited observations, we consider that one of the most straightforward interpretations would be the full-interaction scenario; i.e., a seepage of acidic-oxidizing post-depositional fluids into the mudstones. To discuss the above points, we also have added a new section of “The full-interaction scenario vs the no-interaction scenario” in lines 316–338 of the revised manuscript.

Comment 2-a-2: Furthermore, given the absence of sedimentological evidence for evaporation or freezing as a significant control on rock textures during Sheepbed member deposition (b., above), it seems unlikely that the “high salinities” required to precipitate halite from solution were present in the lake during the deposition of the mudstones.

Response to Comment 2-a-2: **There may be a small misunderstanding. We indeed do not insist high salinity for the paleolake water, but suggest hyposaline water. The texture characteristics for terrestrial hyposaline lakes do not always contradict with those of the Sheepbed mudstones. We have clarified this point in the revised manuscript and have added new Supplementary Fig. 2.**

As mentioned above in Response to Comment 1-b, we admit that the wording of “saline” in the original manuscript was misleading and unclear. In fact, we do not consider that salinity was high enough to produce halite during deposition. We consider that halite was formed after the freezing of the pore water through sublimation, as described in the manuscript. The estimated Na concentrations are only 0.1–0.2 mol/kg, which falls in the category of “hyposaline (weakly-to-moderately saline)” (see Response to Comment 1-b). Our geological surveys for terrestrial hyposaline lakes show no extensive geological evidence for evaporation, such as desiccation cracks, for

hyposaline lakes developed at semiarid steppe regions on Earth (new Supplementary Fig. 2) (see Response to Comment 1-b).

To discuss the consistency between our results and the hydrogeological context of the Sheepbed mudstones, we have added new Supplementary Fig. 2 in revised Supplementary Information. Additionally, to avoid this potential misunderstanding, we have used “hyposaline lake or weakly-to-moderately saline” throughout the revised manuscript, instead of “saline”.

Comment 2-a-3: At the very least, a more valid approach, one which honors the in-situ sedimentological and geochemical results, would be to present a range of possible solution chemistries that satisfy the constraints from phyllosilicate crystal chemistry under the assumption that halite was either present or absent during clay mineral precipitation and equilibration with its co-existing fluid.

Response to Comment 2-a-3: Although we cannot constrain any water chemistry without considering halite, we agree that a more valid approach would be to examine a full range of possibility of our assumptions. With this regard, we have discussed the “no-interaction” scenario in the revised manuscript.

We appreciate the reviewer’s suggestion. Unfortunately, it is theoretically impossible to constrain the water chemistry without considering halite (i.e., without an assumption that sodium and chloride concentrations are higher than those of other components). However, we do agree that an investigation of a full range of possibility is a more robust approach for the present study.

As discussed above in Response to Comment 1-g, we have considered the “no-interaction” scenario, in which fluids and the mudstone matrix were not in equilibrium, as an alternative case for seepage of the fluids into the mudstones (corresponding to lines 270–315 of the revised manuscript). We hope that this large re-structuring in the revised manuscript would improve our manuscript.

Comment 2-b: b. Figure 2 and the supporting text have great potential to reveal important constraints on the pH and DIC concentration of fluids from which Ca-sulfate fracture filling minerals were formed. This issue is actually a matter of some debate within the MSL Curiosity science team [15-17]. Unfortunately, Fukushi and co-workers use the relationships in Figure 2 in an attempt to

provide constraints on the nature of fluids in the surface environment at Gale Crater. Given the deep burial conditions under which the fractures that host Ca-sulfate were formed, and the fact that the host rocks were impermeable to these mineralizing fluids (g., above), it is highly unlikely that the fluids from which Ca-sulfate formed had anything to do with surface environmental conditions. The relationships on Figure 2 should therefore not be used to place constraints on paleo- $p\text{CO}_2$ conditions in the atmosphere, or the nature of fluids that equilibrated with phyllosilicates. Instead, what is potentially revealed by the relationships shown in Figure 2 is that the fluids from which Ca-sulfate precipitated were low in total DIC, and circum-neutral in pH, both of which are expected consequences of rock buffering to remove DIC and acidity derived from the atmosphere during deep burial. Cast in this new light, Fukushi and co-workers may have provided very useful constraints on the nature of deep subsurface diagenetic fluids in Gale crater! An additional useful modeling exercise would be to evaluate the gypsum-calcite relationships depicted on Figure 2 as a function of temperature, which is expected to be higher in the deep subsurface.

Response to Comment 2-b: The estimated high Eh required to form akaganeite cannot be achieved without photochemically-produced oxidants. Based on some reasons shown below, we consider that the high-Eh oxidants were provided via fluid intrusion from the surface in the full-interaction scenario. In the revised manuscript, we have added discussion in the no-interaction scenario that the high-Eh oxidants were present at the time of deposition of the Sheepbed mudstones.

We consider that the estimated dissolved CO_2 would represent atmospheric P_{CO_2} at the time of intrusion of post-depositional fluids because the fluids would have originated from the surface. This is the case because high Eh > 0.5 V at pH ~ 7 is required to form akaganeite (see new Fig.4 or original Fig. 3). The measured correlation between Fe (or Mg) and Cl of the Sheepbed mudstones suggests that akaganeite is associated with smectite in its matrix (McLennan et al., 2014 Science). The recent experimental study also supports the idea that highly-oxidizing conditions are needed to form akaganeite (Peretyazhko et al. 2018 JGR Planets). On Mars, potential oxidants capable of producing sufficient Eh (i.e., Eh > 0.5 V) are O_2 , O_3 , ClO_4^- , and NO_3^- , all of which are formed via photochemical reactions. Sulfate is an insufficient oxidant. Deep subsurface fluids without atmospheric oxidants cannot explain the formation of akaganeite. These results indicate that high-Eh atmospheric oxidants are needed to have

been supplied to the Sheepbed mudstones.

There are two possibilities for the supply of atmospheric oxidants to the Sheepbed mudstones: One is that the lake water was highly oxidizing at the time of deposition of the Sheepbed mudstone (as suggested by the reviewer in Comment 10 below), and the other is intrusion of the acidic-oxidizing post-depositional fluids. We have discussed the latter possibility in the original manuscript for the following reasons: 1) authigenic Fe-bearing minerals (e.g., magnetite and Fe²⁺-smectite) found in the mudstones at Gale suggest alkaline-reducing aqueous environments at the time of deposition (Tosca et al., 2018 *Nature Geosci.*). 2) Findings of Mn oxides within vein fractures of sediments at Gale and Endeavor Craters support the presence of highly-oxidizing subsurface fluids (Lanza et al., 2016 *GRL*; Anderson et al., 2016 *Am. Mineral.*). 3) Seepage of post-depositional fluids into the mudstones can occur if the rewetting event persisted for > 1–100 years, which could have achieved by proposed warming mechanisms in post-Noachian periods (see Response to Comment 1-g).

Having said that, we would admit an importance of investigation of another possibility. In fact, the new no-interaction scenario requires the former possibility of the highly-oxidizing lake water to explain the coexistence of oxidizing akaganeite and reducing pyrrhotite and Fe saponite in the mudstones. The coexistence of akaganeite and pyrrhotite can have happened if transport and burial of detrital pyrrhotite occurred in a short time. This is because oxidation of silt-sized pyrrhotite readily proceed under the oxidizing conditions that allow formation of akaganeite (Janzen et al., 2000 *GCA*). We have estimated this timescale of pyrrhotite oxidation as a few hundred years (see Response to Comment 10 below for details). This timescale is shorter than typical transport time of detrital components for short rivers (e.g., $(1-3) \times 10^3$ years for rivers with length < 100 km: Vignier et al., 2006 *EPSL*), suggesting extremely rapid transport and high accumulation rates of sediments when the Yellowknife Bay sediments were deposited.

Based on the reviewer's comment, we have discussed the possibility of the former case of highly-oxidizing lacustrine environments in the new section of "no-interaction scenario" in lines 301–310 of the revised manuscript.

"Under highly-oxidizing conditions that allow akaganeite formation, a silt-sized pyrrhotite grain would be oxidized in a few hundred years or less (see Sec. S5 of Supplementary Information). Thus, if transport and burial of detrital components occurred within a few hundred years, authigenic akaganeite could coexist with detrital pyrrhotite. However, this timescale for oxidation of pyrrhotite is shorter than the typical transport time of detrital components in short rivers (length of 40–100 km) on a terrestrial basaltic terrain (e.g., $(1-3) \times 10^3$ years)³⁸. Early diagenesis within the lakes would also have taken a long period of time. Hence, the no-interaction scenario would require extremely rapid transport and high accumulation rates of sediments at the time

of deposition of the Yellowknife Bay sediments. The high accumulation rates, in turn, imply a short duration of the early lakes given the thickness of sediments within Gale Crater.”

Comment 2-c: c. Finally, as consequence of the previous comment (Comment 2-b. immediately above) the acid conditions called on to neutralize the alkaline fluids from which clay minerals may have precipitated in order to form Ca-sulfate are unnecessary (manuscript lines 169-176). Instead, the phyllosilicate and sulfate minerals can be readily interpreted in the context of two different generations of fluid: an alkaline pH fluid that formed phyllosilicates during early diagenesis, and another circum-neutral pH fluid that formed Ca-sulfate after lithification during deep burial. On this point, it is worth noting that there is not a single mineral phase in the JK or CB samples that requires low-pH conditions to form [6,12], and so the link to the “acid diagenesis” described by some authors [e.g., 18] for lithologies higher in the Gale stratigraphy is tenuous.

Response to Comment 2-c: **The pH conditions required to form akaganeite and magnetite/Fe-saponite would be incompatible each other. We consider that a mixing of multiple components of water would be one plausible way to explain the observations in the original manuscript, but we also have discussed the alternative possibility in the “no-interaction” scenario in the revised manuscript.**

As mentioned above in Response to Comment 2-b, Peretyazhko et al. (2018) recently examine the formation condition of akaganeite by laboratory experiments. They find that pH 2–8 (and highly oxidizing condition) is required responsible for akaganeite formation (Peretyazhko et al., 2018 JGR Planets). On the other hand, the occurrence of authigenic Fe-minerals (magnetite and its precursor, green rust) in the Sheepbed mudstones suggest alkaline pH (and reducing) conditions (pH > 8) for their generations (Tosca et al., 2018 Nature Geosci.). Based on these results, we consider that multiple water sources with different pH would be a straightforward way to explain the observations in the original manuscript.

Having said that, in the revised manuscript, we have examined the water chemistry for the alternative no-interaction scenario (see Response to Comment 2-b). In the no interaction scenario, we discuss the possibility of explanation of the water chemistry without any interactions between the post-depositional fluids and the matrix of the mudstones. To preserve the redox disequilibria of Fe-bearing minerals (i.e., akaganeite and pyrrhotite) without mixing of the post-depositional fluids, the

no-interaction scenario would require preservation of detrital pyrrhotite due to extremely rapid transport and high accumulation of sediments. This may imply a short duration of the early lakes (see Responses to Comments 10 for details). As mentioned above in Response to Comment 2-b, we have added the discussion in lines 300–310 of the revised manuscript.

Other comments

Comment 3: Traceability of methodology

The way that the methods and supplemental information file are broken up to make this paper suitable for the Nature Communications format make the body text of this manuscript particularly difficult to read. The arguments in the body text require the reader to read the methods and supplemental before even beginning to read the body text. I think that this submission would be better suited to a longer format journal where the supplementary information can be folded into a longer methods section and incorporated into the article, making for a much more logical flow of ideas.

Response to Comment 3: To improve the traceability of methodology, we have referred the corresponding chapter of Supplementary Information in the main text of the revised manuscript.

We believe that the results of our study are important for a wide range of the research fields, including planetary science, geochemistry, mineralogy, climatology and astrobiology. We thus still consider that *Nature Communication* is the most suitable journal for publication of this study. However, we would admit that readers might have a difficult to trace our methodology in Supplementary Information. With this regard, we have improved the traceability of our methodology by referring the corresponding section number of Supplementary Information, such as Sec. S1 of Supplementary Information, in the revised manuscript.

Comment 4: Reference needed

Sentence starting on line 70 of the manuscript should have a reference.

Response to Comment 4: Agreed. We have added the corresponding reference.

We have cited a textbook (Drever, 1997, ref. 18) of aqueous geochemistry

illustrating the controls of water chemistry during the evaporation of natural water for the reference of this sentence in the revised manuscript.

Comment 5: Where does freezing point depression come from?

Line 75–76 of the manuscript: where does this constraint on freezing point depression come from?

Response to Comment 5: The temperature of freezing point depression is calculated based on Blagden’s Law.

The freezing point depression (ΔT) can be expressed as following equation (i.e., Blagden’s Law):

$$\Delta T = K_f \sum_i m_i$$

where $K_f = 1.85$ (K·kg/mol) and m_i is molal concentration of i th species. The $\sum_i m_i$ of the minimum and maximum estimates for the pore water are 0.33 and 1.2, respectively. Thus, the freezing point depression (ΔT) are calculated as 0.60 °C and 2.2 °C, respectively. This information has been added in lines 124–125 of the revised manuscript as follows.

“Freezing point depression for this hyposaline water is only 1–2°C based on Blagden’s Law”

Comment 6: Na⁺ concentration in veins

Line 97 of the manuscript: what are the Na⁺ concentrations in calcium sulfate veins lower than? This sentence needs to be rephrased for clarity. Also, see reference [13, below].

Response to Comment 6: What we meant here is “Na⁺ concentrations in veins are lower than the matrix of the mudstone”. We have changed the expression in the revised manuscript.

As shown in our new Supplementary Fig. 1, Na is highly likely to have been depleted in veins, compared with the matrix of the mudstones. We have changed the expression of this sentence as follows in lines 137–138 of the revised manuscript.

“the post-depositional fluids at Gale are highly unlikely to be the major source of Na⁺ (Supplementary Fig. 1).”

Comment 7: Confusion on discussing conditions of the pore water

Lines 6-8, line 101 of the manuscript, and elsewhere in the text: there seems to be some confusion in the text about whether you are discussing the conditions in the pore water that the clays last equilibrated with, and whether those conditions are reflective of the lake water, the pore water, or the pore water at the moment before it froze or evaporated, etc. I recognize that the paper is trying to say something about chemical conditions in the lake and the ancient climate in Gale, but the authors occasionally toss in these caveats about the pore fluids representing a snapshot of the last fluid the phyllosilicates were in contact with, which may not have much to do with broader conditions in the lake/climate system when the sediments were first deposited in it.

Response to Comment 7: Agreed. To clarify the considered conditions, we have added a new paragraph prior to the results section in the revised manuscript.

The top-down approach cannot trace the evolution of water chemistry, but it can distinguish whether the snapshot of the water chemistry can be generated by a single process or would require a combination of multiple distinct processes. In the full-interaction scenario, we consider salinity, pH, Eh, and inorganic carbon contents are determined by mixtures of the primary components originating from the paleolake water and the additional components originating from the post-depositional fluids. To avoid confusion, we have emphasized this point repeatedly in the revised manuscript. For instance, we have added the following sentences in lines 68–71 of the revised manuscript;

“In the full-interaction scenario, we consider that the primary components (such as smectite) in the matrix of the sediments equilibrate with the additional components (such as calcium sulfate) in pore water (Fig. 1d). Thereby, the estimated water chemistry would reflect mixtures of the primary and additional components.”

and added the following sentence in lines 133–136 of the revised manuscript;

“Although the dissolved species in Table 1 are mixtures of the primary and additional components within the sediments, we believe that the predominant dissolved species of Na^+ and Cl^- in the pore water would have largely originated from the primary component; namely, lake water trapped within the Yellowknife Bay sediments (Fig. 1a, b).”

and added the following sentence in lines 172–174 of the revised manuscript.

“Hence, in the full-interaction scenario, the pore-water pH for the Yellowknife Bay sediments is determined by both of the primary components from lake water in the early diagenesis and the additional components from the post-depositional fluids in

the late diagenesis.”

As for Na contents, the post-depositional fluids are highly likely to have been depleted in Na based on the re-analyses of Curiosity’s data (see Supplementary Fig. 1, data from Nachon et al., 2014 JGR Planets). Supplementary Fig. 1 strongly suggests that the post-depositional fluids cannot be the major source of Na. Thus, we interpret that Na in the final pore water largely reflects the lake water.

Concerning pH and Eh, we originally suggest that neutral pH and redox disequilibrium of the final pore-water chemistry would have been caused by mixing of alkaline-reducing primary components and acidic-oxidizing additional components in the full-interaction scenario. This view may be supported by recent publications on the formation conditions of magnetite (and its precursor, green rust) (Tosca et al., 2018 Nature Geosci.) and akaganeite (Peretyazhko et al., 2018 JGR Planets). Namely, alkaline-reducing primary components are supported by the formation conditions of magnetite and ferrous saponite; whereas, acidic-oxidizing additional components are suggested by the formation condition of akaganeite. To clarify this explanation in the full-interaction scenario, we have modified the original expression by adding the above references in lines 230–243 of the revised manuscript, as follows.

“Since pH conditions for saponite formation on Earth are usually required to be alkaline ($\text{pH} > 9$)²⁴, the primary pore-water pH for the Yellowknife Bay sediments in the early diagenesis is also most likely alkaline. The alkaline pH for the primary pore water is also favored for authigenic precipitation of magnetite found in the Yellowknife Bay sediments³¹. Accordingly, in the full-interaction scenario, the primary alkaline components need to be neutralized in the late stages by post-depositional fluids (Fig. 1d). The previous studies on mineralogy and geochemistry of the Murray and Stimson formations—overlying Yellowknife Bay Formation—show evidence of acidic alterations by fluids enriched in H_2SO_4 after sediment depositions³². The present study supports the occurrence of post-depositional acidic fluids that would have neutralized primary alkaline components in the pores at Yellowknife Bay. This idea would be supported by the presence of akaganeite in the matrix of the Yellowknife Bay sediments¹³. Given that the formation of akaganeite requires relatively low pH (2–8)²⁶, the co-existence of smectite, magnetite, and akaganeite in the sediments can be explained by the alteration of the alkaline matrix of the mudstone by post-depositional acidic fluids, thereby suggesting dynamic changes in water chemistry.”

Regarding the inorganic carbon content, we suggest low dissolved CO_2 in the final pore water (Table 1). Since the inorganic carbon content is also considered as a result of mixtures of the primary and additional components, this requires both of low dissolved CO_2 in the additional components (i.e., the post-depositional fluids) and low CO_2 contribution from the primary components (e.g., the absence of carbonates in the matrix of the mudstones, as discussed in Bristow et al., 2017 PNAS). Although we discussed low dissolved CO_2 in the post-depositional fluids in the original manuscript,

we did not mention low CO₂ contributions from the primary components. We would consider that a lack of the explanation of the latter constraint may be confusing. To explain this, we have added the following sentences in lines 190–196 of the revised manuscript;

“The estimated dissolved ΣCO_2 in the pore water would be a mixture of both a supply of CO₂ from the primary components (e.g., dissolution of carbonate) and CO₂ dissolved in the post-depositional fluids. Thus, an assumption of no CO₂ supply from the primary components (e.g., absence of carbonate)²⁸ provides the upper limit of the pore water’s pH. Given the present level of partial pressure of atmospheric CO₂ (P_{CO_2}) on Mars as a conservative lower limit of ancient P_{CO_2} levels²⁹ and dissolution equilibrium between the atmosphere and post-depositional fluids, the pore water’s pH would have an upper limit of ~ 7.2 .”

and have modified the sentences in lines 319–322 of the original manuscript as follows in lines 402–406 of the revised manuscript.

“Our results of low levels of dissolved CO₂ in the pore water (Table 1) require both low CO₂ supply from the primary components and low dissolved CO₂ in the post-depositional fluids (see above). The former suggests the absence of carbonate in the matrix of the sediments²⁸. Given that post-depositional fluids would have originated from the surface in the full-interaction scenario, the latter may reflect low P_{CO_2} at the time of the last wetting event at Gale.”

Comment 8: Data of Fe-smectite

Line 157-158: This seems like a weak constraint on Fe²⁺ concentrations given that there is only one data point for interlayer spacing in Fe-smectites in Fig 1a of the supplementary. I recognize that the authors can’t do anything about this shortcoming of the available experimental literature, but the text on lines 157-158 is more definitive than it should be.

Response to Comment 8: Agreed. We have added available experimental data for Fe-smectite into new Supplementary Fig. 3.

We have added new experimental data on the interlayer spacing of Fe²⁺-saturated smectite (Sasamoto et al. 2017 Clay Minerals) into new Supplementary Fig. 3. In addition, we have added the interlayer spacing of Fe³⁺-saturated smectite into this figure (Kamei et al. 1999 Eng. Geol.; Sasamoto et al. 2017 Clay Minerals). Fe²⁺ in the interlayer is extremely unstable in the presence of significant amounts of oxidants even in the absence of liquid water (Manjanna, 2008 Appl. Clay Sci.). Thus, Fe²⁺ in the interlayer of smectite is expected to be already oxidized in the XRD measurements in Curiosity. The basal spacing of Fe³⁺ smectites at RH = 0% ranges from 10.9 to 11.9,

which is also different from the basal spacings for both Cumberland (13.2 Å) and John Klein (10 Å) observed by Curiosity. We believe that those revisions would make the statements more robust.

Comment 9: No evidence for glacial activity at Gale

Line 184: What in-situ observations of glacio-fluvial features in Gale are consistent with the presence of surface ice? The referenced paper in this sentence (manuscript reference 17) is outdated and based on orbital data collected prior to Curiosity's arrival in Gale crater. To the best of my knowledge, none of the purported glacial/ice related features identified along Curiosity's traverse in that paper have been demonstrated to have such an origin by subsequent in-situ observations.

Response to Comment 9: Agreed. We have deleted the expression of the presence of glaciers at Gale and the corresponding reference from the manuscript.

We admit that Curiosity has found no clear evidence for the presence of glaciers at Gale. In our scenario of formation of oxidizing post-depositional fluids, melting of glacier, indeed, is not needed. Melting of snow and frost, or deliquescence of perchlorate would be sufficient. The recent paper (Stamenković et al., 2018 Nature Geosci.) suggests that highly-oxidizing surface/near-surface fluids can be generated upon orbital changes even today's Mars. Based on the reviewer's comment, we have deleted the expression of glacier and the corresponding reference from the manuscript.

Comment 10: Redox disequilibrium at the time of deposition

Line 186-187: Why can't redox disequilibrium at the time that the Sheepbed sediments formed just as easily explain the coexistence of the Fe^{2+/3+}saponite, magnetite, akaganeite, and detrital sulfide minerals found in the Sheepbed member sediments, rather than calling on late stage oxidation by high Eh, low pH fluids?

Response to Comment 10: Coexistence of akaganeite and pyrrhotite could occur only when transport and burial of sediments occur in a short time (a few hundred years or less). This is because oxidants that form akaganeite also effectively oxidize pyrrhotite. The required timescale is shorter than a transport time of sediments for

terrestrial short rivers. We have added the above discussion.

Detrital pyrrhotite can be efficiently oxidized in solution under highly-oxidizing conditions that allow akaganeite formation. According to the previous study of kinetics of dissolution of pyrrhotite (Janzen et al., 2000 GCA), the oxidation rate of pyrrhotite with O₂ ranges ~10⁻⁹ mol/m²/sec for dissolved O₂ levels of 10⁻³ mol/kg. To achieve Eh higher than 0.5 V in solution, the dissolved O₂ is needed to be ~10⁻⁵ mol/kg or greater (Shaw et al., 1990 GCA). Assuming the oxidation rate is proportional to dissolved O₂ levels (e.g., Stumm and Morgan, 1996; in *Aquatic Chemistry*), pyrrhotite dissolution would proceed, at least, at 10⁻¹¹ mol/m²/sec under the conditions that allow formation of akaganeite. Using dissolution rate of 10⁻¹¹ mol/m²/sec, a 10- μ m pyrrhotite grain is completely oxidized in a few hundred years. Since oxidization of pyrrhotite with other oxidants, such as Fe³⁺, is faster than O₂ [Janzen et al., 2000 GCA], this time for oxidation would be a conservative upper limit. Our estimate suggests that coexistence of akaganeite and detrital pyrrhotite can occur only when the transport and burial of sediments occur in a short time of a few hundred years or shorter.

This timescale (a few hundred years) for oxidation of pyrrhotite is shorter than typical time of sediment transport for short rivers in a basaltic terrain (e.g., a few thousand years for rivers with length < 100 km: Vigier et al., 2006 EPSL), suggesting that detrital pyrrhotite is readily oxidized during weathering and transport. In fact, the coexistence of Fe(III) oxides and pyrrhotite in a single depositional environment is rare on Earth. Given the size of Gale Crater (D = 150 km), oxidization of pyrrhotite could have proceeded significantly during the transport. Early diagenesis within early Gale lakes would have taken a further long time. Such a short time of transport and burial of sediments suggests the extremely high accumulation rates of sediments at the time of deposition.

Overall, the no-interaction scenario requires the coexistence of akaganeite and pyrrhotite in a depositional environment, which in turn call for very rapid transport and high accumulation rates of sediments. We have added the following sentences in lines 300–310 of the revised manuscript.

“One possibility to generate the redox disequilibria is survival of detrital reducing minerals in an oxidizing shallow lacustrine environment. Under highly-oxidizing conditions that allow akaganeite formation, a silt-sized pyrrhotite grain would be oxidized in a few hundred years or less (see Sec. S5 of Supplementary Information). Thus, if transport and burial of detrital components occurred within a few hundred years, authigenic akaganeite could coexist with detrital pyrrhotite. However, this timescale for oxidation of pyrrhotite is shorter than the typical transport time of detrital components in short rivers (length of 40–100 km) on a terrestrial basaltic terrain (e.g., (1–3) × 10³ years)³⁸. Early diagenesis within the lakes would also have taken a long period of time. Hence, the no-interaction scenario would require

extremely rapid transport and high accumulation rates of sediments at the time of deposition of the Yellowknife Bay sediments. The high accumulation rates, in turn, imply a short duration of the early lakes given the thickness of sediments within Gale Crater.”

We also have added the explanation of calculation of oxidation of pyrrhotite grain in Sec. S5 of revised Supplementary Information.

Comment 11: Comparison with results of chemical index of alteration

Line 220-221: I think it’s important to be forthcoming and state that these climatological constraints come from analysis of two drill holes from the base of the section—they may not apply to the full stratigraphic section at Gale crater. Only a similarly detailed analysis of lake water conditions based on phyllosilicate structure-composition from other drill holes allow you to extend these conclusions regarding climate higher up in the stratigraphy. It’s also worth noting that the inference of arid/semi-arid conditions developed under cold climate conditions are consistent with independent constraints from bulk chemistry [3,5].

Response to Comment 11: Agreed. We have added a description that our climatological constraints would be only applicable to the Yellowknife Bay sediments. In addition, we have added the descriptions that our results are consistent with the results of low chemical index of alteration of the Yellowknife Bay sediments.

We appreciate the reviewer’s constructive comment. The climatic constrain from the present study is, of course, only applicable to Yellowknife Bay formation. The climatic reconstruction using the mineralogical data from other cores will be future works. We also have added the description that the estimated climatic conditions are consistent with independent constraints from bulk chemistry of the sediments (MacLennan et al 2014 Science; Hurowitz et al. 2017 Science). Following the reviewer’s comment, we have modified the original sentence as follows in lines 366–368 of the revised manuscript.

“we also suggest that the early Gale lakes developed under semiarid climatic conditions, at least, during the deposition of the Yellowknife Bay sediments, which is consistent with the low values of the chemical index of alteration for these sediments^{12,13}.”

Comment 12: Reference needed

Line 412: reference needed.

Response to Comment 12: The reference has been added.

We have added the reference (Bristow et al. 2015 Am. Mineral., ref. 15) in the revised manuscript.

Comment 13

Lines 450-453: see Comments 2-a (above).

Response to Comment 13

Please see Responses to Comments 2-a (above).

Comment 14

Lines 454-459: see Comment 2-b (above).

Response to Comment 14

Please see Response to Comment 2-b (above).

Comment 15: Was Fe²⁺ concentration negligible in pore water?

Line 484: How can one assume Fe²⁺ concentrations are negligible pore waters? Sheepbed member mudstones contain secondary Fe-oxides (akageneite, hematite, magnetite) and Fe-phylosilicate... surely there must have been some Fe²⁺ in solution for these phases to have formed?

Response to Comment 15: We admit that the original sentence was confusing. What we meant here is that {Fe²⁺} can be negligible in the “charge balance calculations”.

The sentence in line 484 in the original manuscript means that the Fe²⁺ concentrations are significantly lower than other major components, such as Na⁺ and Mg²⁺, and can be negligible in the “charge balance calculations”. In fact, the estimated Fe²⁺ concentrations are almost two order higher than that of terrestrial seawater. Based on the reviewer’s comment, we have modified the corresponding sentence in lines 534–535 of the revised manuscript, as follows;

“We assume that the H^+ , OH^- , HCO_3^- and Fe^{2+} are negligible in the charge balance calculations compared with other major species.”

Comment 16: Equation S3 is correct?

Equation S3 of the Supplementary Info: shouldn't the right side of the charge balance equation be written $3\{SO_4^{2-}\}$?

Response to Comment 16: The original equation is correct. The right side of the equation should be $4\{SO_4^{2-}\}$.

We appreciate the reviewer's careful check. The right side of the equation S3 should be $4\{SO_4^{2-}\}$, not $3\{SO_4^{2-}\}$. The charge balance equation of solution is calculated as follows:

$$\{Na^+\} + \{K^+\} + 2\{Mg^{2+}\} + 2\{Ca^{2+}\} = \{Cl^-\} + 2\{SO_4^{2-}\}. \quad (5)$$

$$\text{When } \{Cl\} = 2\{SO_4\} \quad (S2)$$

$$\{Na^+\} + \{K^+\} + 2\{Mg^{2+}\} + 2\{Ca^{2+}\} = 4\{SO_4^{2-}\}. \quad (S3)$$

Comment 17: Figure names in Supplementary Information

Line 45 of the Supplementary info: should read Figs. 5a, 5b, and 5c.

Response to Comment 17: We have corrected them.

We have corrected them to Figs. 5a, 5b, and 5c. Thank you.

Comment 18: $\log K_{Na_Ca} = \log K_{Na_Mg} = 0.2$?

Line 46 of the supplemental: why is the condition $\log K_{Na_Ca} = \log K_{Na_Mg} = 0.2$ being considered at all? I thought the clay basal reflections constrained your analysis to 0.4–0.6 (Supplementary Fig. 2).

Response to Comment 18: This may be a misunderstanding.

The $\log K$ value is the selectivity coefficient to calculate the relationship cation concentrations between the interlayer and solution (not the ratio of $(Na^+ + K^+) / (Na^+ + K^+ + Ca^{2+} + Mg^{2+})$) in new Supplementary Fig. 3b (original Supplementary Fig. 1b). According to the previous study (Tournassat et al. 2008 Soil Sci. Soc. Am. J., ref. 64), we use the $\log K_{Na_Ca} = \log K_{Na_Mg} = 0.4 \pm 0.2$. The value of 0.2 is lower limit of the selectivity coefficients for $\log K_{Na_Ca}$ and $\log K_{Na_Mg}$.

Comment 19: Comparison between JK and CB drill holes

Lines 144-145 are simply incorrect with regards to the JK and CB drill holes.

Response to Comment 19: Agreed. We have corrected the expression.

Based on the reviewer's comment, we have changed the original sentence as follows in lines 173–175 of revised Supplementary Information.

“Although the sulfate vein density at John Klein is locally higher than that at the Cumberland drilling site, sulfate veins are ubiquitously observed at Yellowknife Bay¹¹.”

Reviewers' comments:

Reviewer #4 (Remarks to the Author):

The manuscript (MS) is aimed at interpretation of mineralogy of sediments in Gale crater on Mars in terms of water composition and an early climate on Mars. One important conclusion is that smectite clays did not interact with NaCl-rich brines for a long time. The conclusion seems valid. This conclusion is interpreted in terms of a low-salinity lake water in the crater. The manuscript needs revision aimed at a more balanced description of possible and alternative scenarios, clearer explanation of the preferable scenario, and a better linking of models with modeling environments.

General comments

The MS is not balanced and did not discuss alternative more simple explanations. The MS presents one of possible (not a unique) explanations of observed data and is based on multiple assumptions. For example, the authors assume that clay minerals formed during diagenesis rather than in remote weathering sites (some works support the latter). The MS concludes that the lake was not saline, but the lake remains putative (one of explanations). What if the supposed Na-deficiency in minerals reflects formation conditions of clays in the presence of unsaturated (with respect to NaCl) waters in weathering sites? Another possibility: the formed clays may not have sufficient time to interact with waters after deposition with mud (clay-rich) flows. This alternative scenario corresponds to the observed diversity of clay minerals and to the high concentration of amorphous components. The diversity suggests a minor role of diagenetic process (except formation of late sulfate veins).

The described numerical models could be better linked to modeling environments in Fig. 1. The relations between modeling environments (Fig. 1) and models look confusing in several parts of the MS. For example, Table 1 is in the section that models a last wetting event but the table represents the composition of lake water that supposedly existed during initial diagenesis. In general, it was unclear how the lake water composition could be evaluated from models that address water-clay interaction during the re-wetting event (last aqueous event that may not be related to any lake).

Clays in two drilling sites have different basal spacing/composition (one is Na-rich and another is Mg-rich). In the main text and Methods, the MS concludes that both compositions correspond to 40-60% of Na-rich clays. I am not the expert, but the methods did not provide a clear explanation of these evaluations. To me, different spacing indicates different composition of clays that formed in different environments before deposition.

The conclusion about the lack of NaCl-rich waters is not supported by data on rate of cation exchange in clays. Fast freezing could have limited ion exchange at low-water conditions.

The title could be changed to reveal a low NaCl-salinity rather than redox processes.

Specific comments (! mark most important notes)

Abstract: The abstract is too short to understand what is done. It required editing (see below).

Abstract: The authors may mention source of data (MSL) and that solid materials (not water) have been analyzed by MSL. Please explain used methods.

Abstract, line 7: Please specify "Na-Cl concentrations suggest...". Do you mean MSL data on solid samples or presented numerical models?

Abstract, Line 8. What is the exact salinity? NaCl could deposit through evaporation of large (not saline) or small (more saline) bodies of water. Formally, the presence of NaCl may not reveal the initial volume of water and initial salinity.

Abstract, Line 9. The text states "observed redox disequilibria in the pore water". How can disequilibria be observed if nobody analyzed the pore water?

Abstract, Line 10. These worlds are not needed "chemistry of the fluids".

Line 12; "providing chemical energy" -> "providing chemical sources of energy" (there is no "chemical energy").

Line 33. "mineralogy and geochemistry" -> "phase and chemical composition"

Line 36. "probably" deposited {this is still a model interpretation}

Line 37 "river" -> "water stream"; "lake " or "lakes"?

Not sure we have enough evidence for lake(s) but waters streams (e.g. mud flows) likely occurred.

!!-Line 38-40. There are papers showing that clay minerals came with streams as a mud component (not diagenetic). Mud deposits mean deposition from mud (clay-rich) flows. Layers of rocks are compositionally diverse and suggest deposition of different mud/debris flows with a subordinate role of diagenetic processes. But the origin of clay minerals may not be very important for this research.

Line 44. Is it gypsum in the veins, not anhydrite, CaSO_4 (as in the text). Correct?

Line 48. "chemistries" -> "compositions"

! Line 51. I would write "contained SO_4^{2-} and Ca^{2+} ". Note that Na^+ , Mg^{2+} and Cl^- could be much more abundant but corresponding salts did not deposit because of high solubility. Instead, low-solubility Ca sulfate(s) precipitated in fractures.

Line 52. Is it vol. or mass %? Please specify.

Line 52. Suggested addition in { }: "this {NaCl-related} salinity" ..

Line 53-56. Is this conclusion about an early NaCl is based on this paper? If so, it should not be in the introduction.

Na-Al correlation is because of Na in in feldspar.

Line 62. Why "intrusion". What if (likely) solutions came from above?

Line 63. Do we have MSL data on last fluid-sediment interaction to constrain models?

Line 64, Why "duration" rather than detail MSL data on 'around vein' composition?

Line 66. What is "distance"? Do you mean "thickness" of veins?

Line 65-58. It is not clear how it was evaluated. Did you model diffusion of solution from a fracture to the distance of 1 cm from the fracture? Please clarify in the updated text.

Line 70. Do you really mean solid-solid equilibration? If NOT, consider ions in solution (Ca^{2+} ..)

!! Line 74. I am confused because previous text states that clays formed during diagenesis (not from lake water). But this (line 74) sentence implies that clays chemically precipitated from the lake water. It is unlikely to me and inconsistent with sedimentation of clays on Earth. Clays are typically supplied with streams (e.g., delta deposits). However, chemical modeling could be used to constrain water chemistry at which clays minerals formed at the first place (not in the lake).

Fig. 1. (a) Please mark green and brown forms in the quadrangle. Are any clays in initial sediments? They were likely present (despite several publications). (d) "& fluids" may not be needed (optional). (e) "Ca sulfates".

Line 92-93. "provide the constraints of water chemistry of the liquid water" -> "constrain the composition of liquid water"

! Line 92. Gypsum has low solubility and its presence may not indicate composition of "the pore water at the last rewetting event immediately before the disappearance of liquid water", as stated above. Ion-exchange in smectites is not fast (any data on the rate?). I think, only NaCl will indicate the composition of water "immediately before the disappearance".

Line 106-107. Possible language problem in this sentence: It is unclear what is "cation fraction of (Na + K) at 0.4 to 0.6". Changed from what fraction to what fraction? What cations (Mg?) represent the rest (another 0.4-0.6)?

Line 111. What would be an effect of high-salinity solutions?

! Line 113. Are concentrations of Ca and SO₄ controlled by solubility of gypsum? Are concentration of Na and Cl controlled by solubility of halite/hydrohalite? Are concentration of Mg controlled by solubility of Mg sulfates? What was the salinity?

Line 116. What is warm (please specify the temperature)? Why not to have low-T (~0C or below evaporation with or without partial freezing)?

Line 117. All salts (and not salts) are soluble. "Soluble" -> "high-solubility".

Line 117. The increase is not "gradual" because saturation of different phases is reached at different concentrations.

! Line 118. This statement does not look correct: "salts are most likely to have formed during sublimation after freezing." Experiments show that salts precipitate sequentially (for example, Ca-sulfate, Mg-sulfate, NaCl, and finally Ca-Mg chlorides). The sequential precipitation was modeled in many "martian" and "terrestrial" papers in which evaporation and freezing was calculated with Pitzer approaches.

! Line 121. Yes, if evaporation/freezing was slow enough. What do we know about kinetics of Na⁺ ion exchange in smectites? Can we evaluate fate of freezing/evaporation from these data?

Line 121. hydrohalite (in case of freezing)

Line 124. Why freezing? Both freezing and evaporation are considered above.

Line 125. This temperature could be estimated with high accuracy with Pitzer modeling of activities.

! Line 125. This temperature (0 to -2 C) is for initial ice formation (not complete freezing). Eutectic of Cl-rich solutions would be at much lower temperature (~ -20 or so, depending on the ion composition).

Line 128. "water chemistry" -> "Solution composition"

Table 1. Na -> Na⁺, K-> K⁺ etc.

! Table 1 Concentrations are not defined: It is molality (mol/kg H₂O) or molarity (mol/kg solution)?

Table 1. Is this low SO₄ content corresponds to precipitated gypsum?

Line 133. English: Dissolved species cannot be components of smectites.

Line 134. How do we know about "additional" components in smectites? It is probably an assumption. ! Smectites are different and their composition could (likely to me) reflect formation conditions away from the current deposition place. Otherwise, smectites would have a similar composition.

Line 135. I would delete "primary component" to avoid discussion about primary and additional components.

! Line 135. Clays formed away from the current deposition place could have exchanged cations with lake water and got more Na than they had. In such a case, Na-rich clays could be considered as “primary” phases formed in situ (in contact with sub-lake waters) from initially deposited clays.

! Line 152. Another explanation is no lake. In such a case, the estimated water composition (Table 1) corresponds to pore water during formation of clays or during clay-pore water exchange but still without a lake.

Line 199. At what temperature? What is the error bar for the red curve?

Line 211. Please check Wt% or Vol% smectite?

Line 212. What is “ferrian”? Do you mean “ferric” (= Fe³⁺-bearing)

Line 214. I would rather suggest formation of Fe³⁺-bearing saponite through alteration of basalt by aerobic waters before sedimentation at the current place.

Line 225. Magnetite and especially pyrrhotite could be primary minerals.

Line 231. During early diagenesis or at primary conditions of saponite formation through in situ alteration of basalt (not through diagenesis).

Line 233. Not sure that authors of [31] are correct. Magnetite could be (and likely) primary.

Line 250. This is unclear: “allowing interactions with the atmosphere”.

Line 252. They could survive in ice as super-cold liquid acid droplets in the ice and migrate toward the rock (discussed in martian weathering literature).

Line 252. “acidity” -> “protons” (optional).

! Line 256. In addition, the amount of acids and trapped oxidants could be insufficient to modify the rock-dominated system.

Line 255. Different redox reactions have different rates: some reactions are inhibited but reactions rates with strong oxidants (O₂, for example) are slow but not negligible at O₂-poor and cold martian environments (Burns, 1993; several recent works).

Line 256. Freezing is likely but evaporation was mentioned above. Please be consistent.

Line 265. “total activities” -> “total concentrations” Please clearly specify used total concentrations of elements.

Line 270. It is hard to follow this section below. Some notes are listed below.

!! Line 272. I am confused. There is no no-alteration in Fig 1 b. The fill alteration scenario was related to the rewetting event (Fig. 1d, Table 1 ?). Why is the no-interaction scenario related to early diagenesis? Do you mean “no alteration scenario” during early diagenesis? How did clays form?

!! Please explain the meaning of no-alteration scenario. How is it different from another scenario?

! Line 302. Yes, but not in a closed system with limited oxidants. Diagenetic systems could be semi-closed or closed systems.

! Line 305. Martian atmospheric O₂ is not as abundant as on Earth (~5000 times less), so the oxidation rate is correspondingly slower (see Burns, 1993).

Line 321. Gypsum or anhydrite?

! Line 321. Why not to have Ca-sulfate some deposition from primary diagenetic solutions? Sulfates are abundant on Mars and could be present in all solutions.

Line 323. Those values could be reconsidered.

Line 338. I would add “ ... during the last wetting exclude that affected the sediments.”

!!! The authors could discuss another explanation: Insufficient water-rock interaction during the re-wetting event due to a low water content, low permeability of clay-rich rocks and a short duration/fast freezing. In such a case, the composition of clay minerals would mainly reflect previous environments (an early diagenesis or formation before deposition).

!! Line 334-337. Again, it was unclear how the lake water composition could be evaluated from models that address the re-wetting event.

Line 353, 364. Evaporation of freezing? Lake's life time will be longer if the lake is mostly covered by ice.

Line 393. disequilibria

Line 395-397. Redox interactions decrease amounts of energy for metabolism. The lack of abiotic redox reactions provides sources of energy.

Line 452. Note that volcanic SO₂ leads to cooling of Earth's atmosphere.

! Line 450. Do you mean solubility of Mg-saponite? Mg is metal.

Line 451. Do you mean “brucite structural group” of brucite as mineral? I think, solubility of saponite should be discussed, not brucite of “Mg”.

!! Line 473: Smectites section. The text states “The peak position of the 001 reflection of John Klein is 10.1Å, suggesting that the cation composition of the John Klein smectite could be dominated by Na+ and/or K+”. I failed to understand how different clay compositions (Na-rich and Mg-rich with different basal spacing) in John Klein and Cumberland, respectively, were interpreted in terms of “cation fractions of (Na++K+) and Mg²⁺ needs to be 0.4–0.6.” in both locations? To me, the simplest explanation of different clay composition is deposition of compositionally different clays with inefficient changes during diagenesis.

Line 466-470. I do not understand the meaning of this text, especially the last sentence.

Line 493. Hydrohalite could form upon freezing.

Line 499. “XRD” -> “CheMin”? “XRD” is a method, not an instrument.

Line 516. More information would be useful (In Supplementary Materials)

! Line 519. The presence of abundant amorphous material suggests a limited water-rock reactions and water-smectite ion exchange during diagenesis.

Line 523. Several activity coefficient are needed. Do you mean coefficients? What are sources for thermodynamic data for Fe-saponite and ions?

Line 524. mol/(kg H₂O)

Line 590. Do you mean a re-wetting episode or initial diagenesis?

Line 609. Released from which mineral? What if Fe-smectite was (likely) deposited with the mudflow?

Line 613. What is the oxidant if only water is present?

Line 612-614. Why is it oxidized? Are there enough oxidants to oxidize abundant Fe²⁺ in saponite? The spacing tells us that Fe³⁺ is not abundant in smectite.

Line 617. Do not understand why this sentence appears here and do not understand from which mineral Fe²⁺ is released and how Fe²⁺ concentrations can “maintain” spacing: “Hence, released Fe²⁺ concentrations must be sufficiently low to maintain the observed basal spacing of the smectites.”

Line 621 Is albite really detected? Thermodynamically, analcime could be more appropriate.

Line 647. What is the value of K(21) and how it was obtained?

Supplementary material: Line 200. Concentration of O₂ in dissolved water would reflect the low O₂ level of the martain atmosphere. The authors could evaluate the rate of pyrrhotite oxidation by the low martian O₂ = 0.0014x0.006 bar compared to 0.2 bar O₂ on Earth.

Line 204. Please remind about Eh >0.5 V. It is related to akaganeite stability? Do we know thermodynamic data for akaganeite. What is the uncertainty?

Reviewer #5 (Remarks to the Author):

K. Fukushi, Y. Sekine, and R. Wordsworth use the mineralogy and geochemistry of sediments from the Yellowknife Bay formation in Gale crater, Mars to apply a top-down approach and quantitatively estimate the pore water chemistry during early and late diagenesis. Based on their calculations, they suggest early Gale lakes were hyposaline, developed in semiarid climates, and hydrological cycles

persisted for 10^4 - 10^6 years. They demonstrate that sedimentological and geomorphological observations are consistent with their conclusions. Furthermore, they suggest that late diagenetic fluids would have been acidic and oxidizing to explain the mineral assemblage of the John Klein and Cumberland drill targets measured by the CheMin instrument on the Curiosity rover.

Using mineralogy and geochemistry to constrain fluid chemistry of ancient Mars, as Fukushi et al. have done here, is important for characterizing past aqueous environments and the habitability of the martian surface and shallow subsurface. The calculations and most assumptions are sound and the manuscript is well written. Some of the mineralogical inputs into their model are no longer valid, and the authors should address these before the manuscript is accepted. Additionally, the assumption that John Klein and Cumberland were altered by the same diagenetic fluids warrants further justification. I elaborate on these major comments below and also have a few minor comments. I recommend this manuscript be accepted after these comments are addressed. If the authors have any questions about my comments, they should feel free to email me, Liz Rampe, at elizabeth.b.rampe@nasa.gov.

Major Comments

Some of the mineralogical inputs into the models are no longer valid. One important constraint that Fukushi et al. use for their models of pore water chemistry is the presence of 0.1-0.4 wt.% halite in the Yellowknife Bay sediments. Vaniman et al. (2014) *Science* indeed reported 0.1 wt.% halite in John Klein and Cumberland based on refinements of CheMin data (I don't know the source of the reported 0.4 wt.% halite), but the CheMin team has recently applied an important correction to the patterns collected earlier in the mission and re-analyzed these corrected patterns. Machining offsets in the sample wheel result in differences of 10s of microns from the ideal diffraction position, and a linear relationship between plagioclase unit-cell parameters has allowed the team to account for these differences in diffraction position (Morrison et al., 2018). The analysis of the corrected patterns of John Klein and Cumberland showed no halite or pyrrhotite above the detection limits of CheMin (~1 wt.%). There could be halite present in abundances <1 wt.%, and I suggest the authors re-frame their arguments and calculations with this new constraint. The authors may also use akaganeite to constrain salinity of diagenetic fluids because the identification of akaganeite is robust.

The assumption that John Klein and Cumberland were altered by diagenetic fluids with the same composition, resulting in similar interlayer cation compositions, warrants further justification. ChemCam data from raised ridges in Yellowknife Bay demonstrate that these features are enriched in MgO by 1.2-1.7 times over that of the surrounding mudstone (Leveille et al., 2014). This demonstrates that there is geochemical heterogeneity in the mudstone, which may have originated from diagenetic fluids altering different portions of the sediments. The authors should take these observations into consideration for their conceptual models. Furthermore, the basal spacings of

cation-exchanged smectite measured in the laboratory do not exclusively support the authors' hypothesis that the smectite in John Klein and Cumberland have similar compositions. Supplementary Figure 3c demonstrates that smectite with Na+K/Mg ratios of 0.4-0.6 show differences in their d001 of ~3 angstroms (similar to that observed for John Klein and Cumberland smectite). Supplementary Figure 3c, however, reports basal spacings of smectite measured at 40% relative humidity, rather than ~0% relative humidity, as the samples experience in the CheMin instrument on Curiosity. This difference in basal spacings of ~3 angstroms at 40% RH may not hold true at 0% RH. The authors should further justify their assumption that John Klein and Cumberland were altered by the same diagenetic fluids in light of this. Supplementary Figure 3a shows basal spacings for cation-exchanged smectite at 0% RH, which seems to demonstrate that the interlayer sites of the smectite in John Klein must be dominated by Na+ and/or K+, while those of the smectite in Cumberland must be dominated by Ca2+ and/or Mg2+. Therefore, the alternate hypothesis that the smectite in John Klein and Cumberland have different interlayer cation compositions is also valid.

Minor Comments

Line 35: Suggested edit: delete "spatially enclosed" or expand upon it because I don't know what it is referring to. Is the Yellowknife Bay spatially enclosed or are the drill core samples?

Line 37: Replace "the lake" with "a lake" because there may have been many on the crater floor.

Line 38-40: The bulk mineralogy does not suggest that most of the chemical reactions generated phyllosilicates during early diagenesis. The similarity of the bulk composition of the Yellowknife Bay sediments to the composition of the bulk martian crust suggests early diagenesis in a closed system (e.g., McLennan et al., 2014). I suggest this sentence be re-worded.

End of line 45: I suggest a brief description of Aeolis Mons (e.g., "These rewetting events occurred within Gale even after the formation of Aeolis Mons, a 5 km mound of layered sedimentary rock in Gale crater.").

Supplementary Figure 1: I disagree that the correlation of Al and Na demonstrates that Na in the final pore water was derived from the paleo lake water. The authors use this figure to justify some assumptions in their geochemical models in their response to the reviewers. The majority of the Na in the mudstone is likely in feldspar, and the correlation of Al and Na may be a consequence of abundant feldspar in the mudstone. Again, CheMin does not detect halite so this phase cannot be used to constrain Na in the pore water. A correlation between Na and Cl is necessary to demonstrate

the presence of halite in the mudstone. However, the 10 angstrom basal spacing for smectite in John Klein suggests Na may be present in the interlayer site. The authors may use this constraint to characterize Na in the pore water.

Line 64: There is a reference for Figure 1f, but this doesn't exist.

Line 92: Change "presences" to "presence." Also, the assumption of gypsum in Yellowknife Bay sediments should be made here because CheMin did not detect any gypsum in John Klein or Cumberland. CheMin detected bassanite and anhydrite, and bassanite commonly forms from the dehydration of gypsum. This did not occur within the CheMin instrument as reported in subsequent samples by Vaniman et al. (2018). Gypsum dehydrates to bassanite in CheMin over the course of ~35 sols. CheMin did not observe gypsum in the first analyses of John Klein or Cumberland.

Line 163: Without petrographic information, the association between smectite and akaganeite and their presence in the matrix cannot be confirmed. Suggested edit: "Furthermore, akaganeite, a chloride-bearing ferric oxyhydroxide, occurs with smectite in the Yellowknife Bay mudstone."

Line 193: There is not a complete lack of carbonate in the Yellowknife Bay sediments. Ming et al. (2014) Science report on the evolution of gases from John Klein and Cumberland in the Sample Analysis at Mars (SAM) instrument. They identify CO₂ in abundances of up to 0.8 wt.% that could be from Fe-/Mg-carbonates and organics. This should be used as an upper limit for carbonate abundance in Yellowknife Bay.

Line 211: Delete "of" to read "~20 wt.% smectite."

Lines 225, 296, 298: Delete pyrrhotite.

Line 237: Rampe et al. (2017) reported on acidic alteration in the Murray formation. Yen et al. (2017) reported on acidic fluids in the Stimson formation. I suggest citing Yen et al. here, too.

Line 251: Suggest replacing "fogs" with "fog"

Line 258: Suggest replacing "alterations" with "alteration" or "episodes of alteration"

Lines 301-307: With the absence of pyrrhotite, there is no need to explain the coexistence of akaganeite with pyrrhotite.

Line 343: Suggest deleting “formation”

Line 358: Cite a reference for an ancient martian ocean.

Line 361: Cite a reference for the difficulty in achieving higher-temperature conditions with climate models.

Line 364: I suggest using a different word than “precipitation” because fluids may not have entered Gale crater via rain or snow. Perhaps state that “evaporation dominated fluid input.”

Line 372: Some Japanese characters snuck into the manuscript.

Line 406: Instead of a low PCO₂ at the time of the last wetting event at Gale, could the relative absence of carbonates indicate that the diagenetic fluids were not in equilibrium with the CO₂ in a thicker atmosphere? For example, if fluids were sourced from rapid melting of ice or sourced from the subsurface, perhaps the fluids did not have ample time to interact with the CO₂ in the atmosphere to dissolve substantial CO₂.

Lines 412-414: The estimated timing of this last wetting event (Amazonian) is consistent with the youthful age of jarosite in the Mojave2 sample measured from the Murray formation stratigraphically above the Yellowknife Bay formation. Martin et al. (2017) report on the K-Ar age of jarosite measured by the SAM instrument and find the age is 2.12 +/- 0.36 Ga. The authors may choose to cite this to further support the timing of the last acidic-oxidizing aqueous event in Gale crater.

Line 417: “volcanoes” is misspelled.

Line 492 (section about halite): Delete this section because CheMin does not positively identify halite.

Lines 499, 502: "bassanite" is misspelled.

Lines 501-502: Suggestion to delete the sentence stating bassanite is a product of gypsum dehydration in CheMin because that was not observed for the John Klein and Cumberland samples. (That behavior was observed for some samples in the Murray formation.)

New References

Léveillé, R. J., et al. (2014) Chemistry of fracture-filling raised ridges in Yellowknife Bay, Gale crater: Window into past aqueous activity and habitability on Mars. *J. Geophys. Res. Planets*, 119, 2398-2415.

Martin, P. E., et al. (2017) A two-step K-Ar experiment on Mars: Dating the diagenetic formation of jarosite from Amazonian groundwaters. *J. Geophys. Res. Planets*, 122, 2803-2818.

Ming, D. W., et al. (2014) Volatile and organic compositions of sedimentary rocks in Yellowknife Bay, Gale crater, Mars. *Science*, 343, 1245267.

Morrison, S. M., et al. (2018) Crystal chemistry of martian minerals from Bradbury Landing through Naukluft Plateau, Gale crater, Mars. *American Mineralogist*, 103, 857-871.

Yen, A. S., et al. (2017) Multiple stages of aqueous alteration along fractures in mudstone and sandstone strata in Gale crater, Mars. *Earth and Planet. Sci. Lett.*, 471, 186-198.

Review of “Dynamic climate and redox interactions on early Mars inferred from water chemistry at Gale,” by Fukushi et al., submitted to *Nature Communications*.

Review by Edwin Kite, University of Chicago. I review non-anonymously.

Fukushi et al. analyse the composition, mineralogy, and smectite layer-spacing of lacustrine mudrocks sampled by the MSL Mars rover at Yellowknife Bay (YKB), Gale crater, Mars. This lake deposit has previously been shown to record an environment that was habitable (Grotzinger et al Science 2014). Using a thermodynamic method that has been shown to work on Earth but has never previously been applied to Mars, Fukushi et al infer the salinity and pH of the lake waters at Gale and also estimate the duration of the habitable lake environment.

This is a well-written and exciting manuscript on a crucial topic. Both the methods (as applied to Mars) and the results are new. The “top-down” method in particular advances the state-of-the-art and could help stimulate lots of new work in the Mars geochemical modeling community (which right now is stuck in a rut of using 1D forward reaction-transport models that often return ambiguous answers). Another very nice feature of the manuscript is that it synthesizes a lot of data that have not previously been brought together in one paper. The assumptions in the manuscript are in most cases clearly explained and argued. However, the arguments are often probability arguments - alternative scenarios are not “ruled out,” they are only “argued against”. Therefore, it is possible that the scenario advocated for in the manuscript will turn out to be wrong. Nevertheless, the abstract acknowledges that the data are only suggestive, and the text discusses alternative explanations in most cases, so the paper is honest. Therefore, given that the paper is novel in terms both of methods and results, and is likely to be of interest to a wide community not just of planetary scientists but also of Precambrian researchers and paleo-oceanographers. I think only changes that are minor in the context of the broad sweep of the manuscript are needed before publication in *Nature Communications*.

(I am not an expert on mineralogy, so I will focus my comments on the geology/context, and the physical/chemical models. If the Editor would like another mineralogy expert to review this manuscript, then I recommend Clara Blättler at the University of Chicago, or Mohit Melwani Daswani at JPL. I did not participate in the first round of review. I read the entire text, the first-round comments, the response to first-round comments, and I skimmed the supplementary. I did not check the equations and I did not check Figures S5-S14.)

Comments

- The workflow hinges on halite. But there is no definite detection of halite by CheMin at YKB (Vaniman et al. Science 2014). Indeed, none of the references on line 92 supplied to support the “presences of [...] halite” actually claim a halite detection. This is not a big problem in itself, because halite could be present below detection limit. However, Cl could be hosted by oxychlorine

- species (Sutter et al. International Journal of Astrobiology 2017, Hogancamp et al. JGR-Planets 2018). See especially Section 4.9 and Fig. 18 of Sutter et al JGR-Planets 2017, and Section 4.1 of Farley et al. EPSL 2016, for their discussion. What would happen to the conclusions of the manuscript if say 25% of the Cl was hosted by oxychlorine species (chlorate/perchlorate)? What if 75%?
- The lake-lifetime calculation assumes that the Na is leached from the catchment soil/rock by the same wet event that filled the lake. Thus a lake lifetime of $>10^4$ years is obtained. But this need not be true; during a long cold period, large quantities of volcanic acids (e.g. HCl) can interact with the soil, forming salts that are quickly flushed out during much-more-brief surface-runoff events. This second scenario is proposed for Mars' THEMIS-detected halide lake deposits by Melwani Daswani & Kite (JGR, 2017). Therefore, the lake-lifetime constraint does not follow from the data presented (without additional unstated assumptions).
 - Schieber et al. Sedimentology 2017 provide a detailed argument (abstract, and their section "Perspectives on the origin of clays") that at least some of the clays are detrital. This contradicts line 38 of the manuscript. Please explain in the manuscript the response to Schieber et al.'s argument.

Line-by-line comments

- Abstract. Suggest explain methods (one or two sentences). Suggest clarify that conclusions are "within our model assumptions".
- l.10 Please clarify that post-depositional fluids are being referred to here.
- l.121-126 – This depends on the rate of water-table lowering relative to the rate of groundwater flow. See Andrews-Hanna & Lewis, JGR-Planets, 2011 (or Horvath & Andrews-Hanna, GRL, 2017) for a detailed quantitative scenario. If the water-table is lowering slowly relative to the evaporation rate (i.e. evaporation is mostly balanced by upwelling of water from depth, as in Andrews-Hanna's global deep groundwater circulation model) then the sediment can be bathed in pervasive groundwaters for a long time.
- l. 179-183 - The absence of calcite is used as part of a thermodynamic argument. But Halevy & Schrag (GRL, 2009) show that small concentrations of CO₂ are enough to prevent calcite precipitation. Also Tosca et al. (Nature Geoscience, 2018) show experimentally that kinetic effects can prevent carbonate precipitation even at high pCO₂.
- l. 259 – It is not clear to me where the argument for "acidic-oxidizing conditions [...] were relatively short-lived" comes from.
- l. 329-331 – See my main comment above on charging-up the soil with Cl-salts during dry periods.
- l. 341 – "unique constraints" – This is a very strong phrasing. Alternative explanations exist. Suggest "constraints" instead.
- l. 347-348 - "proposed lake volume" – Here the lake volume is estimated using Palucis et al JGR Planets 2016. Palucis et al. refer to post-Mount-Sharp lakes. However Grotzinger et al. Science 2015 interpret YKB sediments as

pre-Mount-Sharp. In other words, Palucis et al. interpret their lakes as much younger (a late-stage, relatively minor event) than the YKB lake event as interpreted by Grotzinger et al. Science 2015 (a Gale-filling, pre Mount Sharp event). For the lake volume estimation, the use of Grotzinger et al. Science 2015 interpretation would strengthen the lake-lifetime conclusions. Overall, the timing of YKB lake deposits relative to Mt. Sharp formation is not certain, although this doesn't affect the strong astrobiological interest of these deposits.

- l. 372-373 – Both the valley networks and the Al-rich surface clays mostly predate the formation of Gale crater. Please clarify.
- l.388-390 – Oxidants could also accumulate in upstream soils and be washed down during a wet event. Please clarify that possible oxidants include oxychlorine compounds (e.g., chlorate).
- l.394-397. This is speculative.
- l.407-411 – See Tosca et al. Nature Geoscience 2018 for counterarguments.
- l.415 – Both Turbet (PhD thesis 2018, <http://www.lmd.jussieu.fr/~mturbet/these Martin Turbet/>) and Steakley et al. (4th Intl Conf Early Mars 2017), find that warm climates last just a few years after a big impact. Gale is one of the last 150km-sized impacts on Mars, are there any post-Gale impacts big enough to trigger a warming event of the required duration?
- l. 662-665 – See main comment above and Melwani Daswani & Kite (JGR Planets, 2017) for an alternative interpretation that would give much higher dissolved-Na concentrations.

Little things

- Figure 1, “Aeolis” is misspelt (twice)
- Figure S10, on my printout the x-axis labels for panels (a) and (b) were in the wrong places.

Responses to comments from Reviewer #4 for ‘Semiarid climate and hyposaline lake on early Mars inferred from water chemistry at Gale’ by Fukushi, K. et al. (MS# NCOMMS-18-21962A)

We are grateful to reviewer #4 for the constructive comments and suggestions. Below, we give our responses in turn following each comment, with the reviewers’ comments being in Arial font and our responses being in Times font.

Major Comment 1: The MS is not balanced and did not discuss alternative more simple explanations. The MS present one of possible (not a unique) explanation of observed data and is based on multiple assumptions. For example, the authors assume that clays minerals formed during diagenesis rather than in remoted weathering sites (some works support the latter). The MS concludes that the lake was not saline, but the lake remains putative (one of explanations). What if the supposed Na-deficiency in minerals reflects formation conditions of clays in the presence of unsaturated (with respect to NaCl) waters in weathering sites? Another possibility: the formed clays may not have sufficient time to interact with waters after deposition with mud (clay-rich) flows. This alternative scenario corresponds to the observed diversity of clay minerals and to the high concentration of amorphous components. The diversity suggests a minor role of diagenetic process (except formation of late sulfate veins)

Response to Major Comment 1

The reviewer mentioned that the manuscript needs to be balanced by discussing alternative possibilities. Based on this comment, we have discussed multiple scenarios of distinct stages of alterations in the revised manuscript as much as possible. For instance, we have added discussion on 1) a possibility of detrital origin of clay minerals, 2), an effect of timescale of interactions of clays with water, and 3) an effect of previous rewetting events before the last wetting event in the revised manuscript. In addition, 4) to balance discussion, we have largely modified the abstract by including discussion of alternative scenarios.

However, the alternative scenarios raised by the reviewer (water chemistry reflecting formation conditions, and incomplete interactions between clays and fluids) are essentially based on the reviewer’s concern of slow kinetics of the cation exchange reactions of smectite’s interlayers. Nevertheless, the cation exchange reactions of smectite’s interlayers are known to be a rapid process with low activation energy (order

of sub-second) (Ogwada and Sparks, 1986; Crooks et al 1993), whereas the formation and alteration of smectite take much longer times (order of 10^2 years or longer). The present study has used the exchangeable cation compositions of smectite's interlayer (not the bulk chemical composition of smectite) to estimate the water chemistry. Thus, the alternative scenarios raised by the reviewer (water chemistry reflecting formation conditions, and incomplete interactions between clays and fluids) are not applicable to our model. In the revised manuscript, we have explained the theoretical background of our methodology in more detail. We hope that the additional explanations would help the readers to avoid considering unnecessary many alternative scenarios. Below, we describe our modifications on the above points 1) – 4) in more detail.

Concerning the point 1), the reviewer mentions the possibility that clays formed at source regions of the sediments. Even if this is the case, our results of the estimated pore-water compositions do not change. This is because we use the exchangeable cation compositions of smectite's interlayer to estimate water chemistry, as mentioned above. The cation exchange reactions between smectite's interlayer and surrounding water proceed rapidly (within a second) (Ogwada and Sparks, 1986; Crooks et al 1993). Plus, its activation energy is very low (diffusion limited). Thus, this reaction proceeds effectively even at low temperatures (e.g., within a few second even at $\sim 0^\circ\text{C}$). Regardless of formation conditions of clay minerals (detrital or diagenesis), smectite's interlayers can record the cations of surrounding water that finally interacted with the clay rocks. Given such rapid exchange reactions and multiple evidence for water activity (water stream and post-depositional wetting events) at Yellowknife Bay, smectite's interlayer should record the pore water chemistry at the deposition site. In the original manuscript, we assume early diagenetic origin of clay minerals; however, we agree the possibility of detrital origin of some clay minerals. Our results of water chemistry do not rely on the formation conditions of clay minerals. To discuss this point, we have added the following sentences in lines 27–38 of p. 3 of the revised manuscript.

“Smectite is unique in possessing exchangeable cations in interlayers⁷. Although there is a wide diversity of mineralogical characteristics of smectites including the chemical compositions and formation processes, all smectite possess the cation exchange ability (Supplementary Fig. 1). The cation exchange reaction of smectite is an instantaneous process with sub-second timescale in suspension solutions^{8,9}. The rate limiting step of the exchange reaction is diffusion and, thus, the activation energy is very low^{8,9}. Therefore, the exchangeable cations in smectite can readily record the surrounding solution composition even at low temperatures (e.g., $\sim 0^\circ\text{C}$)⁴⁻⁶. On Mars, there is a debate about the formation process of smectite in sediments, i.e., *in-situ* formation by early diagenesis¹⁰⁻¹³ vs. detrital deposition³. However, regardless of the chemical compositions and formation conditions of smectite, the top-down approach can provide a robust “snapshot” of the chemical

properties of the pore water that finally interacts with the clay rocks⁴ **if the clay rocks contain smectite.**”

We also agree the possibility of detrital origin of clay minerals, suggested by Schieber et al. (2017). We have modified Fig. 1 to include detrital clay minerals in the revised manuscript. In addition, we have added the following descriptions of the possibility of detrital origin of clay minerals in the revised manuscript.

“The Yellowknife Bay sediments contain ~20 wt% of smectite¹². These smectites would have formed in the early diagenesis within the lake^{12,13} (Fig. 1b) **and/or may have been of detrital origin**³ (Fig. 1a).” (lines 47 – 49 in p. 4)

“Fe³⁺ saponite **needs to have been originally Fe²⁺ saponite in the sediments, whichever it is of diagenesis or detrital origin.**” (lines 228 – 229 in p. 12)

“**Although magnetite and saponite might be of detrital origins, their relatively-high abundance suggests that the pore-water pH would be compatible with their thermochemical stability.**” (lines 240 – 242 in p. 13)

Regarding the point 2) of the possibility of incomplete cation exchange reactions, cation exchange reactions between smectite and surrounding water proceed in order of second or less even at low temperatures (see above). Thus, the presence of amorphous phases in the Yellowknife Bay sediments do not mean that the cation exchange reactions proceed only incompletely because even formation of amorphous phases takes much longer times (e.g., Tosca et al., 2004). As shown in the point 1) above, we have added the detailed descriptions of kinetics of cation exchange reactions in the revised manuscript.

Concerning the point 3), we have discussed alternative possibilities of multiple wetting events at Yellowknife Bay in the revised manuscript. In particular, we have discussed the possibility of groundwater upwelling at Gale in the early post-depositional stages (e.g., Horvath and Andrews-Hanna, 2017). If this occurred, bathing of the sediments by upwelling groundwater could dilute salinity in the sediments. Thus, we consider that the estimated Na in the pore water would be a lower limit of that of the early lakes. To discuss this point, we have added the following sentences in the revised manuscript.

“**Groundwater within terrestrial basalts contains typically low Na concentrations (10^{-4} – 10^{-3} mol/kg; see Methods). Given the possibility that the primary components could have been leached due to upwelling groundwater into Gale¹⁹, the estimated Na concentration would be a lower limit of the primary Na⁺ concentration in the bottom water of the early lakes. Nevertheless, we consider that the original Na⁺ concentration in the bottom water would not be remarkably higher than those in Table 1 because of lack of evidence for drying hypersaline lakes (e.g., desiccation cracks and rip-up chips)²⁵ at Yellowknife Bay¹⁴. We consider that if the early lakes existed, its Na⁺ concentration would have been on the same order of magnitude as that of the pore water in the last wetting event.**” (lines 156 – 163 in p. 9-10)

In the no-interaction scenario (see the main text), upwelling groundwater could have caused reduction of the oxidized sediments. We also discuss this effect in the section of “the no-interaction scenario”, as follows.

“One possibility to generate the redox disequilibria in the no-interaction scenario is interactions of upwelling groundwater at Gale¹⁹ with initially oxidized sediments¹⁰ after deposition. Groundwater within Martian crusts would become reducing and alkaline¹⁰. If reducing groundwater upwelled into oxidized sediments at Gale, Fe²⁺ saponite would have formed through reduction of oxidized smectite, e.g., nontronite, by groundwater. Akaganeite in the sediments could have been also reduced into magnetite by groundwater; however, this conversion might have proceeded only incompletely due to kinetics. The occurrence of the redox disequilibria due to groundwater upwelling would be also applicable to the full-interaction scenario.” (lines 303 – 310 of p. 15-16)

In response to the comments from the reviewer #6, we have also added the possibility of enhanced chemical weathering due to accumulated volcanic acids, as follows.

“Accumulation of volcanic acids during short-term intermittency of lakes would decrease solution pH and, thus, increase dissolution of Na from rocks². However, acidity of solutions would be neutralized within a short time ($\sim 10^2$ years) via interactions with rock¹. Thereby, the estimated total duration of 10^4 – 10^6 years would not be dramatically reduced by accumulation of volcanic acids.” (lines 353 – 357 of p. 17)

As for the point 4), we have modified the abstract by adding the descriptions on alternative scenarios (see below).

“Here we constrain these properties of pore water within lacustrine sediments of Gale Crater, Mars, using smectite interlayer compositions. Regardless of formation conditions of smectite, the pore water that last interacted with the sediments was of Na-Cl type with mild salinity (~ 0.1 – 0.5 mol/kg) and circumneutral pH. To interpret this, multiple scenarios for post-depositional alterations are considered. In any scenarios, the estimated Na-Cl concentrations would reflect hyposaline, early lakes developed in 10^4 – 10^6 -year-long semiarid climates. Assuming that post-depositional sulfate-rich fluids interacted with the sediments, the redox disequilibria in secondary minerals suggest infiltration of oxidizing fluids into reducing sediments. Assuming no interactions, the redox disequilibria could have generated by interactions of upwelling groundwater with oxidized sediments in early post-depositional stages.” (lines 4 – 14 of p. 2)

Overall, we have added the detailed descriptions of theoretical background of our methods and alternative possibilities that could affect our estimated water chemistry as much as possible. We hope that this modification would improve the balance of discussion and would make our conclusions more robust.

Major Comment 2: The described numerical models could be better linked to modeling environments in Fig. 1. The relations between modeling environments (Fig.1) and models look confusing in several parts of the MS. For example, Table

1 is in the section that models a last wetting event but the table represent the composition of lake water that supposedly existed during initial diagenesis. In general, it was unclear how the lake water composition could be evaluated from models that address water-clay interaction during the re-wetting event (last aqueous event that may not be related to any lake)

Response to Major Comment 2

First, we would like to clarify that the water chemistry in Table 1 represents pore-water compositions that finally interacted with the sediments and does not directly represent the water composition of the early lakes. We consider that after early post-depositional stages of the lake sediments, primordial liquid water (possibly trapped bottom water of the lake) in pores of the sediments would have been lost. However, dissolved species in the pore water would have been left within the sediments, e.g., as trace of evaporites. If post-depositional sulfate-rich fluids introduced into the matrix of the sediments, the primordial evaporites would have been re-dissolved and mixed with the additional components from the post-depositional fluids.

To explain this more carefully, and to better link to the geological context shown in Fig. 1, we have modified the sentences that explain how the primary components would have been preserved in the sediments in lines 58 – 66 of p. 4 of the revised manuscript, as follows.

“Bottom water in lakes is continuously trapped within pores of sediments and buried together¹⁸ (Fig. 1b). If groundwater largely upwelled into Gale Crater¹⁹, this could have also affected pore water chemistry in the early post-depositional stages. After loss of liquid water trapped in pores of the Yellowknife Bay sediments, dissolved components would have been left in the sediments, e.g., as trace of evaporates (Fig. 1c). The presence of sulfate-rich fractures strongly suggests that the post-depositional fluids provided additional SO_4^{2-} into the sediments in the last wetting event^{11,16}. If the post-depositional SO_4^{2-} -rich fluids infiltrate into smectite-rich matrix of the sediments, the primary evaporates left in the matrix would have been re-dissolved and mixed with the additional components (Fig. 1d).”

We suggest that Na in the pore water are largely derived from the lake water (i.e., the primary components). This is the case because the post-depositional sulfate-rich fluids were highly likely depleted in Na, as evident from the fact that Na contents are almost zero in Ca-sulfate veins with the highest SO_3 contents (Nachon et al., 2014). In addition, Ca-sulfate veins are commonly found in the Murray and Stimson sediments at Gale. Ca-sulfate veins in the Murray and Stimson sediments are also depleted in Na (Yen et al., 2018). To clarify this, we have added the following sentences to explain why Na in the pore water would be largely derived from the early lake water in lines 151 – 155 of

p. 9 of the revised manuscript.

“Additionally, the post-depositional SO_4^{2-} -rich fluids in the last wetting event were depleted in Na¹⁶. Na contents are almost zero at sulfate-rich veins with the highest SO_3 contents¹⁶. No enrichments of Na in sulfate-rich veins in the Murray and Stimson formations²³ may also support depletion of Na in post-depositional SO_4^{2-} -rich fluids, if the fluids are in common with Yellowknife Bay²⁴.”

Major Comment 3: Clays in two drilling sites have different basal spacing/composition (one is Na rich and another is Mg-rich). In the main text and Methods, the MS concludes that both composition corresponds to 40-60% of Na-rich clays. I am not the expert, but the methods did not provide a clear explanation of these evaluations. To me, different spacing indicates different composition of clays that formed in different environment before deposition.

Response to Major Comment 3

In the original manuscript, we calculated the compositions of the pore water assuming that the water chemistry of two drilling sites are common. To evaluate this, we have re-calculated the compositions of the pore water for each drilling site using the cationic compositions of smectite's interlayer from the XRD peak intensity of the 001 reflection. To calculate the peak intensity of the 001 reflection, the LP factor (an instrumental factor) is needed. Dr. Elizabeth Rampe, a member of the Curiosity team, kindly has provided the LP factor of Curiosity's XRD (CheMin), which enables us to do this re-calculation. Our new results of the compositions of the pore water are shown in revised Table 1 and Supplementary Fig. 2. Within the errors, both of the pore-water compositions agree each other, supporting that the water chemistry at the two drilling sites is very similar or even common.

We believe that this re-calculation fundamentally improves accuracy of our model and that our estimates of water chemistry make more robust. According to this re-calculation, we have largely re-written subsection of “Interlayer composition of smectite” of Methods of the revised manuscript (see lines 426 – 474 of p. 20-22).

Major Comment 4: The conclusion about the lack of NaCl-rich waters is not supported by data on rate of cation exchange in clays. Fast freezing could have limited ion exchange at low-water conditions.

Response to Major Comment 4

As mentioned in Response to Major Comment 1, the cation exchange reaction

between interlayer of smectite and surrounding water is known to be a rapid and spontaneous process (within a second) (Ogwada and Sparks, 1986; Crooks et al 1993). The rate limiting step of the exchange reaction is diffusion; thus, its activation energy is very low. This means that the exchange reaction proceeds rapidly even at low temperatures (e.g., within a few seconds even at 0°C). Therefore, despite fast freezing of water on early Mars, the estimated water chemistry should reflect the pore water immediately before freezing. As described in Response to Major Comment 1, we have added the detailed descriptions of kinetics of the cation exchange reactions in the revised manuscript (see our Response to Major Comment 1).

Major Comment 5: The title could be changed to reveal a low NaCl-salinity rather than redox processes.

Response to Major Comment 5

We consider that the conclusion of hyposaline early Gale lakes is robust, and agree that including the term of “hyposaline” in the title. We have changed the title to “**Semiarid** climate and **hyposaline lake** on early Mars inferred from water chemistry at Gale”.

Specific Comments-----

Comment 6: Abstract: The abstract is too short to understand what is done. It required editing (see below).

Response to Comment 6:

Due to the word limitation of the abstract (150 words maximum), we unfortunately cannot explain detailed methods and results in the abstract. Within the word limitation, however, we have largely modified the abstract (e.g., by adding more information about alternative scenarios (see Response to Major Comment 1) and by adding concrete values of salinity (see Response to Comment 9 below)). Below in our Responses to Comments 7–12, we show our modifications for the abstract. We hope that this would improve the abstract.

Comment 7: The authors may mention source of data (MSL) and that solid

materials (not water) have been analyzed by MSL. Please explain used methods.

Response to Comment 7

We have added “**using smectite interlayer compositions**” in line 5 of p. 2.

Comment 8: Abstract, line 7: Please specify “Na-Cl concentrations suggest...”. Do you mean MSL data on solid samples or presented numerical models?

Response to Comment 8

We have added “**estimated**” in line 9 of p. 2.

Comment 9

Abstract, Line 8. What is the exact salinity? NaCl could deposit through evaporation of large (not saline) or small (more saline) bodies of water. Formally, the presence of NaCl may not reveal the initial volume of water and initial salinity.

Response to Comment 9

We have added the values of estimated salinity of “**(~0.1–0.5 mol/kg)**” in line 7 in p. 2.

Comment 10

Abstract, Line 9. The text states “observed redox disequilibria in the pore water”. How can disequilibria be observed if nobody analyzed the pore water?

Response to Comment 10

The redox disequilibria are based on the secondary minerals found at Yellowknife Bay (akaganeite, magnetite, and Fe^{2+} saponite). These minerals are in redox disequilibria as shown in Fig. 4. To describe this, we have added “redox disequilibria **in secondary minerals**” in line 11 of the revised abstract.

Comment 11

Abstract, Line 10. These words are not needed “chemistry of the fluids”.

Response to Comment 11

This part has been deleted in the revised manuscript.

Comment 12: Abstract, Line 12; “providing chemical energy” -> “providing chemical sources of energy” (there is no “chemical energy”).

Response to Comment 12

This part has been deleted in the revised manuscript.

Comment 13: Line 33. “mineralogy and geochemistry” -> “phase and chemical composition”

Response to Comment 13

This has been modified as the reviewer suggested (line 43 of p. 4).

Comment 14: Line 36. “probably” deposited {this is still a model interpretation}

Response to Comment 14

This has been modified as the reviewer suggested (line 46 of p. 4).

Comment 15: Line 37 “river” -> “water stream”; “lake” or “lakes”? Not sure we have enough evidence for lake(s) but waters streams (e.g. mud flows) likely occurred.

Response to Comment 15

This has been modified as the reviewer suggested (line 47 of p. 4).

Comment 16: Line 38-40. There are papers showing that clay minerals came with streams as a mud component (not diagenetic). Mud deposits mean deposition from mud (clay-rich) flows. Layers of rocks are compositionally diverse and suggest deposition of different mud/debris flows with a subordinate role of diagenetic processes. But the origin of clay minerals may not be very important for this research.

Response to Comment 16

As mentioned in Response to Major Comment 1, the formation and transport processes of clay minerals do not affect the conclusions of the estimated pore-water

chemistry. This is because the cation exchange reactions are a rapid process. Even in a mudflow, a time for interactions with clay and water would exceed one second. Having said that, we agree the possibility of detrital origin of clay minerals. Thus, we have added the descriptions of the possibility of detrital clay minerals in line 47-49 of p. 4 of the revised manuscript as follows.

“The Yellowknife Bay sediments contain ~20 wt.% of smectite¹¹. These smectites would have formed in the early diagenesis within the lake^{11,14} and/or may have been of detrital origin (Schieber et al 2017) (Fig. 1a).”

Comment 17: Line 44. Is it gypsum in the veins, not anhydrite, CaSO₄ (as in the text). Correct?

Response to Comment 17

The vein contains anhydrite and bassanite, which are considered to be altered from gypsum. We have changed this term to “calcium sulfates” in line 52 of p. 4 and remove the specific chemical composition of “CaSO₄” from the revised manuscript.

Comment 18: Line 48. “chemistries” -> “compositions”

Response to Comment 18

This has been modified as the reviewer suggested (line 58 of p. 4).

Comment 19: Line 51. I would write “contained SO₄²⁻ and Ca²⁺”. Note that Na⁺, Mg²⁺ and Cl⁻ could be much more abundant but corresponding salts did not deposit because of high solubility. Instead, low-solubility Ca sulfate(s) precipitated in fractures.

Response to Comment 19

We have changed the expressions to “provided additional SO₄²⁻” (line 64 of p. 4). In addition, we have added “SO₄²⁻-rich” fluids throughout the revised manuscript.

Comment 20: Line 52. Is it vol. or mass%? Please specify.

Response to Comment 20

The detection of halite (0.1 wt%) in Vaniman et al. (2014) has been recently modified to be less than the detection limit of 1 wt% (Morrison et al., 2018). Thus, this part has been deleted from the revised manuscript.

Comment 21: Line 52. Suggested addition in { }: “this {NaCl-related} salinity”.

Response to Comment 21

This has been deleted from the revised manuscript, according to removal of the description of halite (see above Response to Comment 20).

Comment 22: Line 53-56. Is this conclusion about an early NaCl is based on this paper? If so, it should not be in the introduction. Na-Al correlation is because of Na in in feldspar.

Response to Comment 22

This has been also deleted in the revised manuscript, according to a delete of the description of halite (see above Response to Comment 20).

Comment 23: Line 62. Why “intrusion”. What if (likely) solutions came from above?

Response to Comment 23

We do not restrict the supply process of water of the rewetting event. This part has been modified as follows in the revised manuscript.

“In one, the post-depositional SO_4^{2-} -rich fluids in the last wetting event fully interact with the sediments (the full-interaction scenario),” (line 68-69 of p. 4)

Comment 24: Line 63. Do we have MSL data on last fluid-sediment interaction to constrain models?

Response to Comment 24

Curiosity’s ChemCam provides constraints on the fluid composition of the last wetting event (Nachon et al., 2014). Na contents are almost zero in Ca-sulfate veins with the highest SO_3 contents (Nachon et al., 2014). These results strongly suggest that Na was depleted in the last fluids, compared with the matrix of the Yellowknife Bay mudstone (see our Response to Major Comment 2). To describe this, we have modified this sentence in lines 151 – 153 of p. 9 of the revised manuscript, as follows.

“Additionally, the post-depositional SO_4^{2-} -rich fluids in the last wetting event were depleted in Na^{16} . Na contents are almost zero at sulfate-rich veins with the highest SO_3 contents¹⁶.”

Schmidt et al. (2018) also show that the matrix of the Yellowknife Bay mudstone contains Ca-sulfates (1.4 ± 1.4 wt%) (see line 315-319 of p. 16). This suggests that sulfate-rich fluids in the last wetting event might have intruded into the mudstone matrix. Although these data could support our full-interaction scenario, the Ca-sulfate contents are still low (1.4 ± 1.4 wt%). Thus, we cannot conclude which scenario is the case only based on the Curiosity's data. We discuss both of the full- and no-interaction scenarios.

Comment 25: Line 64, Why “duration” rather than detail MSL data on “around vein” composition?

Response to Comment 25

Curiosity's available data on compositions of Ca-sulfate veins are unable to determine whether sulfate-rich fluids intruded into the mudstone matrix in the last wetting event (see above Response to Comment 24). Thus, we explain the possibility of occurrence of intrusion of sulfate-rich fluids based on the duration of the wetting event.

Comment 26: Line 66. What is “distance”? Do you mean “thickness” of veins?

Response to Comment 26

What we meant here is a distance between two Ca-sulfate veins. We have modified the sentence as follows

“the typical distance **between** the calcium sulfate fractures in the Yellowknife Bay sediments (~ 1 cm)¹³,” (line 73 of p. 5)

Comment 27: Line 65-68. It is not clear how it was evaluated. Did you model diffusion of solution from a fracture to the distance of 1 cm from the fracture? Please clarify in the updated text.

Response to Comment 27

To clarify this, we have added “**through diffusion**” in line 75 of p. 5 as follows in the revised manuscript.

Comment 28: Line 70. Do you really mean solid-solid equilibration? If NOT, consider ions in solution (Ca²⁺).

Response to Comment 28

We consider that pore water is in equilibrium with both of the matrix of the Yellowknife Bay sediments and Ca-sulfate veins (fluid-solid equilibrium). To clarify this, we have modified this sentence as follows in the revised manuscript.

“In the full-interaction scenario, we consider that the pore water was equilibrated with not only the primary components in the matrix of the sediments but also the additional components (such as calcium sulfate) in veins (Fig. 1d).” (lines 76 – 78 of p. 5)

Comment 29: Line 74. I am confused because previous text states that clays formed during diagenesis (not from lake water). But this (line 74) sentence implies that clays chemically precipitated from the lake water. It is unlikely to me and inconsistent with sedimentation of clays on Earth. Clays are typically supplied with streams (e.g., delta deposits). However, chemical modeling could be used to constrain water chemistry at which clays minerals formed at the first place (not in the lake).

Response to Comment 29

We apologize our expression of “we estimate the chemical composition of the lake water” in the original manuscript. This was misleading because we do not estimate directly the lake water chemistry. What we can constrain is the pore-water composition. To avoid this misunderstanding, we have added more detailed descriptions of the theoretical background of our methodology in lines 27 – 38 of p. 3 of the revised manuscript (see Response to Major Comment 1).

Comment 30: Fig. 1. (a) Please mark green and brown forms in the quadrangle. Are any clays in initial sediments? They were likely present (despite several publications). (d) “& fluids” may not be needed (optional). (e) “Ca sulfates”.

Response to Comment 30

We have modified Fig. 1 as the reviewer suggested.

Comment 31: Line 92-93. “provide the constraints of water chemistry of the liquid water” -> “constrain the composition of liquid water”

Response to Comment 31

This has been modified as the reviewer suggested (line 101-102 of p. 7).

Comment 32: Line 92. Gypsum has low solubility and its presence may not indicate composition of “the pore water at the last rewetting event immediately before the disappearance of liquid water”, as stated above. Ion-exchange in smectites is not fast (any data on the rate?). I think, only NaCl will indicate the composition of water “immediately before the disappearance”.

Response to Comment 32

Gypsum is known to be a solubility limiting solid phase in natural aqueous environments if gypsum is present (e.g., Hardie and Eugster, 1970, Min. Soc. Am.). This happens because dissolution/precipitation of gypsum are fast processes (e.g., Mbogoro et al., 2011, J. Phys. Chem. C), and because the solubility of gypsum is low. The low solubility of gypsum allows this mineral to precipitate easily from solutions, so that gypsum can control the concentrations of Ca^{2+} and SO_4^{2-} in the solutions. Dissolution and precipitation of gypsum is a spontaneous process at ambient temperatures (e.g., Mbogoro et al., 2011). Following the above comment, we have cited the previous papers (Hardie and Eugster, 1970, ref. #59) in line 488 of p. 22, which shows that gypsum is frequently a solubility limiting solid phase in solution.

Concerning kinetics of the cation exchange reactions of smectite, there are numbers of literatures that show the cation exchange reaction is fast with low activation energy (Ogwada and Sparks, 1986; Crooks et al 1993; etc.). We have added the description of the kinetics of the cation exchange reactions (see our Response to Major Comment 1 above).

Comment 33: Line 106-107. Possible language problem in this sentence: It is unclear what is “cation fraction of (Na + K) at 0.4 to 0.6”. Changed from what fraction to what fraction? What cations (Mg?) represent the rest (another 0.4-0.6)?

Response to Comment 33

To re-calculate water chemistry for each drilling sites, we have changed the methodology (see our Response to Major Comment 3). According to this change, this part has been deleted from the revised manuscript.

Comment 34: Line 111. What would be an effect of high-salinity solutions?

Response to Comment 34

Salinity can be calculated from the estimated chemical compositions of pore-water from the present modeling approach.

Comment 35: Line 113. Are concentrations of Ca and SO₄ controlled by solubility of gypsum? Are concentration of Na and Cl controlled by solubility of halite/hydrohalite? Are concentration of Mg controlled by solubility of Mg sulfates? What was the salinity?

Response to Comment 35

Owing to fast dissolution and precipitation of gypsum and its low solubility, concentrations of Ca²⁺ and SO₄²⁻ are considered to be controlled by dissolution equilibrium of gypsum (see our Response to Comment 32). However, Na⁺ and Cl⁻ are not controlled by halite owing to high solubility of halite, and Mg²⁺ is not controlled by Mg-sulfate because Mg-sulfate is absent in the sediments. As described in Methods, concentrations of other elements than Ca²⁺ and SO₄²⁻ are determined based on the interlayer compositions of smectite under the assumption of the Na-Cl type solution.

Comment 36: Line 116. What is warm (please specify the temperature)? Why not to have low-T (~0C or below evaporation with or without partial freezing)?

Response to Comment 36

We have added “**above freezing temperature**” in this sentence (line 125 of p. 8).

Comment 37: Line 117. The increase is not “gradual” because saturation of different phases is reached at different concentrations.

Response to Comment 37

The term of “gradually” has been deleted.

Comment 38: Line 118. This statement does not look correct: “salts are most likely to have formed during sublimation after freezing.” Experiments show that salts precipitate sequentially (for example, Ca-sulfate, Mg-sulfate, NaCl, and finally Ca-Mg chlorides). The sequential precipitation was modeled in many “martian” and “terrestrial” papers in which evaporation and freezing was calculated with Pitzer approaches.

Response to Comment 38

This sentence has been deleted.

Comment 39: Line 121. Yes, if evaporation/freezing was slow enough. What do we know about kinetics of Na⁺ ion exchange in smectites? Can we evaluate fate of freezing/evaporation from these data?

Response to Comment 39

There are many literatures on kinetics of the cation exchange reactions. We have added the descriptions of kinetics of the cation exchange reactions in lines 27 – 38 of p. 3 of the revised manuscript (see our Response to Major Comment 1).

Comment 40: Line 121. hydrohalite (in case of freezing)

Response to Comment 40

We consider the condition above freezing temperature (see Comment 36) in these sentences. Thus, we do not add the term of “hydrohalite” in the revised manuscript.

Comment 41: Line 124. Why freezing? Both freezing and evaporation are considered above.

Line 125. This temperature could be estimated with high accuracy with Pitzer modeling of activities.

Line 125. This temperature (0 to -2 C) is for initial ice formation (not complete freezing). Eutectic of Cl-rich solutions would be at much lower temperature (~ -20 or so, depending on the ion composition).

Response to Comment 41

Water-clay interactions would be inhibited at the timing of initial ice formation (0 to -2°C). This is the case because initial ice formation of pore water occurs at interfaces with solid particles (Arenson and Segó, 2006). Arenson and Segó (2006) show that ice formation initiates at the interfaces and that interactions of remaining brine with the solid particles are inhibited due to the ice surrounding the solid particles. To explain this, we have added the following sentences in lines 131 – 136 of p. 8 of the revised manuscript.

“In this case, concentrations of salinity in remained liquid water would have also occurred upon freezing. However, freezing of pore water most likely initiates at the interfaces with solid particles²². This suggests that upon freezing, highly-concentrated brine would be unable to directly interact with smectite particles. Hence, the pore-water chemistry recorded in the Yellowknife Bay sediments most likely represents saturated pore water immediately before freezing in the last wetting event.”

Comment 42: Line 128. “water chemistry” -> “Solution composition”

Response to Comment 42

This has been modified as the reviewer suggested (line 138 of p. 8).

Comment 43: Table 1. Na -> Na+, K-> K+ etc.

Response to Comment 43

The concentrations in Table 1 represent total dissolved components. We have added the following explanations in lines 140 – 141 of p. 8 of the revised manuscript.

“Na, K, Mg, Ca, Cl, SO₄, Cl, ΣCO₂, Fe represent the total dissolved components of sodium, potassium, magnesium, calcium, chlorine, sulfate, dissolved inorganic carbon and iron, respectively.”

Comment 44: Table 1 Concentrations are not defined: It is molality (mol/kg H₂O) or molarity (mol/kg solution)?

Table 1. Is this low SO₄ content corresponds to precipitated gypsum?

Response to Comment 44

It is molality (mol/kg H₂O). We have modified the description in Methods of the revised manuscript (line 507 of p. 23). The estimated SO₄²⁻ concentration is in dissolution equilibrium with gypsum for the estimated Ca²⁺ concentration.

Comment 45: Line 133. English: Dissolved species cannot be components of smectites.

Response to Comment 45

This has been modified as follows in the revised manuscript.

“The dissolved components in Table 1 originate from both the primary components from the pore water in the bottom sediment of the early lake and the additional components from post-depositional fluids in the rewetting events.” (line 144-146 of p. 9)

Comment 46: Line 134. How do we know about “additional” components in smectites? It is probably an assumption. Smectites are different and their composition could (likely to me) reflect formation conditions away from the current deposition place. Otherwise, smectites would have a similar composition.

Response to Comment 46

The reviewer might misunderstand that we used the bulk chemical compositions of smectite to estimate pore-water chemistry. However, this is not correct. As described in our Responses to Major Comment 1, we used only the exchangeable cations in the interlayers of smectite (not the bulk composition). Thus, the additional components can affect the interlayer compositions if post-depositional fluids interacted with clay minerals for one second or longer. To avoid this misunderstanding, we have added the detailed descriptions about the cation exchange reactions of smectite’s interlayers in lines 27 – 38 of p. 2 of the revised manuscript (see our Response to Major Comment 1).

Comment 47: Line 135. I would delete “primary component” to avoid discussion about primary and additional components.

Line 135. Clays formed away from the current deposition place could have exchanged cations with lake water and got more Na than they had. In such a case, Na-rich clays could be considered as “primary” phases formed in situ (in contact with sub-lake waters) from initially deposited clays.

Response to Comment 47

As mentioned above in our Response to Major Comment 1, the cation exchange reactions are a rapid process, so that the additional components from post-depositional fluids can affect the interlayer compositions of clay mineral. We thereby consider that the discussion of the primary and additional components is necessary to interpret the model results. To explain how the primary component would be preserved in the sediments, we have added the following sentences in lines 146 – 155 of p. 9 in the revised manuscript (see our Response to Major Comment 1).

“We believe that the predominant dissolved **components** of Na and Cl in the pore water would have largely originated from the primary component; namely, lake water trapped within the sediments (Fig. 1a, b). This is because **both Na⁺ and Cl⁻ behave as conservative species which are hardly lost in subsequent chemical reactions (e.g., diagenesis), resulting in preservation of information about bottom water within co-buried smectite on Earth¹⁸. Additionally, the post-depositional SO₄²⁻-rich fluids in the last wetting event were depleted in Na¹⁶. Na contents are**

almost zero at sulfate-rich veins with the highest SO₃ contents¹⁶. No enrichments of Na in sulfate-rich veins in the Murray and Stimson formations²³ may also support depletion of Na in post-depositional SO₄²⁻-rich fluids, if the fluids are in common with Yellowknife Bay²⁴.”

Comment 48: Line 152. Another explanation is no lake. In such a case, the estimated water composition (Table 1) corresponds to pore water during formation of clays or during clay-pore water exchange but still without a lake.

Response to Comment 48

The Sheepbed Member is characterized by well-sorted, fine-grain deposits (Grotzinger et al., 2014). These geological characteristics are hard to be explained by turbidity massive mudflows without a lake, because they require settling from suspension in a water or atmospheric column. Suspension in a water column is most likely rather than that in an atmospheric column (e.g., volcanic ash) given existence of secondary minerals within the matrix of the sediments (Vaniman et al., 2014). Since this has been largely discussed previously (e.g., Grotzinger et al., 2014, etc.), and since we have cited these references to explain the geological setting in Fig. 1, we would not explain the possibility of no lakes in this paragraph in detail. However, as the reviewer mentioned, the existence of lake is an assumption. Thus, we have added “if the early lakes existed,” in line 162 of p. 10 of the revised manuscript

Comment 49: Line 199. At what temperature? What is the error bar for the red curve?

Response to Comment 49

We have calculated the water chemistry at 0°C. This information has been added in the caption of Table 1 of the revised manuscript. The errors of pH and total dissolved CO₂ concentrations are given in the range of pH and ΣCO₂ concentration in Table 1.

Comment 50: Line 211. Please check wt% or vol% smectite?

Response to Comment 50

This is wt.%, as described in this sentence.

Comment 51: Line 212. What is “ferrian”? Do you mean “ferric” (= Fe³⁺-bearing).

Response to Comment 51

Yes. The term of “ferrian” is frequently used for Fe³⁺-bearing smectite in the research field of Clay Mineralogy. We have added “(Fe³⁺)” to explain this in the revised manuscript.

Comment 52: Line 214. I would rather suggest formation of Fe³⁺-bearing saponite through alteration of basalt by aerobic waters before sedimentation at the current place.

Response to Comment 52

We consider that ferrian (Fe³⁺) saponite would not have been formed directly by alteration of basalt by aerobic waters. This is because ferrian saponite needs to be formed through oxidation of ferrous (Fe²⁺) saponite (Bristow et al., 2017, Treiman et al., 2014, etc.), which forms under reducing and alkaline aqueous conditions. In other words, formation of Fe³⁺ saponite requires two steps; first, Fe²⁺ saponite forms under reducing conditions, and then, it is oxidation of Fe³⁺ under oxidizing conditions. These two steps cannot be achieved by a single process of alteration of basalt by aerobic waters. To explain these points, we have added the following sentences in line 226 – 229 of p. 12 of the revised manuscript.

“Since Fe³⁺ saponite generally forms through oxidative alterations of reduced ferrous (Fe²⁺) saponite, the formation and deposition of Fe³⁺ saponite through oxidative weathering of basaltic rocks are unlikely. Fe³⁺ saponite needs to have been originally Fe²⁺ saponite in the sediments, whichever it is of diagenesis or detrital origin.”

Comment 53: Line 225. Magnetite and especially pyrrhotite could be primary minerals.

Response to Comment 53

The re-analysis of XRD data analysis of Curiosity’s CheMin shows that the abundance of pyrrhotite at Yellowknife Bay is less than detection limit (<1 wt%) (Morrison et al., 2018, Am. Min.). Thus, we have excluded pyrrhotite from the discussion and Fig. 3. On the other hand, magnetite and saponite could be of detrital origin. To explain this, we have modified the sentence as follows in the revised manuscript.

“Although magnetite and saponite might be of detrital origins, their relatively-high abundance suggests that the pore-water pH would be compatible with their thermochemical stability.” (line 240-242 of p. 13)

Comment 54: Line 231. During early diagenesis or at primary conditions of saponite formation through in situ alteration of basalt (not through digenesis).
Line 233. Not sure that authors of [31] are correct. Magnetite could be (and likely) primary.

Response to Comment 54

See our Response to Comment 55. We have added the possibility of detrital origins of magnetite and saponite in the revised manuscript.

Comment 55: Line 250. This is unclear: “allowing interactions with the atmosphere”.

Response to Comment 55

This has been modified as follows in the revised manuscript.

“so that the fluids can contain the photochemical products.” (line 252-253 of p. 13)

Comment 56: Line 252. They could survive in ice as super-cold liquid acid droplets in the ice and migrate toward the rock (discussed in martian weathering literature).

Response to Comment 56

We have cited the previous paper (Zolotov and Mironenko, 2007: ref. #1) for acid-sediment interactions at low temperatures for this sentence in the revised manuscript.

Comment 57: Line 252. “acidity” -> “protons” (optional).

Response to Comment 57

This has been modified as the reviewer suggested (line 252 of p. 13).

Comment 58: Line 256. In addition, the amount of acids and trapped oxidants could be insufficient to modify the rock-dominated system.

Line 255. Different redox reactions have different rates: some reactions are inhibited but reactions rates with strong oxidants (O₂, for example) are slow but

not negligible at O₂-poor and cold martian environments (Burns, 1993; several recent works).

Response to Comment 58

Given the possibilities suggested by the reviewer above, these possibilities have been added in the revised manuscript, as follows.

“Persistence of the disequilibria in the mineral assemblages strongly suggests that redox reactions in the pore water were very slow because of low temperatures¹¹ and proceeded only partially due to the relatively short existence time of fluids until freezing³⁹, **limited amounts of trapped oxidants, and/or high activation energy of Fe²⁺ oxidation⁴⁰.**” (line 256-259 of p. 13-14)

Comment 59: Freezing is likely but evaporation was mentioned above. Please be consistent.

Response to Comment 59

We have mentioned “sublimation” of H₂O ice after freezing (not evaporation) (see lines 131 – 134 of p. 8 of the revised manuscript). We consider that low-temperature reactions are consistent with our scenario of freezing and subsequent sublimation.

Comment 60: Line 265. “total activities” -> “total concentrations” Please clearly specify used total concentrations of elements.

Response to Comment 60

This has been modified in line 269 of the revised manuscript.

Comment 61: Line 272. I am confused. There is no no-alteration in Fig 1 b. The full alteration scenario was related to the rewetting event (Fig. 1d, Table 1?). Why is the no-interaction scenario related to early diagenesis? Do you mean “no alteration scenario” during early diagenesis? How did clays form? Please explain the meaning of no-alteration scenario. How is it different from another scenario?

Response to Comment 61

As explained in the Introduction section (e.g., lines 67 – 85 of p. 4-5 of the revised manuscript), the no-interaction (not no-alteration) scenario considers that

post-depositional SO_4^{2-} -rich fluids did not chemically interact with the matrix of the Yellowknife Bay sediments; whereas, the full-interaction scenario considers that SO_4^{2-} -rich fluids chemically interacted with the matrix (see lines 67 – 79). In both of the scenarios, clays are considered to have formed during early diagenesis or transported from the source regions. The difference in the full- and no-interaction scenarios is the presence/absence of gypsum in the aqueous systems; that is, whether Ca^{2+} and SO_4^{2-} concentrations are controlled by dissolution equilibrium of gypsum or not (the latter is the case of the no-interaction scenario). Thus, in the no-interaction scenario, the estimated water chemistry reflects the pore water in early diagenesis.

To improve the comprehensibility of the no-interaction scenario to the readers, we have modified the descriptions to explain the no-interaction scenario in more detailed (lines 79 – 85 of p. 5 of the revised manuscript).

“In the no-interaction scenario, we assume that the observed secondary minerals in the matrix of the sediments are not influenced by the post-depositional SO_4^{2-} -rich fluids in the last wetting event (Fig. 1d). In the latter scenario, we estimate the chemical composition of the pore water using only the secondary minerals that are considered to be contained in the matrix of the sediments, i.e., akaganeite. The estimated pore water in the no-interaction scenario would reflect the primary components trapped within the sediments before the last wetting event.”

Furthermore, we think that the fact that an explanation of the no-interaction scenario is described only in Introduction might prevent the comprehensibility of this scenario. To remind the situations of the no-interaction scenario, we have added the following sentences in Section of “the no-interactions scenario” (lines 275 – 283 of p. 14-15). We hope that these revisions improve the comprehensibility of the no-interaction scenario.

“In the no-interaction scenario, we consider that SO_4^{2-} -rich fluids did not chemically interact with the matrix of the sediments (see above). In this case, the composition of pore water that finally interacted with the matrix reflects that in the early post-depositional stages immediately before the disappearance of liquid water (Fig. 1b). In no-interaction scenario, we did not use dissolution equilibrium of calcium sulfate but used the presence of akaganeite in the matrix in addition to the exchangeable cations in the interlayer of the smectite (see Methods).

Comment 62: Line 302. Yes, but not in a closed system with limited oxidants. Diagenetic systems could be semi-closed or closed systems.

Response to Comment 62

This part has been deleted from the revised manuscript due to the updated analysis of Curiosity’s XRD data of no detection of pyrrhotite (Morrison et al., 2018).

Comment 63: Line 305. Martian atmospheric O₂ is not as abundant as on Earth (~5000 times less), so the oxidation rate is correspondingly slower (see Burns, 1993).

Response to Comment 63

Same as above in our Response to Comment 62, this part has been deleted from the revised manuscript.

Comment 64: Line 321. Gypsum or anhydrite?

Response to Comment 64

The vein is composed originally of gypsum. We have mentioned this in line 486-488 of p. 22 of the revised manuscript.

Comment 65: Line 321. Why not to have Ca-sulfate some deposition from primary diagenetic solutions? Sulfates are abundant on Mars and could be present in all solutions.

Response to Comment 65

Because the Ca-sulfate might have been from the primary solution, and because the Ca-sulfate contents are still low ($1.4 \pm 1.4\%$), we cannot conclude that SO₄²⁻-rich fluids intruded into the matrix of the sediments. In this section, we compare the full- and no-interaction scenarios but do not conclude which scenario is the case (as described in line X of the revised manuscript). Given the possibility of Ca-sulfate deposition from primary solution, we have modified this sentence in line 317 – 319 of p. 16, as follows.

“Although this calcium sulfate in the matrix could have been of diagenetic origin, this suggests that SO₄²⁻-rich fluids would have diffused into the matrix of the mudstone in the last wetting event.”

Comment 66: Line 323. Those values could be reconsidered.

Response to Comment 66

These values are estimated based on diffusion of post-depositional fluids into the Yellowknife Bay sediments (see lines 72 – 75 of p. 5 of the revised manuscript). The diffusion theory of liquid into porous media is well established. We used typical values for parameters of permeability of clays (dried marine clays: 10^{-7} – 10^{-10} cm/s from ref. 20) and thickness of clay matrix (1 cm from ref. 11) between Ca-sulfate veins. Although

these values contain uncertainties of orders of two ($1 - 10^2$ yrs), we consider that the real timescale would have been within this uncertainty. In lines 72 – 75 of the revised manuscript, we have clarified that we consider “diffusion” of fluids into the sediments to estimate the timescale (see our Response to Comment 27).

Comment 67: Line 338. I would add “... during the last wetting exclude that affected the sediments.”

The authors could discuss another explanation: Insufficient water-rock interaction during the re-wetting event due to a low water content, low permeability of clay-rich rocks and a short duration/fast freezing. In such a case, the composition of clay minerals would mainly reflect previous environments (an early diagenesis or formation before deposition).

Response to Comment 67

The situation that the reviewer mentioned in this comment is exactly corresponding to the no-interaction scenario (also see Response to Comment 61 above). We have described carefully the situation of the no-interaction scenario in Introduction (lines 79 – 85 of p. 5) and in the section of “the no-interaction scenario” (lines 274 – 310 of p. 15-16). We hope that these revisions improve the comprehensibility of the no-interaction scenario.

Comment 68: Line 334-337. Again, it was unclear how the lake water composition could be evaluated from models that address the re-wetting event.

Response to Comment 68

See Response to Major Comment 2.

Comment 69: Line 353, 364. Evaporation of freezing? Lake’s life time will be longer if the lake is mostly covered by ice.

Response to Comment 69

The duration of lake does not change for a surface frozen lake because sublimation of surface ice occurs. In fact, we consider evaporation or sublimation of H₂O from the lake by assuming surface temperature down to 260–298 K (see Methods). We have modified the sentence as follows, “evaporation (or sublimation if lake surface was frozen)” (line 347 of p. 17 in the main text and line 634 of p. 28 of Methods)

Comment 70: Line 393. disequilibria

Response to Comment 70

This has been modified as the reviewer suggested (line 392 of p. 19).

Comment 71: Line 395-397. Redox interactions decrease amounts of energy for metabolism. The lack of abiotic redox reactions provides sources of energy.

Response to Comment 71

The term of “and biochemistry” has been deleted from the revised manuscript.

Comment 72: Line 422. Note that volcanic SO₂ leads to cooling of Earth’s atmosphere.

Response to Comment 72

Given the possibility of sulfate aerosol formation, the possibility of SO₂ eruption has been deleted from the revised manuscript.

Comment 73: Line 450. Do you mean solubility of Mg-saponite? Mg is metal. Line 451. Do you mean “brucite structural group” of brucite as mineral? I think, solubility of saponite should be discussed, not brucite of “Mg”.

Response to Comment 73

The term of “solubility of Mg” has been modified to “concentration of ΣMg^{2+} ($=\text{Mg}^{2+} + \text{MgOH}^+$)” in line X of the revised manuscript.

At pH > 12, brucite formation prevents high concentrations of Mg^{2+} and MgOH^+ in solutions. At pH ~10–12, Mg-saponite formation prevents high concentrations of Mg^{2+} . This sentence has been modified as follows in the revised manuscript (line 447-449 of p. 21).

“However, concentration of ΣMg^{2+} ($=\text{Mg}^{2+} + \text{MgOH}^+$) also becomes extremely low at high pH (pH > 10) because of the formation of Mg-bearing secondary minerals (saponite and brucite).”

Comment 74: Line 473: Smectites section. The text states “The peak position of the 001 reflection of John Klein is 10.1Å, suggesting that the cation composition of the John Klein smectite could be dominated by Na⁺ and/or K⁺”. I failed to

understand how different clay compositions (Na-rich and Mg-rich with different basal spacing) in John Klein and Cumberland, respectively, were interpreted in terms of “cation fractions of (Na⁺⁺+K⁺) and Mg²⁺ needs to be 0.4-0.6.” in both locations? To me, the simplest explanation of different clay composition is deposition of compositionally different clays with inefficient changes during diagenesis.

Response to Comment 74

As mentioned in our Response to Major Comment 3, we re-estimated the pore-water chemistry for each drilling site (John Klein and Cumberland) using the LP factor of Curiosity’s CheMin provided by Dr. Elizabeth Rampe. The obtained pore-water compositions are consistent each other, suggesting that similar or even common pore water was present at these two drilling sites. Based on this change in methods, we have largely re-written the smectite section in lines 115 – 122 of p. 8 of the revised manuscript (see our Response to Major Comment 3).

Comment 75: Line 466-470. I do not understand the meaning of this text, especially the last sentence.

Response to Comment 75

Due to the change in methods (see our Responses to Major Comment 3 and Comment 74), this part has been deleted from the revised manuscript.

Comment 76: Line 493. Hydrohalite could form upon freezing.

Response to Comment 76

This part has been deleted from the revised manuscript too.

Comment 77: Line 499. “XRD” -> “CheMin”? “XRD” is a method, not an instrument.

Response to Comment 77

This has been modified as the reviewer suggested (line 483 of p. 22).

Comment 78: Line 516. More information would be useful (In Supplementary Materials)

Response to Comment 78

The detailed descriptions on the databased construction has been added in Sec. S2 of revised Supplementary Information. In Methods of the revised manuscript, we have added the following sentence in line 503-505 of p. 23.

“The detailed descriptions for the database construction of akaganeite and ferrihydrite are given in Supplementary Information (Sec. S2).”

Comment 79: Line 519. The presence of abundant amorphous material suggests limited water-rock reactions and water-smectite ion exchange during diagenesis.

Response to Comment 79

This is not true. As mentioned in our Response to Major Comment 1, the cation exchange reactions are a rapid process with low activation energy (order of sub-second) (Ogwada and Sparks, 1986; Crooks et al 1993). On the other hand, even the formation of amorphous materials requires longer times (e.g., hours to years) (e.g., Tosca et al., 2004, JGR). Thus, the presence of abundant amorphous phases does not mean the cation exchange reactions are limited (see Response to Major Comment 1).

Comment 80: Line 523. Several activity coefficients are needed. Do you mean coefficients? What are sources for thermodynamic data for Fe-saponite and ions?

Line 524. mol/(kg H₂O)

Response to Comment 80

Yes, this means activity coefficients. The sentence has been revised as the reviewer suggested. The sources for thermodynamic data except for Fe-saponite, ferrihydrite, and akaganeite come from “thermo.dat” incorporated in GWB. The details of the estimations of thermodynamic data of ferrihydrite and akaganeite are given in Sec. S2 of revised Supplementary Information. The solubility of the Fe-saponite comes from the estimation by Wilson et al. (2006) (ref. #9 in Supplementary Information).

Comment 81: Line 590. Do you mean a re-wetting episode or initial diagenesis?

Response to Comment 81

This is the post-depositional SO_4^{2-} -rich fluids that formed Ca-sulfate veins. We have added the explanations of the no-interaction scenario in the revised manuscript (see our Response to Comment 61). We hope that this revision has improved the manuscript.

Comment 82: Line 609. Released from which mineral? What if Fe-smectite was (likely) deposited with the mudflow?

Line 617. Do not understand why this sentence appears here and do not understand from which mineral Fe+2 is released and how Fe+2 concentrations can "maintain" spacing: "Hence, released Fe $^{2+}$ concentrations must be sufficiently low to maintain the observed basal spacing of the smectites."

Response to Comment 82

Fe $^{2+}$ is released from Fe $^{2+}$ -saponite. The cation exchange reactions are rapid (within a second). Thus, even in a mudflow, Fe $^{2+}$ can be exchanged with cations in the interlayers. We have added the following sentence in line 585 – 587 of p. 26 of the revised manuscript.

“Smectite usually occurs as fine particles and the dissolution rates must be higher than the primary minerals. It can be assumed that the concentrations of Mg and Fe in the solution were equilibrium with respect to saponite.”

Comment 83: Line 613. What is the oxidant if only water is present?

Line 612-614. Why is it oxidized? Are there enough oxidants to oxidize abundant Fe $^{2+}$ in saponite? The spacing tells us that Fe $^{3+}$ is not abundant in smectite.

Response to Comment 83

Oxidation of Fe $^{2+}$ in the interlayer of smectite proceeds rapidly by atmospheric O_2 . In addition, atmospheric chemistry on Mars produces other reactive oxidants, such as O_3 and H_2O_2 (Lasne et al., 2014). In the original manuscript, we assume that Fe $^{3+}$ in the interlayer of saponite to interpret the XRD data. However, the new methodology employed in the revised manuscript does not require this assumption. Thus, we have deleted this part from the revised manuscript.

Comment 84: Line 621 Is albite really detected? Thermodynamically, analcime could be more appropriate.

Response to Comment 84

This is based on the interpretation of Bristow et al. (2015). A recent paper on re-analysis of Curiosity's XRD data (Morrison et al., 2018, Am. Min) also supports this interpretation.

Comment 85: Line 647. What is the value of K(21) and how it was obtained?

Response to Comment 85

We have added the detailed descriptions of our database construction of akaganeite in Sec. S2 of Supplementary Information.

Comment 86: Supplementary material: Line 200. Concentration of O₂ in dissolved water would reflect the low O₂ level of the martian atmosphere. The authors could evaluate the rate of pyrrhotite oxidation by the low martian O₂ = 0.0014x0.006 bar compared to 0.2 bar O₂ on Earth.

Response to Comment 86

According to the re-analysis of XRD data by Curiosity, the abundance of pyrrhotite becomes below the detection limit (1 wt%) (Morrison et al., 2018). Thus, this part has been deleted from the revised manuscript.

Comment 87: Line 204. Please remind about Eh >0.5 V. It is related to akaganeite stability? Do we know thermodynamic data for akaganeite. What is the uncertainty?

Response to Comment 87

This is based on the stability of akaganeite. The descriptions on the methodology and uncertainty have been described in Sec. 2 of revised Supplementary Information. Due to the re-calculation of pore-water compositions for each drilling site, a lower limit of Eh has been modified as > +0.3 V. However, the conclusions of need of high Eh oxidants are not changed.

Responses to comments from Dr. Elizabeth B. Rampe (the reviewer #5) for ‘Semiarid climate and hyposaline lake on early Mars inferred from water chemistry at Gale’ by Fukushi, K. et al. (MS# NCOMMS-18-21962A)

We are grateful to Dr. Elizabeth B. Rampe for the constructive comments and suggestions. In particular, she has kindly provided the LP factor of Curiosity’s CheMin, which has greatly improved the estimate of pore-water chemistry at the drilling sites. Below, we give our responses in turn following each comment, with the reviewers’ comments being in Arial font and our responses being in Times font.

Major Comment 1: Some of the mineralogical inputs into the models are no longer valid. One important constraint that Fukushi et al. use for their models of pore water chemistry is the presence of 0.1-0.4 wt.% halite in the Yellowknife Bay sediments. Vaniman et al. (2014) Science indeed reported 0.1 wt.% halite in John Klein and Cumberland based on refinements of CheMin data (I don’t know the source of the reported 0.4 wt.% halite), but the CheMin team has recently applied an important correction to the patterns collected earlier in the mission and re-analyzed these corrected patterns. Machining offsets in the sample wheel result in differences of 10s of microns from the ideal diffraction position, and a linear relationship between plagioclase unit-cell parameters has allowed the team to account for these differences in diffraction position (Morrison et al., 2018). The analysis of the corrected patterns of John Klein and Cumberland showed no halite or pyrrhotite above the detection limits of CheMin (~1 wt.%). There could be halite present in abundances <1 wt.%, and I suggest the authors re-frame their arguments and calculations with this new constraint. The authors may also use akaganeite to constrain salinity of diagenetic fluids because the identification of akaganeite is robust.

Response to Major Comment 1

We appreciate the reviewer for the updated information on re-analysis of Curiosity’s CheMin data (Morrison et al., 2018). According to the no definitive detection of halite and pyrrhotite, we have modified the manuscript concerning the following two points; 1) instead of halite, we have used the presence of akaganeite and saponite to constrain the type of water chemistry of pore water, and 2) without pyrrhotite, we have re-calculated Eh-pH stability diagram for the estimated pore-water compositions. As shown below, we can infer a Na-Cl type solution for the pore water

based on the presence of akaganeite and saponite. Even without pyrrhotite, we can also show the occurrence of redox disequilibria of the secondary minerals (akaganeite, magnetite, and Fe^{2+} saponite) in the Yellowknife Bay sediments. Thus, our conclusions do not change significantly according to no definitive detection of halite and pyrrhotite. Below, we describe the detailed modifications concerning the above two points.

Concerning the point 1) of no definitive detection of halite, we have used akaganeite to constrain the predominant anion of pore water. Peretyachko et al. (2018 JGR Planet) shows that formation of akaganeite requires high Cl^- concentrations (i.e., $\text{Cl}^- > 0.05 \text{ mol/kg}$) in solutions. In addition, another experimental study indicates that Cl^- in akaganeite can be replaced with other anions, such as SO_4^{2-} and OH^- , when it interacts with solutions dominated with those anions (Peretyachko et al., 2019 ACS Earth & Space Chem.). These experimental studies indicate that the existence of Cl^- -type akaganeite in the Yellowknife Bay sediments requires Cl^- -dominant solutions during formation and diagenesis (Peretyachko et al., 2019 ACS Earth & Space Chem.). In other words, Cl^- should be the predominant anion in the pore water that finally interacted with the Yellowknife Bay sediments. To explain this point, we have added the following sentences in line 493 – 497 of p. 22-23 of the revised manuscript.

“Given the fact that akaganeite can accommodate the anions other than chloride, the presence of Cl^- type akaganeite in Yellowknife Bay strongly suggests that the akaganeite was in contact with Cl^- -dominant solutions compared with other anions, such as sulfate⁶⁰. By combining the dominant cation type of the water constrained from the presence of saponite, the original pore water in the sediments would be Na- Cl water type.”

We have also used the presence of saponite to constrain the predominant cation of the pore water. We have reviewed formation conditions of terrestrial saponite (e.g., Bristow et al., 2009) (also see new Supplementary Table 1). As far as we know, saponite deposition occurs only in saline lakes, low-temperature upwelling solutions from seawater, and modern oceans. All of these aqueous environments are characterized as Na- Cl type solutions with relatively high salinity (hyposaline to saline) (see new Supplementary Table 1). In addition, saponite is stable only under neutral to alkaline pH conditions (Fig. 4). Under such high pH conditions, concentrations of divalent cations, such as Mg^{2+} and Ca^{2+} , must be low due to precipitations of secondary minerals (Drever, 1996), such as Mg-Ca phyllosilicates, $\text{Mg}(\text{OH})_2$, MgCO_3 , CaCO_3 , etc. This leads to the dominance of Na^+ in high pH solutions. To discuss this briefly, we have added the subsection of “saponite” (see the following sentences) in Methods in lines 475–482 of p. 22 of the revised manuscript.

“Under terrestrial conditions, saponite depositions are observed in saline lakes,

low-temperature upwelling solution from seafloor, and modern oceans (Supplementary Table 1). The pH of the solutions is commonly neutral to alkaline. The predominate cations in these natural waters are always Na^+ rather than Mg^{2+} because saponite is thermodynamically stable at high pH (Fig. 3). Under such alkaline conditions, concentrations of divalent cations, such as Mg^{2+} and Ca^{2+} , must be very low due to formation of Mg/Ca-bearing secondary minerals²¹. Consequently, the low concentrations of divalent cations lead to the dominance of Na^+ in the solutions. Thus, the presence of saponite in the Yellowknife Bay sediments strongly suggests that the predominant cation in the pore water is most likely Na^+ .”

Regarding the point 2) of no definitive detection of pyrrhotite in the sediments, we have re-calculated the stability of Fe-bearing secondary minerals without pyrrhotite (see new Fig. 4 of the revised manuscript). Even when pyrrhotite is absent, the co-existence of akaganeite, magnetite, and Fe^{2+} saponite is in redox disequilibria. Thus, our conclusion of redox disequilibria in the Yellowknife Bay sediments is not changed. Due to no definitive detection of pyrrhotite, we have deleted discussion of oxidative weathering of detrital pyrrhotite in the original manuscript.

Major Comment 2: The assumption that John Klein and Cumberland were altered by diagenetic fluids with the same composition, resulting in similar interlayer cation compositions, warrants further justification. ChemCam data from raised ridges in Yellowknife Bay demonstrate that these features are enriched in MgO by 1.2-1.7 times over that of the surrounding mudstone (Leveille et al., 2014). This demonstrates that there is geochemical heterogeneity in the mudstone, which may have originated from diagenetic fluids altering different portions of the sediments. The authors should take these observations into consideration for their conceptual models.

Furthermore, the basal spacings of cation-exchanged smectite measured in the laboratory do not exclusively support the authors' hypothesis that the smectite in John Klein and Cumberland have similar compositions. Supplementary Figure 3c demonstrates that smectite with Na+K/Mg ratios of 0.4-0.6 show differences in their d001 of ~3 angstroms (similar to that observed for John Klein and Cumberland smectite). Supplementary Figure 3c, however, reports basal spacings of smectite measured at 40% relative humidity, rather than ~0% relative humidity, as the samples experience in the CheMin instrument on Curiosity. This difference in basal spacings of ~3 angstroms at 40% RH may not hold true at 0% RH. The authors should further justify their assumption that John Klein and Cumberland were altered by the same diagenetic fluids in light of this. Supplementary Figure 3a shows basal spacings for cation-exchanged smectite

at 0% RH, which seems to demonstrate that the interlayer sites of the smectite in John Klein must be dominated by Na⁺ and/or K⁺, while those of the smectite in Cumberland must be dominated by Ca²⁺ and/or Mg²⁺. Therefore, the alternate hypothesis that the smectite in John Klein and Cumberland have different interlayer cation compositions is also valid.

Response to Major Comment 2

In the revised manuscript, we have re-calculated the composition of the pore water for each drilling site (John Klein and Cumberland) using the cationic compositions of smectite's interlayer based on the XRD peak intensity of the 001 reflection. To calculate the peak intensity of the 001 reflection, the LP factor (instrumental factor) is needed. The reviewer has kindly provided the LP factor of Curiosity's CheMin, which enables us to do this re-calculation. Our new results are shown in revised Table 1 and Supplementary Fig. 2 (Supplementary Fig. 2 is attached in this letter below). Within the errors, both of the pore-water compositions agree each other, supporting the original assumption that the water chemistry at the two drilling sites was common.

We believe that this re-calculation fundamentally improves the accuracy and reliability of our model. Thanks to the reviewer's kind help, we consider that our estimates of water chemistry become more robust. According to this re-calculation, we have largely re-written subsection of "Interlayer composition of smectite" of Methods of the revised manuscript (see lines 426 – 474 of p. 20-22). The descriptions on the results of water chemistry have been modified as follows, in lines 115 – 122 of p. 7-8 of

the revised manuscript.

“Table 1 and Supplementary Fig. 2 show the compositions of the pore water that last interacted with the Yellowknife Bay sediments for the full-interaction scenario. The concentrations of all components from the Cumberland site overlap those from the John Klein site (Supplementary Fig. 2). This suggests that the pore-water compositions at the two spatially-closed sites would have been very similar or even common. This is consistent with a view of the full-interaction scenario that the post-depositional fluids infiltrated into the sediments. Since the pore-water composition from the Cumberland site is more constrained than those from John Klein site, the former would be the representative water chemistry of the pore water prevailing in the Yellowknife Bay sediment.”

Minor Comments-----

Comment 3: Line 35: Suggested edit: delete “spatially enclosed” or expand upon it because I don’t know what it is referring to. Is the Yellowknife Bay spatially enclosed or are the drill core samples?

Response to Comment 3:

What we meant in the original sentence is that the locations of two drilling core sites (John Klein and Cumberland) are close with the direct distance of ~3 m (Grotzinger et al., 2014). To avoid confusion, we have deleted this expression from the revised manuscript.

Comment 4: Line 37: Replace “the lake” with “a lake” because there may have been many on the crater floor.

Response to Comment 4

This has been modified as the reviewer suggested (line 47 of p. 4).

Comment 5: Line 38-40: The bulk mineralogy does not suggest that most of the chemical reactions generated phyllosilicates during early diagenesis. The similarity of the bulk composition of the Yellowknife Bay sediments to the composition of the bulk martian crust suggests early diagenesis in a closed system (e.g., McLennan et al., 2014). I suggest this sentence be re-worded.

Response to Comment 5

We agree the possibility that clay minerals found at Yellowknife Bay are of

detrital origin. Because we used the interlayer compositions of smectite to constrain the water chemistry of pore water, and because the cation exchange reactions between the interlayers and surround water are a rapid process (order of sub-second), our results of pore-water chemistry do not depend on the formation condition of smectite. In the revised manuscript, we have added the descriptions of the possibility of detrital origin of phyllosilicate as follows.

“The Yellowknife Bay sediments contain ~20 wt% of smectite¹². These smectites would have formed in the early diagenesis within the lake^{12,13} (Fig. 1b) and/or may have been of detrital origin³ (Fig. 1a).” (lines 47 – 49 in p. 4)

“Fe³⁺ saponite needs to have been originally Fe²⁺ saponite in the sediments, whichever it is of diagenesis or detrital origin.” (lines 228 – 229 in p. 12)

“Although magnetite and saponite might be of detrital origins, their relatively-high abundance suggests that the pore-water pH would be compatible with their thermochemical stability.” (lines 240 – 242 in p. 13)

Comment 6: End of line 45: I suggest a brief description of Aeolis Mons (e.g., “These rewetting events occurred within Gale even after the formation of Aeolis Mons, a 5 km mound of layered sedimentary rock in Gale crater.”).

Response to Comment 6

This has been modified as the reviewer suggested (line 54-55 of p. 4).

Comment 7: Supplementary Figure 1: I disagree that the correlation of Al and Na demonstrates that Na in the final pore water was derived from the paleo lake water. The authors use this figure to justify some assumptions in their geochemical models in their response to the reviewers. The majority of the Na in the mudstone is likely in feldspar, and the correlation of Al and Na may be a consequence of abundant feldspar in the mudstone. Again, CheMin does not detect halite so this phase cannot be used to constrain Na in the pore water. A correlation between Na and Cl is necessary to demonstrate the presence of halite in the mudstone. However, the 10 angstrom basal spacing for smectite in John Klein suggests Na may be present in the interlayer site. The authors may use this constraint to characterize Na in the pore water.

Response to Comment 7

We agree that the correlation of Al and Na in original Supplementary Fig. 1 does not always mean that Na in the pore water was derived from the sediments. Following

the reviewer's comment, we have deleted original Supplementary Fig. 1 and the related descriptions in the revised manuscript. However, we still consider that post-depositional sulfate-rich fluids would not be a source of Na, as Na is depleted in Ca-sulfate veins (Nachon et al., 2014 JGR). The NaO contents are almost zero at Ca-sulfate veins with the highest SO₃ contents (Nachon et al., 2014). In addition, Yen et al. (2017 EPSL) also show no enrichments of Na in veins and slight leaching of Na from the Murray and Stimson formations. If the post-depositional sulfate-rich fluids were in common with Yellowknife Bay (Rampe et al., 2017), the sulfate-rich fluids would not be a source of Na. To explain this, we have modified the sentence as follows, in lines 151 – 155 of p. 9 of the revised manuscript.

“Additionally, the post-depositional SO₄²⁻-rich fluids in the last wetting event were depleted in Na¹⁶. Na contents are almost zero at sulfate-rich veins with the highest SO₃ contents¹⁶. No enrichments of Na in sulfate-rich veins in the Murray and Stimson formations²³ may also support depletion of Na in post-depositional SO₄²⁻-rich fluids, if the fluids are in common with Yellowknife Bay²⁴.”

Comment 8: Line 64: There is a reference for Figure 1f, but this doesn't exist.

Response to Comment 8

We appreciate the reviewer's careful check. Fig. 1f is not correct. We have changed to Fig. 1e in the revised manuscript (line 71 of p. 5).

Comment 9: Line 92: Change “presences” to “presence.”

Response to Comment 9

This has been modified as the reviewer suggested (line 101 of p. 7).

Comment 10: Line 92. Also, the assumption of gypsum in Yellowknife Bay sediments should be made here because CheMin did not detect any gypsum in John Klein or Cumberland. CheMin detected bassanite and anhydrite, and bassanite commonly forms from the dehydration of gypsum. This did not occur within the CheMin instrument as reported in subsequent samples by Vaniman et al. (2018). Gypsum dehydrates to bassanite in CheMin over the course of ~35 sols. CheMin did not observe gypsum in the first analyses of John Klein or Cumberland.

Response to Comment 10

Agreed. The following sentence of the assumption of the presence of gypsum has been added in Methods (lines 485 of p. 22).

“Bassanite is considered to be a likely product of gypsum ($\text{CaSO}_4 \cdot 2\text{H}_2\text{O}$) dehydration⁵⁸.”

Comment 11: Line 163: Without petrographic information, the association between smectite and akaganeite and their presence in the matrix cannot be confirmed. Suggested edit: “Furthermore, akaganeite, a chloride-bearing ferric oxyhydroxide, occurs with smectite in the Yellowknife Bay mudstone.”

Response to Comment 11

We appreciate the reviewer’s helpful editing. We have modified the sentence as the reviewer suggested (line 181-182 of p. 10).

Comment 12: Line 193: There is not a complete lack of carbonate in the Yellowknife Bay sediments. Ming et al. (2014) Science report on the evolution of gases from John Klein and Cumberland in the Sample Analysis at Mars (SAM) instrument. They identify CO_2 in abundances of up to 0.8 wt.% that could be from Fe-/Mg-carbonates and organics. This should be used as an upper limit for carbonate abundance in Yellowknife Bay.

Response to Comment 12

We appreciate again the reviewer for the updated analyses on Curiosity’s CheMin data. This sentence has been modified by adding an upper limit of carbonate content as follows (line 191-192 of p. 11). Ming et al. (2014) has been also added in the reference list.

“However, calcium carbonates are very low (< 0.8 wt%) in the Yellowknife Bay sediments^{12,32,33}.”

Comment 13: Line 211: Delete “of” to read “~20 wt.% smectite.”

Lines 225, 296, 298: Delete pyrrhotite.

Response to Comment 13

These have been deleted from the revised manuscript.

Comment 14: Line 237: Rampe et al. (2017) reported on acidic alteration in the

Murray formation. Yen et al. (2017) reported on acidic fluids in the Stimson formation. I suggest citing Yen et al. here, too.

Response to Comment 14

We have added Yen et al. (2017) here in the revised manuscript.

Comment 15: Line 251: Suggest replacing “fogs” with “fog”.

Response to Comment 15

This has been modified as the reviewer suggested (line 253 of p. 13).

Comment 16: Line 258: Suggest replacing “alterations” with “alteration” or “episodes of alteration”.

Response to Comment 16

This part has been modified to “**episodes of alteration**” as the reviewer suggested (line 262 of p. 14).

Comment 17: Lines 301-307: With the absence of pyrrhotite, there is no need to explain the coexistence of akaganeite with pyrrhotite.

Response to Comment 17

We have deleted the discussion on the co-existence of akaganeite and pyrrhotite from the revised manuscript. However, we still consider that redox disequilibria between akaganeite and Fe^{2+} saponite need to be explained in the no-interaction scenario. We have added the following sentences in lines 303 – 310 of p. 15-16 of the revised manuscript.

“One possibility to generate the redox disequilibria in the no-interaction scenario is interactions of upwelling groundwater at Gale¹⁹ with initially oxidized sediments¹⁰ after deposition. Groundwater within Martian crusts would become reducing and alkaline¹⁰. If reducing groundwater upwelled into oxidized sediments at Gale, Fe^{2+} saponite would have formed through reduction of oxidized smectite, e.g., nontronite, by groundwater. Akaganeite in the sediments could have been also reduced into magnetite by groundwater; however, this conversion might have proceeded only incompletely due to kinetics. The occurrence of the redox disequilibria due to groundwater upwelling would be also applicable to the full-interaction scenario”

Comment 18: Line 343: Suggest deleting “formation”

Response to Comment 18

This has been deleted from the revised manuscript.

Comment 19: Line 358: Cite a reference for an ancient martian ocean.

Response to Comment 19

The discussion on aeolian NaCl supply has been added in response to a previous reviewer's comment. However, we consider that the aeolian NaCl supply from a Martian ocean would be implausible since 1) there is no evidence of a saline Martian ocean (lack of salt depositions within potential shorelines), and 2) there is no need of co-existence of ocean and early Gale lakes. Thus, we have deleted this sentence from the revised manuscript.

Comment 20: Line 361: Cite a reference for the difficulty in achieving higher-temperature conditions with climate models.

Response to Comment 20

We have cited Wordsworth et al. (2017) (ref. #44) in the revised manuscript.

Comment 21: Line 364: I suggest using a different word than "precipitation" because fluids may not have entered Gale crater via rain or snow. Perhaps state that "evaporation dominated fluid input."

Response to Comment 21

Agreed. We have modified this sentence as the reviewer suggested (line 360 of p. 18).

Comment 22: Line 372: Some Japanese characters snuck into the manuscript.

Response to Comment 22

We apologize for this. This has been deleted from the revised manuscript.

Comment 23: Line 406: Instead of a low PCO₂ at the time of the last wetting event at Gale, could the relative absence of carbonates indicate that the diagenetic fluids were not in equilibrium with the CO₂ in a thicker atmosphere?

For example, if fluids were sourced from rapid melting of ice or sourced from the subsurface, perhaps the fluids did not have ample time to interact with the CO₂ in the atmosphere to dissolve substantial CO₂.

Response to Comment 23

This is a reasonable alternative possibility. We have added the following sentence to explain this possibility in line 405 – 409 of the revised manuscript.

“Alternatively, dissolution equilibrium between post-depositional SO₄²⁻-rich fluids with atmospheric CO₂ was not achieved (e.g., due to rapid melting of ice). Drawdown of CO₂ from fluids during transportation to Yellowknife Bay via carbonate formation is unlikely given the acidity of fluids. Based on a lower limit of pH, we propose an upper limit of P_{CO_2} of ~100 mbar if atmospheric CO₂ was in equilibrium with the fluids (Table 1).”

Comment 24: Lines 412-414: The estimated timing of this last wetting event (Amazonian) is consistent with the youthful age of jarosite in the Mojave2 sample measured from the Murray formation stratigraphically above the Yellowknife Bay formation. Martin et al. (2017) report on the K-Ar age of jarosite measured by the SAM instrument and find the age is 2.12 +/- 0.36 Ga. The authors may choose to cite this to further support the timing of the last acidic-oxidizing aqueous event in Gale crater.

Response to Comment 24

Again, we appreciate the reviewer for the updated information on the K-Ar dating of sulfate-rich veins (Martin et al., 2017). We have added the following sentence in line 415 – 417 of p. 20 of the revised manuscript.

“The estimated timing of this last wetting event is consistent with the youthful age of jarosite in the Mojave2 sample ($2.12 \pm 0.36 \text{ Ga}$)¹⁷.”

Comment 25: Line 417: “volcanoes” is misspelled.

Response to Comment 25

This has been deleted in the revised manuscript.

Comment 26: Line 492 (section about halite): Delete this section because CheMin does not positively identify halite.

Response to Comment 26

We have deleted the section of “halite”.

Comment 27: Lines 499, 502: “bassanite” is misspelled.

Response to Comment 27

We appreciate the reviewer’s careful check. These have been modified in the revised manuscript.

Comment 28: Lines 501-502: Suggestion to delete the sentence stating bassanite is a product of gypsum dehydration in CheMin because that was not observed for the John Klein and Cumberland samples. (That behavior was observed for some samples in the Murray formation.)

Response to Comment 28

We have deleted “within the low humidity conditions inside the CheMin instrument” in the revised manuscript, as the reviewer suggested.

Responses to comments from Dr. Edwin Kite (the reviewer #6) for ‘Semiarid climate and hyposaline lake on early Mars inferred from water chemistry at Gale’ by Fukushi, K. et al. (MS# NCOMMS-18-21962A)

We are grateful to Dr. Edwin Kite for the constructive comments and suggestions. Below, we give our responses in turn following each comment, with the reviewers’ comments being in Arial font and our responses being in Times font.

Major Comment 1: The workflow hinges on halite. But there is no definite detection of halite by CheMin at YKB (Vaniman et al. Science 2014). Indeed, none of the references on line 92 supplied to support the “presences of [...] halite” actually claim a halite detection. This is not a big problem in itself, because halite could be present below detection limit. However, Cl could be hosted by oxychlorine species (Sutter et al. International Journal of Astrobiology 2017, Hogancamp et al. JGR-Planets 2018). See especially Section 4.9 and Fig. 18 of Sutter et al JGR-Planets 2017, and Section 4.1 of Farley et al. EPSL 2016, for their discussion. What would happen to the conclusions of the manuscript if say 25% of the Cl was hosted by oxychlorine species (chlorate/perchlorate)? What if 75%?

Response to Major Comment 1

Concerning the presence of halite, the reviewer #5 has provided the new information of re-analysis of Curiosity’s data (Morrison et al., 2018 Am. Min.). This paper shows that halite contents in the Yellowknife Bay sediments are lower than the detection limit (< 1 wt%). In the revised manuscript, thereby, we have re-framed the arguments to constrain the predominant anion of pore water by using the presence of akaganeite.

We agree that oxychlorine may have existed in the pore water with Cl⁻. In our calculations, a concentration of Cl⁻ represent total chloride concentration (= {Cl⁻} + {ClO₄⁻}). Thus, the estimated cation compositions, salinity, and pH do not change even if oxychlorine existed together with Cl⁻.

We consider that Cl⁻ concentration would have been higher than ClO₄⁻ in the pore water (i.e., Cl⁻ dominant) based on the presence of akaganeite in the sediments (Peretyazhko et al., 2018, 2019). Peretyachko et al. (2018 JGR Planet) shows that formation of akaganeite requires high Cl⁻ concentrations (i.e., Cl⁻ > 0.05 mol/kg) in solutions. In addition, their follow-up experimental study indicates that Cl⁻ in akaganeite

can be replaced with other anions, such as SO_4^{2-} and OH^- , when it interacts with solutions dominated with those anions (Peretyachko et al., 2019 ACS Earth & Space Chem.). The series of the experimental studies indicate that the existence of Cl-type akaganeite in the Yellowknife Bay sediments requires Cl^- -dominant solutions (i.e., $\{\text{Cl}^-\} > \{X\}$; X = other anions including ClO_4^-) (Peretyachko et al., 2019 ACS Earth & Space Chem.). In other words, Cl^- should be the predominant anion in the pore water that finally interacted with the Yellowknife Bay sediments. To explain this point, we have added the following sentences in line 493 – 497 of p. 22-23 of the revised manuscript.

“Given the fact that akaganeite can accommodate the anions other than chloride, the presence of Cl type akaganeite in Yellowknife Bay strongly suggests that the akaganeite was in contact with Cl-dominant solutions compared with other anions, such as sulfate⁶⁰. By combining the dominant cation type of the water constrained from the presence of saponite, the original pore water in the sediments would be Na-Cl water type.”

To explain our results of the cation compositions, pH, and salinity do not change even when ClO_4^- was present in the pore water, we have modified the sentences in the akaganeite section of revised Methods as follows (lines 520 – 525 of p. 23-24).

“According to the present approach to constrain major components, the chemical behavior of ClO_4^- can be treated as same as that of Cl^- . This means that the concentrations of Cl^- obtained from the present study actually represent the concentrations of total chloride ($\{\text{Cl}^-\} + \{\text{ClO}_4^-\}$) and that the results of cation compositions, pH, and salinity of the pore water do not change even when the pore water contained ClO_4^- . However, the presence of the Cl bearing akaganeite in the Yellowknife Bay sediments suggests the concentration of Cl^- must be higher than that of ClO_4^- (ref. ⁶⁰).”

Major Comment 2: The lake-lifetime calculation assumes that the Na is leached from the catchment soil/rock by the same wet event that filled the lake. Thus, a lake lifetime of $>10^4$ years is obtained. But this need not be true; during a long cold period, large quantities of volcanic acids (e.g. HCl) can interact with the soil, forming salts that are quickly flushed out during much-more-brief surface-runoff events. This second scenario is proposed for Mars' THEMIS detected halide lake deposits by Melwani Daswani & Kite (JGR, 2017). Therefore, the lake-lifetime constraint does not follow from the data presented (without additional unstated assumptions).

Response to Major Comment 2

This is an interesting possibility. As the reviewer mentioned, accumulation of volcanic acids should have decreased solution pH and increased the rate of chemical

weathering. Previous laboratory experiments investigated the rate of chemical weathering from synthesized Martian basalts using various acidic solutions (Tosca et al., 2004; Hurowitz et al., 2005). They show that the dissolution rate of Na increases in a factor of ~10 from pH 4 (pH equilibrated with a few bar of CO₂) to pH 1 (solutions with volcanic acids) (Tosca et al., 2004; Hurowitz et al., 2005). However, acidity of the solutions would be neutralized through reactions with rocks in relatively short time (e.g., ~10² years) (Zolotov and Mironenko, 2007). This timescale is significantly shorter than the estimated total duration of the early Gale lakes (~10⁵–10⁶ years). Thus, although accumulation of volcanic acids would have increased an input flux of Na into the lakes, order of total duration would not change dramatically. To discuss this possibility, we have added the following sentences in lines 353 – 357 of p. 17 of the revised manuscript.

“Accumulation of volcanic acids during short-term intermittency of lakes would decrease solution pH and, thus, increase dissolution of Na from rocks². However, acidity of solutions would be neutralized within a short time (~10² years) via interactions with rock¹. Thereby, the estimated total duration of 10⁴–10⁶ years would not be dramatically reduced by accumulation of volcanic acids.”

Major Comment 3: Schieber et al. Sedimentology 2017 provide a detailed argument (abstract, and their section “Perspectives on the origin of clays”) that at least some of the clays are detrital. This contradicts line 38 of the manuscript. Please explain in the manuscript the response to Schieber et al.’s argument.

Response to Major Comment 3:

We agree the possibility that some clay minerals found at Yellowknife Bay may be of detrital origin. We have added the descriptions of the possibility of detrital origin of phyllosilicates found at Yellowknife Bay in the revised manuscript.

“The Yellowknife Bay sediments contain ~20 wt% of smectite¹². These smectites would have formed in the early diagenesis within the lake^{12,13} (Fig. 1b) and/or may have been of detrital origin³ (Fig. 1a).” (lines 47 – 49 in p. 4)

“Fe³⁺ saponite needs to have been originally Fe²⁺ saponite in the sediments, whichever it is of diagenesis or detrital origin.” (line 228 – 229 of p. 12)

“Although magnetite and saponite might be of detrital origins, their relatively-high abundance suggests that the pore-water pH would be compatible with their thermochemical stability.” (lines 240 – 242 in p. 13)

Even if clay minerals are of detrital origin, this does not change our results of pore-water chemistry. This is the case because we have used the interlayer compositions of smectite to constrain the water chemistry of pore water (not the bulk composition of

smectite). Although the bulk composition of smectite can be altered in a long time (decades or longer), the cation exchange reactions between the smectite interlayer and surrounding water proceed instantly even at low temperatures (order of sub-second) (Ogwada and Sparks, 1986; Crooks et al 1993). In other words, regardless of formation conditions of clay minerals (detrital or diagenesis), smectite's interlayer can record the cations of surrounding water that finally interacted with the clay rocks. In the revised manuscript, we have added the descriptions that our estimate of water chemistry does not depend on the formation conditions of smectite, as follows.

“Smectite is unique in possessing exchangeable cations in interlayers⁷. Although there is a wide diversity of mineralogical characteristics of smectites including the chemical compositions and formation processes, all smectite possess the cation exchange ability (Supplementary Fig. 1). The cation exchange reaction of smectite is an instantaneous process with sub-second timescale in suspension solutions^{8,9}. The rate limiting step of the exchange reaction is diffusion and, thus, the activation energy is very low^{8,9}. Therefore, the exchangeable cations in smectite can readily record the surrounding solution composition even at low temperatures (e.g., ~0°C)⁴⁻⁶. On Mars, there is a debate about the formation process of smectite in sediments, i.e., *in-situ* formation by early diagenesis¹⁰⁻¹³ vs. detrital deposition³. However, regardless of the chemical compositions and formation conditions of smectite, the top-down approach can provide a robust “snapshot” of the chemical properties of the pore water that finally interacts with the clay rocks⁴ if the clay rocks contain smectite.” (lines 27 – 38 of p. 3)

Line-by-line comments-----

Comment 4: Abstract. Suggest explain methods (one or two sentences). Suggest clarify that conclusions are “within our model assumptions”.

Response to Comment 4

Due to the word limitation (150 words or less), we have added “using smectite interlayer compositions” in line 5-6 and “To interpret this, multiple scenarios for post-depositional alterations are considered.” in line 7-8 of the revised abstract to briefly explain our methods. To clarify the assumptions of the scenarios, we have added “Assuming that post-depositional sulfate-rich fluids interacted with the sediments,” and “Assuming no interactions,” in the revised manuscript.

Comment 5: line 10 Please clarify that post-depositional fluids are being referred to here.

Response to Comment 5

We have added “post-depositional **sulfate-rich** fluids” in line 10 of the revised manuscript.

Comment 6: line 121-126 – This depends on the rate of water-table lowering relative to the rate of groundwater flow. See Andrews-Hanna & Lewis, JGR-Planets, 2011 (or Horvath & Andrews-Hanna, GRL, 2017) for a detailed quantitative scenario. If the water-table is lowering slowly relative to the evaporation rate (i.e. evaporation is mostly balanced by upwelling of water from depth, as in Andrews-Hanna’s global deep groundwater circulation model) then the sediment can be bathed in pervasive groundwaters for a long time.

Response to Comment 6

In this part, we have discussed the way of loss of liquid water in the pores after the last wetting event based on the salinity of the final pore water. Even if the sediments were bathed in upwelling groundwater, and if salinity of the pore water was diluted with the groundwater, the interlayer composition of smectite should record high salinity (close to the saturation level of halite) upon evaporation of the pore water. This is the case because the interlayer composition of smectite records the pore-water compositions at the last time immediately before the loss of the pore water (less than one second before the loss of water) (see Response to Major Comment 3). Therefore, the mild salinity of the final pore water indicates that the pore water would have lost through freezing and subsequent sublimation, rather than evaporation.

Having said that, the original manuscript did not consider the dilution of NaCl by possible upwelling groundwater. Given this possibility, the estimated NaCl would be a lower limit of NaCl in the bottom water of the early lakes. To explain this, we have added the following sentences in the revised manuscript.

“Groundwater within terrestrial basalts contains typically low Na concentrations (10^{-4} – 10^{-3} mol/kg; see Methods). Given the possibility that the primary components could have been leached due to upwelling groundwater into Gale¹⁹, the estimated Na concentration would be a lower limit of the primary Na⁺ concentration in the bottom water of the early lakes. Nevertheless, we consider that the original Na⁺ concentration in the bottom water would not be remarkably higher than those in Table 1 because of lack of evidence for drying hypersaline lakes (e.g., desiccation cracks and rip-up chips)²⁵ at Yellowknife Bay¹⁴. We consider that if the early lakes existed, its Na⁺ concentration would have been on the same order of magnitude as that of the pore water in the last wetting event.” (lines 156 – 163 of p. 9-10)

Comment 7: line 179-183 - The absence of calcite is used as part of a thermodynamic argument. But Halevy & Schrag (GRL, 2009) show that small concentrations of CO₂ are enough to prevent calcite precipitation. Also Tosca et al. (Nature Geoscience, 2018) show experimentally that kinetic effects can prevent carbonate precipitation even at high pCO₂.

Response to Comment 7

As the reviewer pointed out, calcite formation is known to be kinetically inhibited in the presence of Mg²⁺ in solutions. On the other hand, aragonite is kinetically favored to form. Based on the reviewer's comment, we have done re-calculation using aragonite instead of calcite. The results are not changed significantly (see revised Fig. 3). The reviewer also mentioned the kinetics of formation of Fe-carbonate (siderite) (Tosca et al., 2018); however, the formation of siderite is not applicable to the discussion of Ca-carbonate. To explain this point, we have modified the following sentences in lines 570 – 572 of p. 26 of Methods of the revised manuscript.

“The formation of calcite, most stable calcium carbonate phase, is known to be kinetically inhibited in the presence of significant amount of Mg²⁺ in solutions⁶⁴. Instead, metastable aragonite is favored to form from the solutions⁶⁵ ..”

Comment 8: line 259 – It is not clear to me where the argument for “acidic-oxidizing conditions [...] were relatively short-lived” comes from.

Response to Comment 8

Short-lived interactions are required for persistence of redox disequilibria in the sediments. To clarify this, we have modified this sentence as follows in lines 261 – 263 of p. 14.

“Our results of persistence of redox disequilibria suggest that acidic-oxidizing episodes of alteration would have occurred at low temperatures and were relatively short-lived at Gale.”

Comment 9: line 329-331 – See my main comment above on charging-up the soil with Cl salts during dry periods.

Response to Comment 9

The recent re-analyses of Curiosity's XRD data suggest the no definitive detection of pyrrhotite (Morrison et al., 2018). Thus, we have deleted the discussion of preservation of detrital pyrrhotite, including this part.

Comment 10: line 341 – “unique constraints” – This is a very strong phrasing. Alternative explanations exist. Suggest “constraints” instead.

Response to Comment 10

This term has been modified as the reviewer suggested (line 335 of p. 17).

Comment 11: l. 347-348 - “proposed lake volume” – Here the lake volume is estimated using Palucis et al JGR Planets 2016. Palucis et al. refer to post-Mount-Sharp lakes. However Grotzinger et al. Science 2015 interpret YKB sediments as pre-Mount-Sharp. In other words, Palucis et al. interpret their lakes as much younger (a late-stage, relatively minor event) than the YKB lake event as interpreted by Grotzinger et al. Science 2015 (a Gale-filling, pre Mount Sharp event). For the lake volume estimation, the use of Grotzinger et al. Science 2015 interpretation would strengthen the lake-lifetime conclusions. Overall, the timing of YKB lake deposits relative to Mt. Sharp formation is not certain, although this doesn't affect the strong astrobiological interest of these deposits.

Response to Comment 11

This is correct. We have assumed the lake volume based on the estimates by Palucis et al. (2016) because of lack of direct constraints on the volume of the early Gale lakes at the time of deposition of the Yellowknife Bay sediments. Grotzinger et al. (2015) do not estimate directly the early lake volume, but they estimated the approximate areas of early lake within Gale Crater (Fig. 8B of their paper). According to the figure by Grotzinger et al. (2015), the lake area may have been roughly 30,000 km² (lake diameter ~100 km with the central peak ~45 km). Assuming ~200 m of the average lake depth, the lake water mass would have been $(3-6) \times 10^{15}$ kg. This lake water mass is about 3–6 times that estimated in the original manuscript based on Palucis et al. (2016). This would increase the duration of the lakes in a factor of 3–6, but would not change order of the duration given the uncertainties of other factors (e.g., evaporation rate of lake water, etc.). Thus, our conclusion of 10^5 – 10^6 years of lake duration does not change significantly. We have modified the descriptions of the lake volume estimate in Methods as follows in the revised manuscript (lines 623 – 626 in p. 28)

“Although the mass of lake water at the time of deposition of Yellowknife Bay has been poorly constrained, M_{lake} may become $(3-6) \times 10^{15}$ kg for a Gale basin-sized lake (i.e., diameter of ~100 km with the central peak with ~45 km diameter¹⁵, mean lake

depth of a few hundred meters).”

Comment 12: line 372-373 – Both the valley networks and the Al-rich surface clays mostly predate the formation of Gale crater. Please clarify.

Response to Comment 12

Agreed. To clarify this point, we have added the following description in line 368 – 371 of p. 18 of the revised manuscript.

“The estimated total duration of the early Gale lakes agrees with some formation timescales proposed for valley networks⁴⁵ and Al-rich surface clays⁴⁶ that formed in the late Noachian to the early Hesperian, **although the timing of formation of valley networks and Al-rich surface clays may not be coincident with the existence of the early lakes.**”

Comment 13: line 388-390 – Oxidants could also accumulate in upstream soils and be washed down during a wet event. Please clarify that possible oxidants include oxychlorine compounds (e.g., chlorate).

Response to Comment 13

To clarify this point, we have modified the sentence as follows in line 386 – 388 of p. 19 of the revised manuscript.

“we can infer that oxidants, **e.g., perchlorate, chlorate, ozone, and nitrate**, might have also accumulated in surface ice/frost **or upstream soils** during a cold period in the early Hesperian prior to warming.”

Comment 14: line 394-397. This is speculative.

Response to Comment 14

We have removed “and biochemistry” from this sentence since biochemistry would be too speculative.

Comment 15: line 407-411 – See Tosca et al. Nature Geoscience 2018 for counterarguments.

Response to Comment 15

We have added the information of Tosca et al. (2018) in this sentence as follows (line 409-411 of p. 19).

“**Although formation of Fe carbonate can be kinetically inhibited even at high**

CO₂ levels⁵¹, a previous bottom-up approach to estimate dissolved CO₂ provided an upper limit of P_{CO_2} levels as tens of mbar²⁸.”

Comment 16: line 415 – Both Turbet (PhD thesis 2018, http://www.lmd.jussieu.fr/~mturbet/these_Martin_Turbet/) and Steakley et al. (4th Intl Conf Early Mars 2017), find that warm climates last just a few years after a big impact. Gale is one of the last 150 km-sized impacts on Mars, are there any post-Gale impacts big enough to trigger a warming event of the required duration?

Response to Comment 16

The timing of the rewetting event is recently constrained as ~2 Ga based on the age of jarosite (Martin et al., 2017). We have added this information in the revised manuscript (line 413 – 417 of p. 20). An impact event occurred at 2 Ga would have been too small to cause sufficient warming to melt surface ice. Thus, we have deleted “an asteroidal impact” from the revised manuscript.

Comment 17: line 662-665 – See main comment above and Melwani Daswani & Kite (JGR Planets, 2017) for an alternative interpretation that would give much higher dissolved-Na concentrations.

Response to Comment 17

See our Response to Major Comment 2 above.

Comment 18: Figure 1, “Aeolis” is misspelt (twice)

Response to Comment 18

We appreciate the reviewer’s careful check. This has been modified in the revised manuscript.

Comment 19: Figure S10, on my printout the x-axis labels for panels (a) and (b) were in the wrong places.

Response to Comment 19

Original Supplementary Fig. 10 has been deleted from the revised manuscript owing to the revision of the methodology to estimate the water chemistry for each drilling site.

Reviewers' comments:

Reviewer #4 (Remarks to the Author):

The authors did a great job revising the manuscript. It could be considered for publication.

I have only minor notes that could be easily fixed:

- * Title: the words "inferred from water chemistry" could be modified to "inferred from mineral (smectite) composition" or something like that. (Water has not been analyzed.)
- * Line 78. English: addition of "with" is probably needed.
- * Lines 303-310: 'Crusts' need to be changed to 'crust'.
- * In one or two places, the words "almost zero" (about Na content) are useless without a value.

Reviewer #5 (Remarks to the Author):

The authors thoughtfully addressed my comments on the original manuscript. The methods are sound and the results are important for constraining lake water chemistry in ancient Gale crater. Modeling the d001 peaks of the smectite in the CheMin patterns of John Klein and Cumberland to derive interlayer cation composition and infer pore water chemistry is novel. The authors adequately addressed my previous comment that using halite to constrain pore water chemistry was not appropriate because CheMin did not detect halite in Yellowknife Bay, and they used the presence of akaganeite to constrain chloride content. One comment that the authors must still address is the difference in the d001 of the smectite in John Klein and Cumberland. The authors conclude that the pore water composition in John Klein and Cumberland were similar, but the d001 are significantly different (i.e., John Klein has a peak at 10 Å, whereas Cumberland has a peak at ~14 Å). If the pore water compositions were similar, then why are the d001 peaks different? I have a few other minor comments below, mostly just wordsmithing. Again, if the authors have questions about my comments, they should feel free to contact me, Elizabeth Rampe, at elizabeth.b.rampe@nasa.gov.

Line 13: Add the word "been" to the sentence: "...could have been generated..."

The term “clay rocks” is not commonly used. It could be changed to “clay-bearing rocks” or “mudstone.”

Line 27: I suggest using the term “interlayer sites” instead of “interlayers.”

Line 29: Suggested edit: “...all smectite possesses cation exchange capacity.”

Line 63, 66: Suggested edit: “evaporites” instead of “evaporates”

Line 78: The term “equilibrated” might not be the correct term since the mineral assemblage is not in equilibrium. Consider using the term “reacted” instead of “equilibrated.”

Line 101: Use “provide” instead of “provides.”

Line 119-120: Suggested edit: “This suggests that pore water compositions at the two adjacent sites would have been similar and may define the pore water chemistry throughout the Yellowknife Bay sediments.”

Line 150: Suggested edit: “generally retained” instead of “hardly lost”

Line 157: Suggested edit: “typically contains” instead of “contains typically”

Line 171-172: Suggested edit: “is compatible with” instead of “would not be incompatible with”

Line 187: The authors suggest akaganeite formed during rewetting events, but they used the presence of akaganeite to constrain salinity and pH of the pore-water chemistry in the no-interaction scenario. This seems contradictory. Please address this.

Line 418: Expand a bit on the location of the Mojave2 sample (e.g., “...is consistent with the youthful age of jarosite in the Mojave2 sample, drilled from a lacustrine mudstone ~60 m higher in the stratigraphic section than the Yellowknife Bay sediments.”

Line 432: "Pseudo-Voigt" is misspelled.

Review of the 31 July 2019 submission of “Semiarid climate and hyposaline lake on early Mars inferred from water chemistry at Gale,” by Fukushi et al., submitted to Nature Communications.

Review by Edwin Kite, University of Chicago (I review non-anonymously).

I have read the response to my earlier comments, but I have not re-read the entire manuscript, nor have I thoroughly read the responses to the comments of reviewers #4 and #5.

The responses to the major comment 1, the major comment 3, and the minor comments from my earlier review are all satisfactory. The response to major comment 2 (lake timescales) from my earlier review is incomplete. (My comment may have been worded in a confusing way, because the authors’ response does not take account of the point that was intended). This comment refers to line 615-653 of the revised manuscript. The workflow here (“Estimates of total duration for hydrological activities at Gale”) assumes “typical Na^+ concentrations in terrestrial surface/groundwater within terrestrial basaltic rocks”. This is a reasonable approach for groundwater upwelling. However, it is a scenario-dependent conclusion that needn’t apply for precipitation runoff. My concern here is that terrestrial analogy might underestimate the salinity of runoff on Early Mars. As an example, soil on Mars today is quite salty. If a lot of rain fell on Mars soil today, then the runoff would be saltier than typical salinity for terrestrial surface/groundwater within terrestrial basaltic rocks. If on Early Mars relatively brief periods of runoff were intercalated with long periods with no runoff, then the soil would get salty during the no-runoff periods via interaction of volcanic $\text{HCl}/\text{H}_2\text{SO}_4$ e.t.c. with the soil. I agree with the authors that the interaction of volcanic $\text{HCl}/\text{H}_2\text{SO}_4$ with the soil would lead to quick ($<10^2$ yr) neutralization. However the soluble salts resulting from neutralization (chlorides, sulfates, e.tc.) would be swiftly flushed down to the lake during subsequent periods with runoff. (This point has been made by Melwani Daswani & Kite JGR 2017, in the context of the chloride lakes). In this scenario, the Na^+ supply rate is higher, and so the lake lifetime can be much lower than calculated by the authors. Therefore, the 10^4 - 10^6 -year long semiarid climate stated in the abstract to be true “in any scenario” is in fact only true for some scenarios but not others. This possibility should be acknowledged in the manuscript and in the abstract, because it makes the conclusion about lake lifetime scenario-dependent.

I notice that reviewer #4 suggests “no lake” as a possibility for Yellowknife Bay, however the textural evidence (e.g. Siebach et al. JGR, Stack et al. JGR 2014) indicates very high initial porosity, which is very hard to explain without a lake.

Minor English comment: On line 30, “instantaneous” means instantly, however “sub-second” means not instantly! Suggest replace “instantaneous” with “rapid”, or “very rapid.”

Responses to comments from Reviewer #4 for ‘Semiarid climate and hyposaline lake on early Mars inferred from reconstructed water chemistry at Gale’ by Fukushi, K. et al. (MS# NCOMMS-18-21962B)

We are grateful to reviewer #4 for the constructive suggestions. Below, we give our responses in turn following each comment, with the reviewers’ comments being in Arial font and our responses being in Times font.

Comment 1: Title: the words “inferred from water chemistry” could be modified to “inferred from mineral (smectite) composition” or something like that. (Water has not been analyzed.)

Response to Comment 1: According to the reviewer’s comment, we have changed the title as follows (the term “reconstructed” is added):

“Semiarid climate and hyposaline lake on early Mars inferred from **reconstructed water chemistry at Gale”**

We hope to keep the term “water chemistry” in the title because we believe that the novel and important advance by the present study is first to reconstruct quantitative water chemistry (major composition, pH, and redox) using the in-situ observational data of smectite’s interlayer compositions. We consider that the addition of the term “reconstructed” could send a message to reader that water chemistry is not directly analyzed. We hope that this title is satisfactory to the reviewer.

Comment 2: Line 78. English: addition of “with” is probably needed.

Response to Comment 2: The word “with” is added in Line 77, as suggested.

Comment 3: Lines 303-310: ‘Crusts’ need to be changed to ‘crust’.

Response to Comment 3: The word “crusts” is changed to “crust” in Line 314, as suggested.

Comment 4: In one or two places, the words “almost zero” (about Na content) are useless without a value.

Response to Comment 4: The Na content is measured to be less than the detection limit (<0.05 wt%) (Nachon et al., 2014; ref. 14). We have added the concrete number of “(<0.05 wt%)” in Line 159 of the revised manuscript.

Responses to comments from Dr. Elizabeth B. Rampe (the reviewer #5) for ‘Semiarid climate and hyposaline lake on early Mars inferred from reconstructed water chemistry at Gale’ by Fukushi, K. et al. (MS# NCOMMS-18-21962B)

We are grateful to Dr. Elizabeth B. Rampe for the constructive comments and suggestions. Below, we give our responses in turn following each comment, with the reviewers’ comments being in Arial font and our responses being in Times font.

Major Comment 1: One comment that the authors must still address is the difference in the d001 of the smectite in John Klein and Cumberland. The authors conclude that the pore water composition in John Klein and Cumberland were similar, but the d001 are significantly different (i.e., John Klein has a peak at 10 Å, whereas Cumberland has a peak at ~14 Å). If the pore water compositions were similar, then why are the d001 peaks different?

Response to Major Comment 1: Although the d001 peaks positions of John Klein and Cumberland are different, our results of the peak deconvolution analyses show that contributions of the predominant cations, i.e., Na⁺ for John Klein and Mg²⁺ for Cumberland, to the total interlayer cations are almost half (i.e., Na = 0.48 for John Klein and Mg = 0.51 for Cumberland: Supplementary Table 2). This means that both of the smectites are close to a transition of the predominant cation compositions in the interlayer sites. Previous cation-exchange experiments showed that even using same smectite samples, a significant change in the predominant cation compositions of the interlayer sites occurs for a range of solution compositions, if the smectites are close to a transition of the predominant cation compositions (e.g. Tournassat et al 2009 SSSAJ). In other words, in a solution within this range of compositions, smectites with the different cation occupancies can co-exist. Since both of smectites in John Klein and Cumberland are close to a transition of the predominant cation compositions, co-existence of these smectites is possible in solutions with very similar, or even common, chemical compositions. To explain this point, we have added the following sentences in the figure caption of Fig. 2 of the revised manuscript.

“The peak deconvolution analyses show that the contribution of the predominant cations (Na⁺ for John Klein and Mg²⁺ for Cumberland) to the total interlayer cations are almost half (i.e., ~0.5) (Supplementary Table 2). This suggests that both of the smectites from John Klein and Cumberland are close to transition of the predominant cations in the interlayer sites. In a solution within a range of compositions, a transition of the predominant cation occurs in smectite⁵⁵. Within this range of solution composition, smectites with different of cation occupancies can co-exist. Thus, the fact that the top of the peaks of the 001 reflection of John Klein and Cumberland are different does not contradict with the reconstructed similar compositions of pore water at these sites (see the main text).”

Comment 2: Line 13: Add the word “been” to the sentence: “...could have been generated...”

Response to Comment 2: The word “been” is added in Line 13, as suggested.

Comment 3: The term “clay rocks” is not commonly used. It could be changed to “clay-bearing rocks” or “mudstone.”

Response to Comment 3: All of the terms “clay rocks” are changes to “clay-bearing rocks” throughout the revised manuscript: i.e., in Lines 24, 25, 37, and 38.

Comment 4: Line 27: I suggest using the term “interlayer sites” instead of “interlayers.”

Response to Comment 4: The term “interlayers” is changed to “interlayer sites” in Line 27.

Comment 5: Line 29: Suggested edit: “...all smectite possesses cation exchange capacity.”

Response to Comment 5: The term “ability” is changed to “capacity” in Line 29, as suggested.

Comment 6: Line 63, 66: Suggested edit: “evaporites” instead of “evaporates”

Response to Comment 6: We have changed to “evaporites” in Lines 62 and 65.

Comment 7: Line 78: The term “equilibrated” might not be the correct term since the mineral assemblage is not in equilibrium. Consider using the term “reacted” instead of “equilibrated.”

Response to Comment 7: The term “equilibrated” is changed to “reacted” in Line 76, as suggested.

Comment 8: Line 101: Use “provide” instead of “provides.”

Response to Comment 8: We have changed “provides” to “provide” in Line 100.

Comment 9: Line 119-120: Suggested edit: “This suggests that pore water compositions at the two adjacent sites would have been similar and may define the pore water chemistry throughout the Yellowknife Bay sediments.”

Response to Comment 9: Agreed. The above description is revised as the reviewer suggested in Lines 124–125.

Comment 10: Line 150: Suggested edit: “generally retained” instead of “hardly lost”

Response to Comment 10: We have changed “hardly lost” to “generally retained” in Line 156, as suggested.

Comment 11: Line 157: Suggested edit: “typically contains” instead of “contains typically”

Response to Comment 11: We have changed “contains typically” to “typically contains” in Line 163, as suggested.

Comment 12: Line 171-172: Suggested edit: “is compatible with” instead of “would not be incompatible with”

Response to Comment 12: The sentence has been edited to “is compatible” in Line 177.

Comment 13: Line 187: The authors suggest akaganeite formed during rewetting events, but they used the presence of akaganeite to constrain salinity and pH of the pore-water chemistry in the no-interaction scenario. This seems contradictory. Please address this.

Response to Comment 13: The description on akaganeite formation (Line 187 of the original manuscript) are only for the case of the full-interaction scenario. This can be seen in the original description “*although the akaganeite is most likely to have been formed during the rewetting events using primary Cl in the sediments in the full-interaction scenario (see below)*” (now in Lines 192–194 of Section of “The full-interaction scenario” of the revised manuscript).

In the no-interaction scenario, as pointed out by the reviewer, akaganeite needs to be formed other than rewetting event, as mentioned in the original manuscript (now in Lines 309–311 of Section of “The no-interaction scenario” of the revised manuscript): “*In the no-interaction scenario, the observed redox disequilibria (i.e., co-existence of akaganeite, magnetite, and Fe^{2+} saponite) need to be formed other than intrusion of SO_4^{2-} -rich fluids.*”. We have discussed one possibility to form akaganeite and other Fe-bearing minerals in the no-interaction scenario in the original manuscript (now in Lines 312–318 of the revised manuscript): “*One possibility to generate the redox disequilibria in the no-interaction scenario is interactions of upwelling groundwater at Gale¹⁹ with initially oxidized sediments¹⁰ after deposition. Groundwater within Martian crust would become reducing and alkaline¹⁰. If reducing groundwater upwelled into oxidized sediments at Gale, Fe^{2+} saponite would have formed through reduction of oxidized smectite, e.g., nontronite, by groundwater. Akaganeite in the sediments could have been also reduced into magnetite by groundwater; however, this conversion might have proceeded only incompletely due to kinetics.*”. Thereby, as shown in these sentences, we consider that akaganeite was formed within the lake, or transported as a

detrital component, under oxidizing surface conditions, and then they were partly reduced to magnetite in reducing subsurface by interactions with upwelling groundwater. Thereby, in the no-interaction scenario, we use akaganeite to constrain the water chemistry of early post-depositional stages immediately before the disappearance of water (not the rewetting event).

Although we have described about akaganeite formation in the no-interaction scenario in the original manuscript, we agree that the situations are complicated. To further avoid the misunderstanding and confusing of the readers, we have added the following sentence in Line 288–289 of Section of “The no-interaction scenario” of the revised manuscript.

“In the no-interaction scenario, akaganeite needs to be formed before intrusion of SO_4^{2-} -rich fluids given their occurrence in the matrix of the sediments¹¹.”

Comment 14: Line 418: Expand a bit on the location of the Mojave2 sample (e.g., “...is consistent with the youthful age of jarosite in the Mojave2 sample, drilled from a lacustrine mudstone ~60 m higher in the stratigraphic section than the Yellowknife Bay sediments.”

Response to Comment 14: We have added the description of “drilled from a lacustrine mudstone ~60 m higher in the stratigraphic section than the Yellowknife Bay sediments.” in Lines 428–429, as suggested.

Comment 15: Line 432: “Pseudo-Voigt” is misspelled.

Response to Comment 15: We appreciate the reviewer’s careful check. We have fixed the typo in the revised manuscript.

Responses to comments from Dr. Edwin Kite (the reviewer #6) for ‘Semiarid climate and hyposaline lake on early Mars inferred from reconstructed water chemistry at Gale’ by Fukushi, K. et al. (MS# NCOMMS-18-21962B)

We are grateful to Dr. Edwin Kite for the constructive comments and suggestions. Below, we give our responses in turn following each comment, with the reviewers’ comments being in Arial font and our responses being in Times font.

Major Comment 1: The responses to the major comment 1, the major comment 3, and the minor comments from my earlier review are all satisfactory. The response to major comment 2 (lake timescales) from my earlier review is incomplete. (My comment may have been worded in a confusing way, because the authors’ response does not take account of the point that was intended). This comment refers to line 615-653 of the revised manuscript. The workflow here (“Estimates of total duration for hydrological activities at Gale”) assumes “typical Na⁺ concentrations in terrestrial surface/groundwater within terrestrial basaltic rocks”. This is a reasonable approach for groundwater upwelling. However, it is a scenario-dependent conclusion that needn’t apply for precipitation runoff. My concern here is that terrestrial analogy might underestimate the salinity of runoff on Early Mars. As an example, soil on Mars today is quite salty. If a lot of rain fell on Mars soil today, then the runoff would be saltier than typical salinity for terrestrial surface/groundwater within terrestrial basaltic rocks. If on Early Mars relatively brief periods of runoff were intercalated with long periods with no runoff, then the soil would get salty during the no-runoff periods via interaction of volcanic HCl/H₂SO₄ e.t.c. with the soil. I agree with the authors that the interaction of volcanic HCl/H₂SO₄ with the soil would lead to quick (<10² yr) neutralization. However the soluble salts resulting from neutralization (chlorides, sulfates, e.tc.) would be swiftly flushed down to the lake during subsequent periods with runoff. (This point has been made by Melwani Daswani & Kite JGR 2017, in the context of the chloride lakes). In this scenario, the Na⁺ supply rate is higher, and so the lake lifetime can be much lower than calculated by the authors. Therefore, the 10⁴-10⁶-year long semiarid climate stated in the abstract to be true “in any scenario” is in fact only true for some scenarios but not others. This possibility should be acknowledged in the manuscript and in the abstract, because it makes the conclusion about lake lifetime scenario-dependent. I notice that reviewer #4 suggests “no lake” as a possibility for Yellowknife Bay, however the textural evidence (e.g. Siebach et al. JGR, Stack et al. JGR 2014) indicates very high initial porosity, which is very hard to explain without a lake.

Response to Major Comment 1: We appreciate the reviewer for the detailed explanation and suggestion. We now understand the reviewer's concern correctly; that is, if salts were contained in surface soils, as same as today's Martian soils, and if these salts were transported into the lakes through surface runoff, the required lake duration to achieve the estimated salinity may be significantly shortened.

This is certainly correct. We have estimated a quantitative effect of this process. The present-day soils on Mars contain ~1 wt% of Na-bearing salts (chloride and perchloride) according to the in-situ analysis by the Phoenix lander (Hecht et al., 2009, Science). Given both $\sim(2-3) \times 10^{10} \text{ m}^2$ of the catchment area of the early lake (= the surface area of Gale Crater's cavity) and $\sim 3 \times 10^3 \text{ kg/m}^3$ of soil density, the maximum mass of transported salts through surface runoff would be $\sim 5 \times 10^{11} \text{ kg}$ for a 1 m-thick soil layer and $\sim 5 \times 10^{12} \text{ kg}$ for a 10 m-thick soil layer. According to the infiltration theory of rain into dried soils (Brutsaert 2005, in *Hydrology: An Introduction*, Cambridge Univ. Press), persistent existence of surface water due to rain fall and/or snow melting needs to continue for ~20 hours and ~2000 hours to reach soils (assuming volcanic loam) in depth of 1 m and 10 m, respectively. Since these timescales would roughly correspond to proposed timescale of precipitation/snow melting cycle on Mars (Wordsworth et al., 2015 JGR), we consider that the wetting depth of soil layer over Gale Crater would be at most ~10 m.

On the other hand, given ~0.03–0.2 mol/kg of lake salinity, C_{lake} in Eq. (17), and $(3-6) \times 10^{15} \text{ kg}$ of lake water mass, M_{lake} in Eq. (17) in *Methods*, the total salt mass in the lakes would become $\sim(1-10) \times 10^{12} \text{ kg}$. This suggests that if the soils in the catchment area of the lakes contained salts comparable to that of today's soils, saline surface runoff could provide, at most, ~50% or less of the total salinity of the lakes. Considering the salt inputs from saline surface runoff, the required lake duration may be shortened in a factor of ~2 in maximum.

Based on the reviewer's comment on the possibility of saline surface runoff, we have removed "In any scenario" from the abstract. Concerning the main text, we have deleted the descriptions on accumulation of acids on the surface, which were added in response to the reviewer's previous comment. Instead, we have added the following descriptions on supply of surface salts through surface runoff in Lines 372 – 378 of the revised manuscript:

"Similar to soils on present-day Mars, Na-bearing salts would have contained in soils within Gale Crater before the appearance of the lakes². In warm periods, surface runoff could have transported them to the lake², providing salinity efficiently to the lake water. With this regard, our estimate of lake duration would be an upper limit. Assuming ~1 wt% of the salt content based on present-day soils⁴⁶, such an accumulation of surface salts through surface runoff could explain ~50% of the estimated salinity of the lakes in maximum (see Methods). Thus, the required lake duration could be shortened if saline surface runoff occurred."

In Methods, we have added the following descriptions on our estimate of the effect of saline surface runoff (Lines 667 – 684):

“Martian soils, in general, contain abundant Na-bearing salts (e.g., chloride and perchlorate)⁴⁶. As mentioned in the main text, surface runoff could have transported surface salts accumulated within soils efficiently to the early Gale lake in warm periods². We estimate a quantitative effect of this process. Present-day soils on Mars typically contain ~1 wt% of salts⁴⁶. Given $\sim(2-3) \times 10^{10} \text{ m}^2$ of the catchment area of the early lake (= the surface area of Gale Crater’s cavity) and $\sim 3 \times 10^3 \text{ kg/m}^3$ of bulk density of soils, the maximum mass of transported salts through surface runoff can be $\sim 5 \times 10^{11} \text{ kg}$ assuming a 1 m-thick soil layer and $\sim 5 \times 10^{12} \text{ kg}$ for a 10 m-thick soil layer. According to the infiltration theory of rain into dried soils⁷⁰, persistent inputs of surface water due to rain fall and/or snow melting needs to continue for ~20 hours and ~2000 hours to reach the soils (assuming volcanic loam) in depth of 1 m and 10 m, respectively. Since this timescale would roughly correspond to those of precipitation/snow melting on early Mars⁵⁰, we consider that the wetting depth within soils upon rain fall and/or snow melting over Gale Crater would be at most ~10 m. On the other hand, given ~0.03–0.2 mol/kg of lake salinity (see the main text) and $(3-6) \times 10^{15} \text{ kg}$ of lake water mass (see eq. (17)), the total salt mass in the lakes would become $\sim(1-10) \times 10^{13} \text{ kg}$. This suggests that if the soils in the catchment area of the lakes contained salts comparable to that of today’s soils, saline surface runoff could provide ~50% of the total salinity of the lakes in maximum. Assuming the saline surface runoff², thereby, the required lake duration could be reduced in a factor of ~2 or less.”

Comment 2: On line 30, “instantaneous” means instantly, however “sub-second” means not instantly! Suggest replace “instantaneous” with “rapid”, or “very rapid.”

Response to Comment 2: We appreciate this suggestion. The term is changed to “very rapid” in Line 30.

REVIEWERS' COMMENTS:

Reviewer #5 (Remarks to the Author):

Thanks to the authors for explaining the reason for the differences in 001 peak position. I am surprised that significant differences in smectite interlayer composition can occur in solutions with similar composition, but the authors cited a study to demonstrate this. The authors have fully addressed my comments, and I recommend the manuscript be accepted for publication in its current form.

Reviewer #6 (Remarks to the Author):

I have read the authors' response to my comments. (I did not read the authors' response to the other reviewer's comments). I am happy with the response and with the changes to the text. I did not read the revised manuscript because the changes are minor. I am happy for this manuscript to be published.